# Origins and impact of extrachromosomal DNA

Chris Bailey[1,20], Oriol Pich[1,20], Kerstin Thol[2,3], Thomas B. K. Watkins[4,5], Jens Luebeck[6], Andrew Rowan[1], Georgia Stavrou[2,7], Natasha E. Weiser[4,8], Bhargavi Dameracharla[4], Robert Bentham[2,3], Wei-Ting Lu[1], Jeanette Kittel[2,7], S. Y. Cindy Yang[9], Brooke E. Howitt[4], Natasha Sharma[2], Maria Litovchenko[2,3], Roberto Salgado[10,11], King L. Hung[8], Alex J. Cornish[12], David A. Moore[1,2,13], Richard S. Houlston[12], Vineet Bafna[6], Howard Y. Chang[8], Serena Nik-Zainal[14], Nnennaya Kanu[2], Nicholas McGranahan[2,3], Genomics England Consortium*, Adrienne M. Flanagan[15,16], Paul S. Mischel[4,5✉], Mariam Jamal-Hanjani[2,7,17✉] & Charles Swanton[1,2,17✉]

Extrachromosomal DNA (ecDNA) is a major contributor to treatment resistance and poor outcome for patients with cancer[1,2]. Here we examine the diversity of ecDNA elements across cancer, revealing the associated tissue, genetic and mutational contexts. By analysing data from 14,778 patients with 39 tumour types from the 100,000 Genomes Project, we demonstrate that 17.1% of tumour samples contain ecDNA. We reveal a pattern highly indicative of tissue-context-based selection for ecDNAs, linking their genomic content to their tissue of origin. We show that not only is ecDNA a mechanism for amplification of driver oncogenes, but it also a mechanism that frequently amplifies immunomodulatory and inflammatory genes, such as those that modulate lymphocyte-mediated immunity and immune effector processes. Moreover, ecDNAs carrying immunomodulatory genes are associated with reduced tumour T cell infiltration. We identify ecDNAs bearing only enhancers, promoters and lncRNA elements, suggesting the combinatorial power of interactions between ecDNAs in *trans*. We also identify intrinsic and environmental mutational processes linked to ecDNA, including those linked to its formation, such as tobacco exposure, and progression, such as homologous recombination repair deficiency. Clinically, ecDNA detection was associated with tumour stage, more prevalent after targeted therapy and cytotoxic treatments, and associated with metastases and shorter overall survival. These results shed light on why ecDNA is a substantial clinical problem that can cooperatively drive tumour growth signals, alter transcriptional landscapes and suppress the immune system.

ecDNA is a common origin for amplified oncogenes across human cancer[1,2]. Individual ecDNAs are large (typically greater than 500 kilobases (kb) in size), mobile, gene-containing (and regulatory-region-containing) circular DNA particles that can be found in the nuclei of many cancer cells[3,4]. The non-chromosomal inheritance and resultant random segregation of ecDNA during cell division[3,5] promotes high oncogene copy number and intratumoural genetic heterogeneity, facilitating rapid genome change to drive treatment resistance[6,7]. Highly accessible chromatin of ecDNA, and altered gene-regulatory architecture resulting from its circular structure, as well as the propensity to form hubs that promote intermolecular cooperation, generates high-level oncogenic transcription, contributing to tumorigenesis[8,9]. ecDNAs can arise early during tumorigenesis, such as in the transformation from Barrett's high-grade dysplasia to oesophageal adenocarcinoma and have also been detected later in the course of disease progression[4,10,11]. As ecDNAs drive treatment resistance through rapid genome alterations that may consequently accelerate disease progression[6,7], it is important to distinguish ecDNA from other

[1]Cancer Evolution and Genome Instability Laboratory, The Francis Crick Institute, London, UK. [2]Cancer Research UK Lung Cancer Centre of Excellence, University College London Cancer Institute, London, UK. [3]Cancer Genome Evolution Research Group, Cancer Research UK Lung Cancer Centre of Excellence, University College London Cancer Institute, London, UK. [4]Department of Pathology, Stanford University, Stanford, CA, USA. [5]Sarafan ChEM-H, Stanford University, Stanford, CA, USA. [6]Department of Computer Science and Engineering, University of California at San Diego, La Jolla, CA, USA. [7]Cancer Metastasis Laboratory, University College London Cancer Institute, London, UK. [8]Center for Personal Dynamic Regulomes, Stanford University, Stanford, CA, USA. [9]Department of Pediatrics, Stanford University, Palo Alto, CA, USA. [10]Department of Pathology, ZAS Hospitals, Antwerp, Belgium. [11]Division of Research, Peter MacCallum Cancer Centre, Melbourne, Victoria, Australia. [12]Division of Genetics and Epidemiology, The Institute of Cancer Research, London, UK. [13]Department of Cellular Pathology, University College London Hospitals, London, UK. [14]Academic Department of Medical Genetics, School of Clinical Medicine, University of Cambridge, Cambridge, UK. [15]Research Department of Pathology, University College London, UCL Cancer Institute, London, UK. [16]Department of Histopathology, Royal National Orthopaedic Hospital, Stanmore, UK. [17]Department of Medical Oncology, University College London Hospitals, London, UK. [20]These authors contributed equally: Chris Bailey, Oriol Pich. *A list of authors and their affiliations appears at the end of the paper. ✉e-mail: pmischel@stanford.edu; m.jamal-hanjani@ucl.ac.uk; charles.swanton@crick.ac.uk

types of focal amplification. The ability to computationally decipher whether an amplified gene has arisen from ecDNA, and to deconvolute its structure and sequence[12], facilitates the examination of ecDNA content across human cancer. The clinically annotated whole-genome sequencing Genomics England (GEL) 100,000 Genomes Project (100kGP) provides an opportunity to decipher the landscape of ecDNA in human cancer, and to shed light on the underlying molecular processes in which it develops, as well as gaining insight into its clinical impact.

## The body map of ecDNA

We analysed 15,832 samples of whole-genome-sequenced cancers from 14,778 patients recruited across 13 UK National Health Service Genomic Medicine Centres as part of GEL 100kGP (GEL v12 data release)[13]. We used AmpliconArchitect followed by AmpliconClassifier, computational tools that have been shown to identify ecDNA from whole-genome sequencing data[4,10,12] (Fig. 1a and Extended Data Fig. 1a,b). These tools detect amplifications that arose as ecDNAs. To confirm our findings, we conducted fluorescence in situ hybridization (FISH) using *MDM2*, *CDK4*, *PDGFRA* and *MYC* oncogene probes on six dedifferentiated liposarcoma tissue samples, four osteosarcoma tissue samples and one angiosarcoma tissue sample in the GEL cohort that were available for further analysis (Fig. 1a and Extended Data Fig. 2).

We classified 39 solid and haematological tumour subtypes and quantified focal amplifications in 15,832 samples from 14,778 patients (Supplementary Table 1). Of the 14,778 patients with cancer in the study, 1,800 were recorded as receiving hormonal (*n* = 27), immunotherapy (*n* = 57), targeted (*n* = 415) and/or cytotoxic chemotherapy (*n* = 1,653) treatment before biopsy. Staging information was available for 10,780 (72.9%) patients, with 836 (5.7%) patients recorded as having stage 4 disease (Extended Data Fig. 1c). Focal amplifications were defined as regions of the genome between 50 kb and 20 Mb in size, with a minimum copy number of 4.5 and twice the estimated ploidy of the tumour. Tumour purities ranged from 10% to 95% with a mean of 50.1% (Extended Data Fig. 3a). This large-scale analysis enabled us to develop a data-rich map of ecDNA frequency and contents across the human body (Fig. 1c), and resolve the mutational processes, genomic context and clinical implications of ecDNA across multiple cancers. A total of 4,716 unique ecDNAs were identified from 2,532 ecDNA-positive tumours.

ecDNA amplifications were detected in 17.1% of tumour samples, with widely varying frequencies by cancer type, copy number and size (Fig. 1c, Extended Data Fig. 3b–e and Supplementary Tables 2–4). ecDNA was detected in 54.9% of liposarcomas (*n* = 82, 95% confidence interval (CI) 44.7–65.8%), 49.1% of glioblastoma (*n* = 291, 95% CI 43.3–55.0%), 46.4% of HER2+ breast cancer (HER2+ BRCA, *n* = 196, 95% CI 39.3–53.7%; examples of which we were able to identify through FISH in an independent cohort; Extended Data Fig. 4), 37.9% of upper gastrointestinal adenocarcinomas and 22.4% of lung squamous cell carcinomas, 24.6% of bladder cancers and 20.4% of ovarian cancers, among others (Fig. 1d). Some tumour types had a very low prevalence of ecDNA, including oligodendrogliomas, in which ecDNA was not detected (*n* = 57). Further, the amplification of specific oncogenes varied greatly by tissue type (Supplementary Table 1). These results reveal a strong impact of tissue lineage on the frequency and content of ecDNAs.

In 36/37 tumour types where ecDNA was detected, focal amplifications had a higher estimated copy number when derived from ecDNA compared to chromosomal amplifications (Extended Data Fig. 5a). Most ecDNAs arose from a locus on one chromosome (89.9% *n* = 3,705). Some ecDNAs were composed from genes from different chromosomes and were mostly seen in sarcomas and breast cancers (Extended Data Fig. 5b,c). Of note, ecDNA derived from chromosomal translocations observed in breast cancer such as t(8;11), t(8;17) and t(11;17) might arise through the recently described translocation–bridge mechanism[14].

ecDNAs often contained more than a single oncogene on the same ecDNA (46%), primarily driven by their oncogene proximity on the native chromosome from which the ecDNA arose (Extended Data Fig. 6a,b). We also detected tumours with multiple ecDNA species present at different copy-number states bearing distinct oncogenes (Extended Data Fig. 6c–e).

Owing to its non-chromosomal inheritance[5], ecDNA promotes intratumoural genetic heterogeneity. In 578 patients, multiple regions from the same tumour were sampled, with ecDNA detected in 151 tumours (26.1%). Controlling for tumour type, the odds of detecting ecDNA were 2.6 times greater when 2 regions of the same tumour were sampled (odds ratio (OR) 2.6, 95% CI 2.2–3.1; Extended Data Fig. 7a). Moreover, in more than 60% of tumours for which multiregion sequencing was carried out, ecDNA was detected in only a subset of regions (regional; Extended Data Fig. 7b).

## Selection of ecDNA-associated oncogenes

Significant oncogene enrichment on ecDNA against a permuted background (proportion 0.031, *P* < 0.0001; Extended Data Fig. 7c and Methods) was detected, with a greater propensity to amplify oncogenes on ecDNA relative to focal chromosomal amplifications (Extended Data Fig. 7d) and a higher oncogene count per ecDNA when matched for copy number (Extended Data Fig. 7e). Further, genes recurrently amplified on ecDNA were more likely to be oncogenes compared with genes on chromosomal amplifications (Extended Data Fig. 7d), even when matched for amplification size (Extended Data Fig. 7f). These data, along with the non-chromosomal inheritance of ecDNAs during cell division[5], are consistent with evolutionary selection for ecDNAs encoding oncogenes[15].

Oncogenes encoded on ecDNA were associated with a higher copy number than non-ecDNA-driven focal amplifications (Extended Data Fig. 8a). We found examples of well-established driver oncogenes more frequently amplified on ecDNA than on chromosomes, including *FGFR2* (proportion 0.63, median amplification copy number = 16.0), *MDM2* (0.58, 13.7) and *CDK4* (0.56, 14.0). Across all tumour types, oncogenes in the RTK–RAS (*EGFR*, *ERBB2* and *FGFR1*), TP53 (*MDM2*) and cell cycle (*CCND1* and *CDK4*) pathways were most commonly amplified on ecDNA (Extended Data Fig. 8b). Many of the amplified copies of these driver oncogenes contained high copy amplification of missense mutations and in the case of *CDK4* and *EGFR*, occurring pre-ecDNA formation (Fig. 1d).

To further assess the strength of oncogene selection, we analysed the ratio of non-synonymous (d*N*) to synonymous (d*S*) substitutions in relation to missense, nonsense and essential splice mutations. Genes with a high d*N*/d*S* ratio are under positive selection (Methods)[16]. We compared the frequency at which genes were amplified in the GEL cohort with a mutation-based signal of positive selection as derived from the d*N*/d*S* ratio (Methods and Extended Data Fig. 8c). We then compared mutation-based positive selection between non-amplified, ecDNA-amplified and chromosomal-amplified genes, and found that *EGFR* and *ERBB2* mutations are under more potent selection when amplified (Fig. 1e and Supplementary Table 5). These results indicate that ecDNAs containing driver mutations in oncogenes are under strong evolutionary pressure. It was not unexpected that 65.7% of tumours with detected ecDNA contain oncogenes on those ecDNAs (Bushman cancer gene list (http://www.bushmanlab.org/links/genelists); Fig. 2a–c). The fraction of tumours with oncogenes on ecDNA using the Cancer Gene Census (https://www.sanger.ac.uk/data/cancer-gene-census/) was 51%, reflecting differences in the inclusivity of these lists.

## ecDNAs contain immunomodulatory genes

Previous data have suggested that patients with ecDNA-driven cancers are less likely to respond to immune checkpoint inhibitors as they may

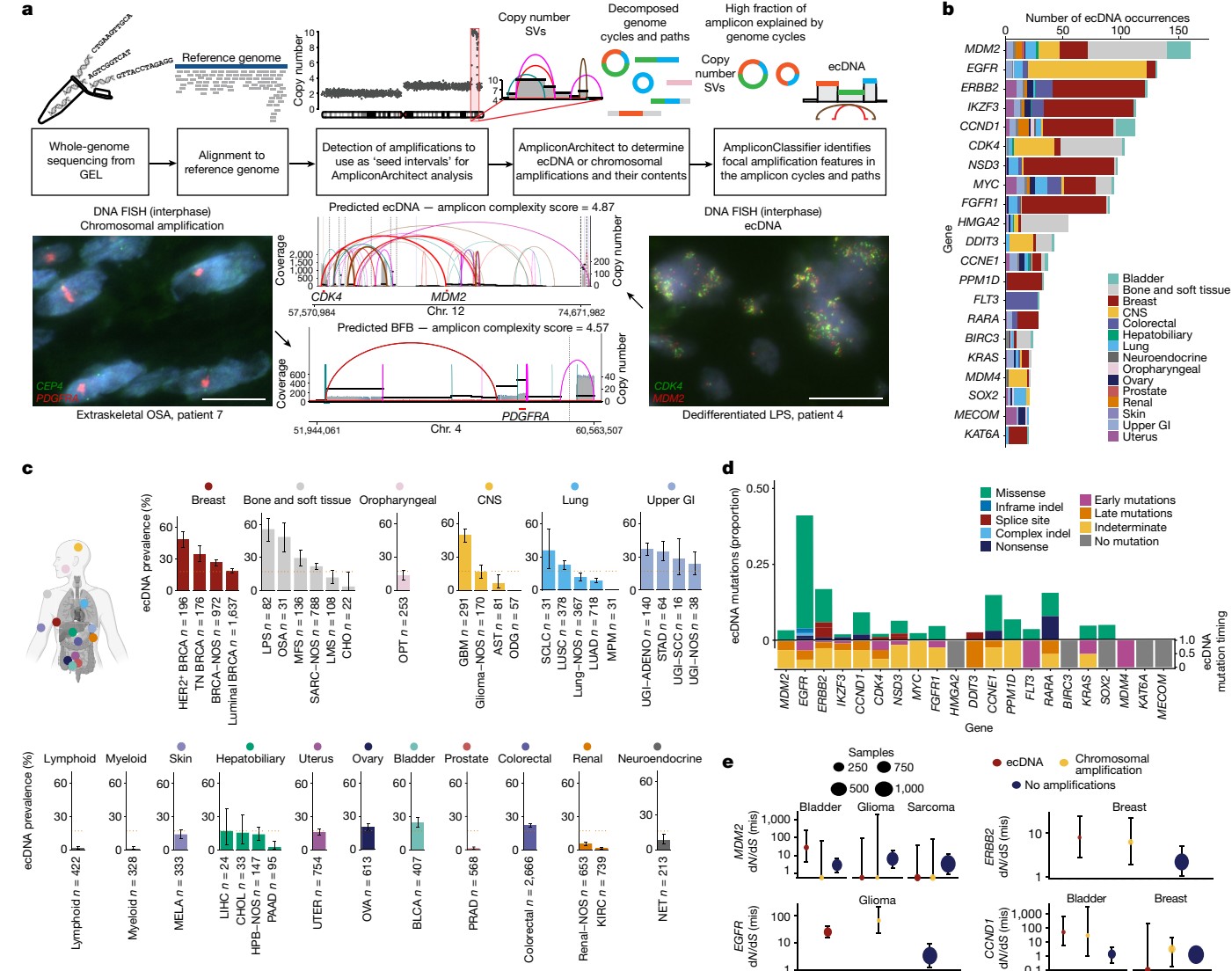

**Fig. 1 | The body map of ecDNA prevalence across 39 tumour types. a**, The analysis pipeline used to process the GEL cohort (top), with representative FISH images and AmpliconArchitect structural variant (SV) views from two GEL patients (bottom). The examples show amplicons predicted to be a chromosomal amplification consistent with its FISH image (left) and an ecDNA consistent with its FISH image (right). Scale bar, 20 μm. **b**, Bar plot showing number of occurrences of ecDNA containing oncogenes with the colour of the bar indicating the number of cases from each tissue type. **c**, Body map of cancer-type-specific ecDNA prevalence. Each sub-panel shows the prevalence of ecDNA (*y* axis) in cancer types specific to a particular tissue type (*x* axis) as shown in the body map schematic. The orange dotted line represents the median ecDNA-driven amplification prevalence across the entire cohort. The error bars represent the 95% confidence interval for the population proportion. **d**, Stacked bar plots displaying the proportion of types of non-synonymous mutations observed in the oncogenes present on ecDNA (top) and the proportion of these non-synonymous mutations in different timing categories (bottom; Methods). Only the mutations affecting the 21 oncogenes most

commonly present on ecDNA are shown. **e**, d*N*/d*S* analysis comparing mutations in selected oncogenes when present in chromosomal amplifications, ecDNA and in areas of the genome with no amplification. The error bars represent the 95% confidence intervals calculated using the genesetdnds from the package dNdScv. ADENO, adenocarcinoma; AST, astrocytoma; BFB, break–fusion–bridge; BLCA, bladder cancer; BRCA, breast cancer; CHO, chordoma; CHOL, cholangiocarcinoma; CNS, central nervous system; GBM, glioblastoma; GI, gastrointestinal; HPB, hepatopancreatobiliary cancer; KIRC, clear cell renal cell carcinoma; LIHC, liver hepatocellular carcinoma; LMS, leiomyosarcoma; LPS, liposarcoma; LUAD, lung adenocarcinoma; LUSC, lung squamous cell carcinoma; MELA, malignant melanoma; MFS, myxofibrosarcoma; ODG, oligodendroglioma; OPT, oropharyngeal tumour; OSA, primary conventional osteosarcoma; OVA, ovarian cancer; PAAD, pancreatic adenocarcinoma; PRAD, prostate adenocarcinoma; SCC, squamous cell carcinoma; SCLC, small cell lung cancer; STAD, stomach adenocarcinoma; TN, triple negative; UGI, upper gastrointestinal; UTER, endometrial cancer. The graphics of the Eppendorf tube in **a** and the body map in **c** were created with BioRender.com.

have a transcriptional pattern suggestive of immunosuppression[4,17]. However, the mechanism underlying this suppression is not fully understood. ecDNA bearing immunomodulatory genes has previously been identified in a biopsy from a patient with Barrett's oesophagus with high-grade dysplasia who went on to develop oesophageal adenocarcinoma[10]. We determined the frequency and tissue context of immunomodulatory gene amplification on ecDNA and investigated whether there is enrichment for these genes. A total of 34% of tumours with

ecDNA had known immunomodulatory genes amplified on ecDNA, most of which were co-amplified with oncogenes located nearby (Fig. 2b,c). However, 41.5% of the tumours with immunomodulatory genes amplified on ecDNA lacked oncogenes on those same ecDNAs, suggesting a functional role for these elements (Fig. 2b,c). These immunomodulatory genes were involved in several processes such as the negative regulation of immune effector process (Gene Ontology code GO:0002698, *q* value = $4.5 \times 10^{-10}$), leukocyte-mediated cytotoxicity

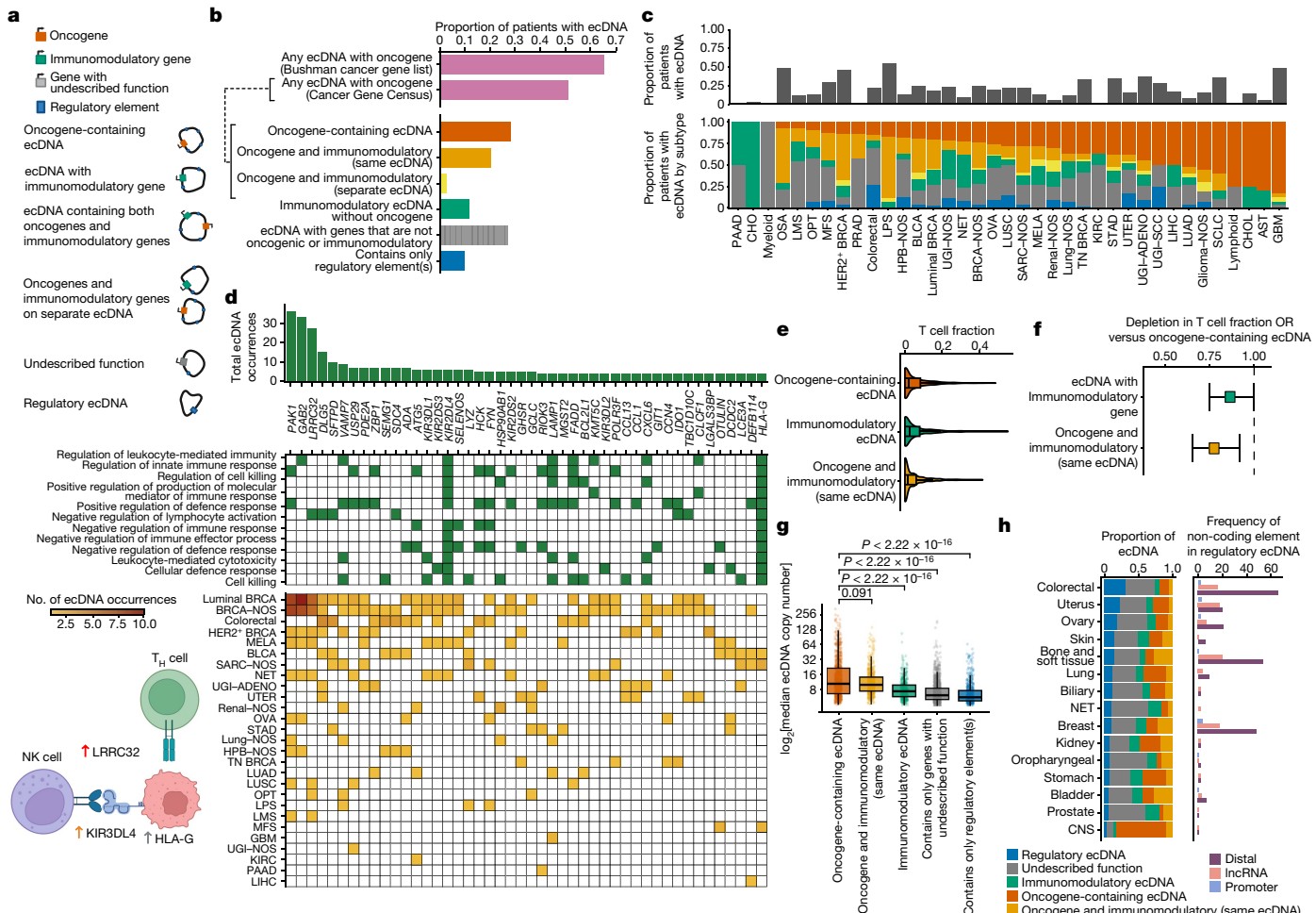

**Fig. 2 | Immunomodulatory and regulatory ecDNA. a**, Schematic showing the subclasses of ecDNA used for analysis. **b**, Bar plots showing the proportion of patients with ecDNA carrying an oncogene defined by either the Cancer Gene Census or the Bushman cancer gene list (top) and showing the proportion of patients with each subclass of ecDNA as demonstrated in the schematic (bottom). The dashed line indicates that all ecDNAs found to carry ≥1 oncogene from the Cancer Gene Census are then further subclassified. **c**, Stacked bar plot showing the proportion of patients of that cancer type demonstrating ecDNA (top) and the proportion of ecDNA in that cancer type in each ecDNA subclass (bottom). **d**, Top: bar plot showing the total number of occurrences (*y* axis) of immunomodulatory genes on ecDNA (*x* axis). Middle: heat map showing GO terms associated with immune genes (green). Bottom: Cancer types in which the immune genes are observed to be on ecDNA; cell colour indicates the number of tumours in which they are observed. Bottom left: schematic showing proposed

mechanism for immunomodulation. $T_H$ cell, T helper cell; NK cell, natural killer cell; $T_H$, T helper. **e**, Violin plot showing the DNA-sequencing-inferred T cell fraction in the presence of ecDNA with oncogenes, ecDNA with immunomodulatory genes, or ecDNA with both oncogenes and immunomodulatory genes. **f**, Forest plot showing the OR of the increase in T cell fraction in the presence of immunomodulatory genes or oncogenic and immunomodulatory genes on ecDNA compared with oncogene-containing ecDNA, controlled for purity. The error bars represent the 95% confidence intervals of the odds ratio. CNS tumours were excluded. **g**, Box plots showing ecDNA copy number by ecDNA subclass. **h**, Bar plots showing the proportion of ecDNA categorized into the different ecDNA subclasses by tissue type (left) and the frequency of non-coding elements on the regulatory subclass of ecDNA (right). The graphics illustrating the ecDNA subclasses in **a** and those in the schematic in **d** were created with BioRender.com.

(GO:0001909, *q* value = 2.2 × 10⁻⁷) and the negative regulation of lymphocyte activation (GO:0051250, *q* value = 1.7 × 10⁻⁴; Fig. 2d, Extended Data Fig. 9a–c and Supplementary Table 6).

To examine any potential impact of immunomodulatory genes amplified on ecDNA, we then compared the estimated T cell fraction[18] of tumours with oncogene-containing ecDNA (Fig. 2e). Controlling for tumour purity, we identified a significant depletion of T cells in samples with ecDNA containing immunomodulatory genes (OR 0.86, 95% CI 0.74–0.99; Fig. 2f) and those with both immunomodulatory genes and oncogenes (OR 0.78, 95% CI 0.66–0.92) compared with tumours with ecDNA containing oncogenes without immunomodulatory genes, potentially contributing to a relatively limited immune response[4,17]. Although the copy number of immunomodulatory genes amplified on ecDNAs that lacked oncogenes did not reach the level of those that contained oncogenes, the copy number was elevated relative to that of other ecDNAs also lacking oncogenes (Wilcoxon *P* < 10⁻¹⁶; Fig. 2g).

## Regulatory ecDNAs

One aspect of ecDNA biology is the ability of ecDNAs with different cargoes to interact in *trans*, to form ecDNA hubs[8]. This is a way to leverage combinatorial interactions, as enhancers on the circular particle interact with promoters on another to drive gene expression[19]. We reasoned that tumour samples may also show evidence for ecDNAs containing only regulatory elements such as promoters, enhancers and lncRNAs (referred to as regulatory ecDNA). Therefore, we annotated the DNA sequences of identified ecDNA with collated lists of enhancers and promoters in human cancers[20] and lncRNAs[21] (Extended Data Fig. 9d). Compared with ecDNA with coding genes, regulatory ecDNA had a significant increase in the number of distal enhancers (27.3 versus 13.7, *P* = 0.00023) and promoters (13.7 versus 3.6, *P* = 0.001) per megabase (Extended Data Fig. 9f). Enhancer-only ecDNAs were frequently co-amplified with ecDNA containing oncogenes on separate

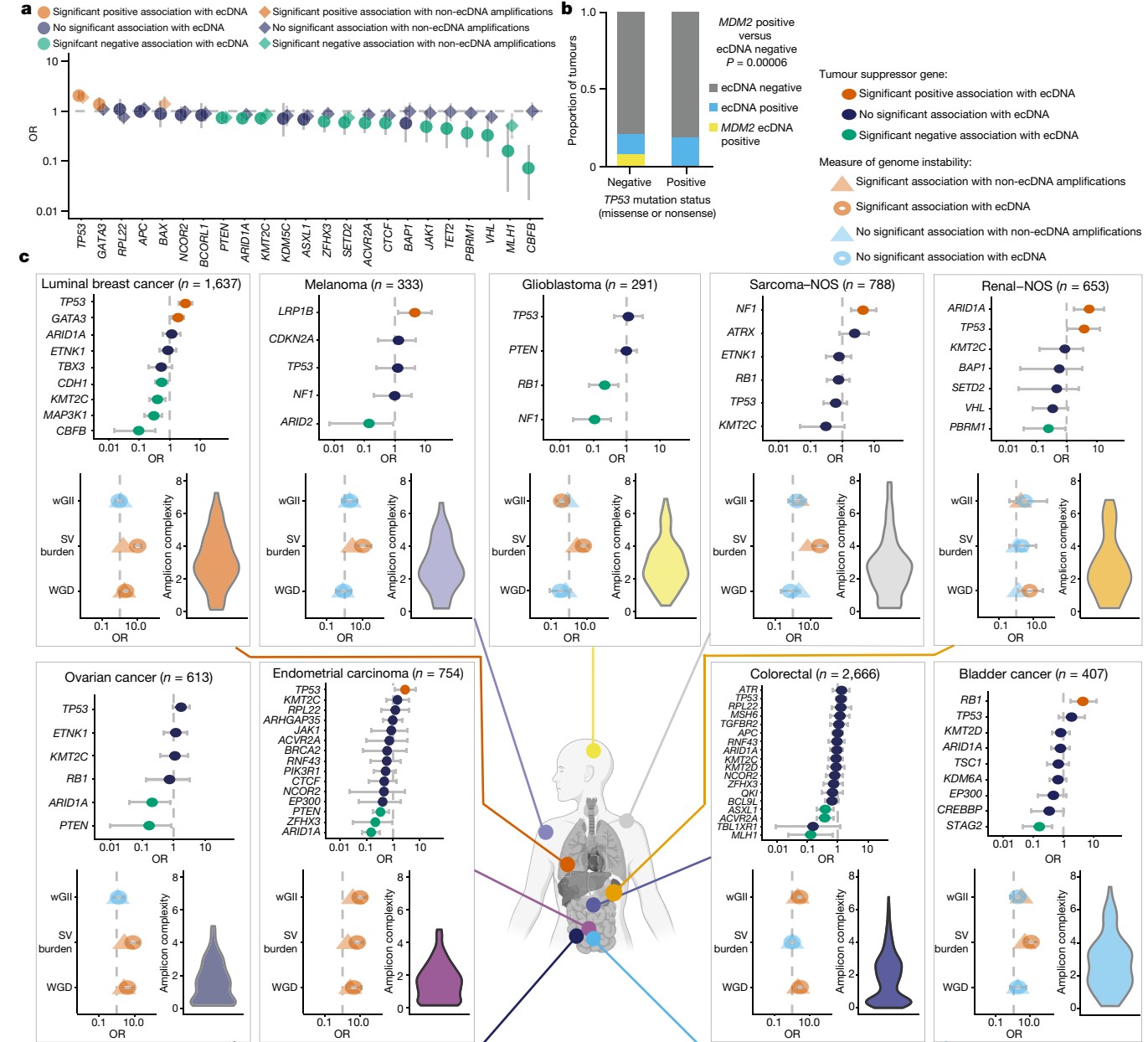

**Fig. 3 | Correlates of genome instability and ecDNA. a**, Forest plot showing the results of a regression model that determines the odds that a tumour will have a high-impact mutation (see Supplementary Information) given the presence or absence of ecDNA or chromosomal amplifications across the entire cohort controlling for cancer type. Associations with ecDNA are represented by circles and those with chromosomal amplifications are represented by diamonds. **b**, Bar plot showing the proportion of tumours across the cohort with any ecDNA (blue), *MDM2* ecDNA (yellow) or no ecDNA (grey) grouped by *TP53* mutation status. **c**, Body map with panels for selected cancer types. Each panel contains a forest plot showing associations between ecDNA presence or absence with high impact tumour suppressor mutations (top); a forest plot showing associations between ecDNA presence or absence with wGII, structural variant burden and whole-genome duplication (WGD; bottom left); and a violin plot demonstrating the amplicon complexity scores for the tumours of that cancer type (bottom right). For **a**,**c**, error bars represent 95% confidence intervals for OR estimates. The graphic of the body map in **c** was created with BioRender.com.

ecDNA ($n = 140$ samples), tended to be smaller (median size 0.12 versus 3.56 Mb, $P < 10^{-16}$; Extended Data Fig. 9g) and had lower copy number (median 6.90 versus 10.05 copies, $P = 0.0078$; Extended Data Fig. 9g).

We next measured the amplicon complexity of enhancer-only ecDNA, which quantifies the number of segments and the diversity of structure decompositions inferred by AmpliconArchitect (Methods and Extended Data Fig. 1b), and found it to be significantly lower than the complexity of ecDNA containing oncogenes (median complexity 3.04 versus 1.10, $P < 10^{-16}$; Extended Data Fig. 9g). Enhancer-only ecDNAs that were co-amplified with ecDNA containing oncogenes had a significantly higher copy number compared with enhancer-only ecDNAs alone

(median, $P = 0.00027$; Extended Data Fig. 9e). These data indicate that regulatory elements are common cargoes in ecDNA, and are amplified through small and simple structures.

## ecDNA and genomic instability

The relationship between ecDNA and specific tumour suppressor mutations, structural and numerical chromosomal instability and whole-genome duplication is largely unexplored across cancer types. Controlling for tumour type, *TP53* mutations were most strongly associated with ecDNA (OR 2.26, 95% CI 1.96–2.62; Fig. 3a). *TP53* mutant

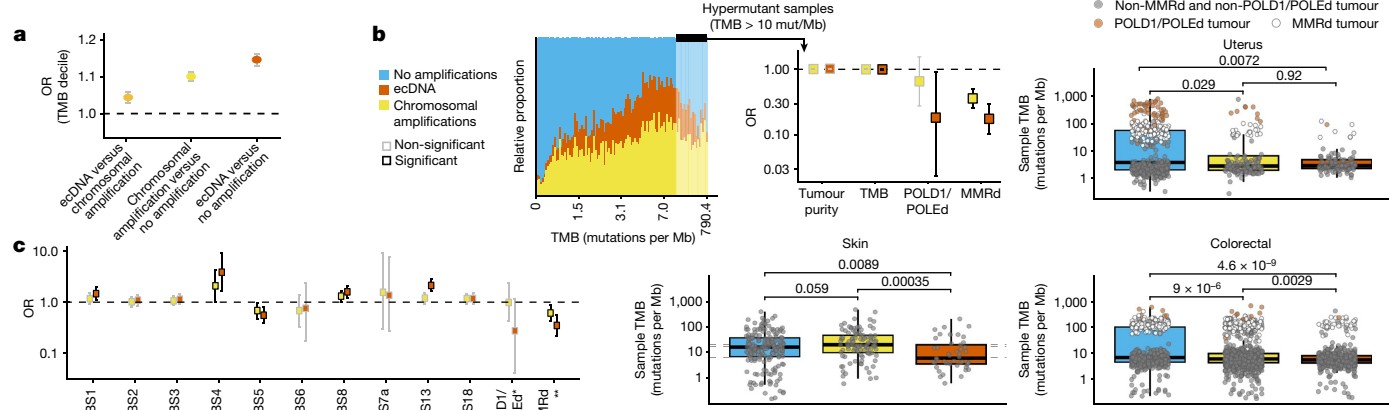

**Fig. 4 | Mutational processes and ecDNA formation. a,** Forest plot depicting the OR of an increased TMB according to the presence of ecDNA or chromosomal amplifications adjusted for purity, age and tumour type. **b,** Top left: the distribution of ecDNA and non-ecDNA focal amplifications in TMB windows. Top middle: forest plot showing the results of a regression model examining the associations of ecDNA and chromosomal amplifications with tumour purity, TMB, and POLD1/POLEd or MMRd status in hypermutant samples. Top right and bottom: box plots of tumour types in which the presence of ecDNA and TMB are negatively correlated. **c,** Forest plot showing the results of a regression model measuring the association of global SBS signatures and the presence of ecDNA in the whole cohort. For **a,b,c,** error bars represent 95% confidence intervals for OR estimates.

tumours (nonsense and missense mutations) and *MDM2* encoded within ecDNAs were mutually exclusive ($\chi^2$ $P = 0.00006$; Fig. 3b). We also detected tissue-type-specific high-impact tumour suppressor pathway mutations and their association with ecDNA. ecDNA was strongly associated with *TP53* mutations in endometrial, renal (not otherwise specified (NOS)) and luminal oestrogen-receptor-positive breast cancer (Fig. 3c and Supplementary Table 7). ecDNA presence was also associated with *NF1* mutations in sarcoma (NOS), *ARID1A* mutations in renal (NOS) and *RB1* mutations in bladder cancer (Fig. 3c). *TP53* was most commonly under selection (using d$N$/d$S$) across cancer types associated with ecDNA (Extended Data Fig. 10a,b and Supplementary Table 8.

To understand whether correlates of genome instability were related to ecDNA presence, we determined whole-genome duplication (defined as sample ploidy of >2.7) and structural variant burden (detected structural variants per megabase) and the weighted genome instability index (wGII; defined as the percentage of gained and lost genetic material relative to the ploidy of the sample; Methods)[22] of examined tumour samples. We concurrently calculated an amplicon complexity score for each ecDNA. Low-complexity amplicons may be associated with the episomal formation of ecDNA, whereas high-complexity amplicons are probably associated with catastrophic events such as chromothripsis[23]. Whole-genome duplication, wGII and structural variant burden were positively associated with the presence of ecDNA in the cohort, dependent on tumour type. Sarcomas were enriched for high-complexity amplicons and were associated with increased structural variant burden (OR 7.98, 95% CI 4.44–14.7), whereas colorectal tumours were enriched for low-complexity ecDNA and were associated with high ploidy (OR 2.65, 95% CI 1.82–3.88) and high wGII, but not structural variant burden, suggesting that distinct processes may lead to or are associated with different ecDNA species (Fig. 3c and Supplementary Table 9).

## ecDNA and mutational processes

The scale of the GEL dataset permitted detection of rare mutational signatures[24]. Therefore, we set out to identify the broad spectrum of mutational processes that can be preferentially found in tumours harbouring ecDNA, using data from ref. 24.

Samples with ecDNA exhibited a higher tumour mutational burden (TMB) compared to those with chromosomal amplifications (OR 1.04, 95% CI 1.02–1.06) and those lacking focal amplifications (OR 1.14, 95% CI 1.13–1.16) when controlled for age, tumour type and purity (Fig. 4a

and Extended Data Fig. 10c); however, this association was limited to non-hypermutator phenotypes (Fig. 4b). Controlling for tumour type, genome-wide single-base substitution (SBS) signatures (which utilize the mutated base and the bases immediately 5′ and 3′ to infer mutational processes; Methods), including SBS1 (deamination), SBS4 (tobacco smoking), SBS8 (unknown) and SBS13 (APOBEC cytidine deaminase) signatures, were more strongly associated with the presence of ecDNA than with other focal amplifications (in keeping with previous observations[25,26]), whereas signatures of mismatch repair deficiency (MMRd; SBS6, SBS15, SBS26 and SBS44) and DNA polymerase δ 1 or DNA polymerase ε deficiency (POLD1/POLEd) with concurrent MMRd (SBS10a, SBS10b, SBS14 and SBS20) were more negatively correlated with ecDNA formation than chromosomal amplifications (Fig. 4b). We also identified an inverse correlation between ecDNA and hypermutation, driven by the association of POLE/POLD1-deficient tumours with the absence of ecDNA detection in colorectal and endometrial cancers (Fig. 4b).

To resolve the mutational processes that may be operating on the ecDNAs themselves, we compared the mutations found on ecDNA with all other remaining mutations in the sample using a maximum-likelihood estimate approach to attribute mutations to specific mutational processes (Methods, Extended Data Fig. 10d and Supplementary Table 10). The ecDNAs were significantly enriched for SBS3 (homologous recombination deficiency) and negatively associated with SBS1 (clock-like deamination), SBS5 (clock-like signature), SBS8 (unknown) and SBS17 (unknown). ecDNAs were also found to have more mutations related to APOBEC (SBS2 and SBS13) relative to the rest of the genome (Extended Data Fig. 10d).

The mutational signatures linked to ecDNA could reflect processes involved in their formation or their ongoing evolution. By mapping mutations on predicted ecDNA, we were able to infer the timing of mutational processes relative to when the ecDNA formed[27]. Mutations that were predicted to be present on all ecDNAs must have occurred before ecDNA formation, whereas mutations that were not present on all ecDNA copies probably occurred following ecDNA formation (Extended Data Fig. 10e). Controlling for tumour type, we found that mutational processes that predominantly acted before ecDNA formation were those associated with tobacco exposure (SBS4 and SBS92), ultraviolet light (SBS7a) and clock-like deamination (SBS1). By contrast, the signature for homologous recombination repair deficiency, SBS3, tended to occur after ecDNA formation (Extended Data Fig. 10f).

In addition, treatments could affect the development of ecDNA. In a patient with glioblastoma treated with temozolomide (TMZ),

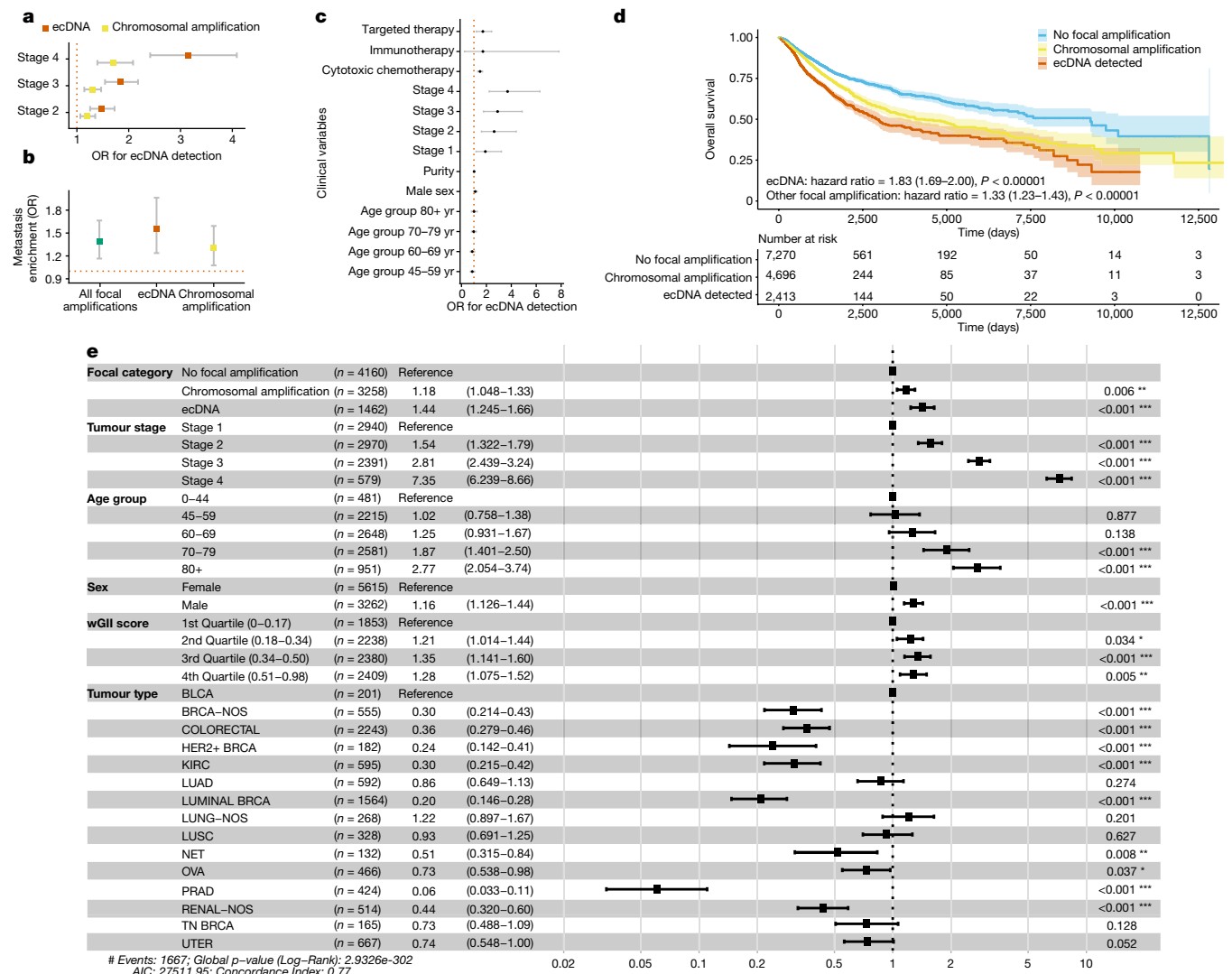

**Fig. 5 | ecDNA and clinical outcome. a**, Forest plot showing the results of a regression model examining ecDNA presence in the context of disease stage. **b**, Forest plot showing the results of a regression model investigating all focal amplifications, ecDNA and non-ecDNA amplifications in the context of metastatic samples. **c**, Forest plot of the results of a regression analysis investigating the association of ecDNA with clinical variables adjusted for cancer type, age, sex and purity. For **a**,**b**,**c**, error bars represent 95% confidence intervals for OR estimates. **d**, Kaplan–Meier plot showing overall survival in the GEL cohort (survival data available for 14,773 patients). The error bars represent 95% confidence intervals for OR estimates. **e**, Forest plot showing the results of a fully adjusted Cox proportional hazards model adjusting for tumour stage, age, sex, wGII and tumour type. **, $P < 0.05$; ***, $P < 0.001$.

we detected an *EGFR* c.3106A>T mutation on each of the 67 copies of ecDNA. We further detected an SBS11 TMZ-induced MMRd hypermutator signature on these *EGFR*-mutant ecDNAs, leading to many distinct mutations on these ecDNAs that seem to be linked to the TMZ treatment, including some which achieved very high allelic frequencies suggestive of selection. Taken together, these data suggest that the TMZ treatment can influence the evolution of ecDNAs, through both mutagenesis and subsequent selection (Extended Data Fig. 10g).

## ecDNA and prognostic relevance

Adjusting for age, sex and tumour type, we found that ecDNA was strongly associated with increasing tumour stage (stage 2 versus stage 1: OR 1.46, 95% CI 1.24–1.68; stage 3 versus stage 1: OR 1.79, 95% CI 1.49–2.08; stage 4 versus stage 1: OR 2.18, 95% CI 1.81–2.54; Fig. 5a), suggesting a stage-dependent association of ecDNA. Further, controlling for tumour type, we found that ecDNA was significantly enriched in metastatic samples (non-paired, OR 1.56, 95% CI 1.24–1.96; Fig. 5b and

Extended Data Fig. 10h), suggesting that ecDNA may play a role in cancer progression and the development of metastasis.

We then explored the association between ecDNA and treatment, revealing in a logistic regression model, adjusted for age, stage, purity and cancer type, that ecDNA detection was significantly associated with prior chemotherapy (OR 2.38, 95% CI 1.73–3.27) and targeted treatment (OR 2.87, 95% CI 1.12–6.43; Fig. 5c). In an adjusted Cox proportional hazards model controlled for tumour type, stage, age, sex and underlying genome instability (wGII), the detection of ecDNA was associated with shorter overall survival (hazard ratio 1.44, 95% CI 1.25–1.66), relative to intra-chromosomal amplifications with no evidence of ecDNA (hazard ratio 1.18, 95% CI 1.05–1.33) or tumours with no focal amplifications (Fig. 5d,e).

## Discussion

ecDNA presents a complex challenge. Its non-chromosomal inheritance drives intratumoural genetic heterogeneity fuelling accelerated evolution, thereby enabling tumours to resist treatment. The highly accessible chromatin of ecDNA alters gene regulatory architecture and fosters

combinatorial interactions between ecDNA particles[2,8,9]. By analysing the largest available single collection of whole-genome-sequenced samples from patients with cancer currently available, we demonstrate the remarkable diversity of ecDNA elements across cancer, illuminating the associated tissue and genetic contexts and the mutational processes to which ecDNA is linked. These results shed light on how ecDNA cooperatively drives tumour growth signals through high-copy-number oncogene amplification, the possible alteration of transcriptional landscapes through regulatory element-only ecDNAs, and how it may regulate the immune system through amplification of immunomodulatory genes.

The detection of intrinsic and environmental mutational processes that tend to occur before or after the emergence of ecDNA, including tobacco exposure early on in tumour evolution and homologous recombination repair deficiency once ecDNA has formed, provides new insight into factors that may contribute to ecDNA formation and progression. This is particularly important given the recent finding that ecDNA may arise in high-grade dysplasia and contribute to tumorigenesis[10]. Further, the finding that ecDNA levels may rise after cytotoxic and targeted therapy treatments also suggests a potential role for combinations of ecDNA-directed and conventional or precision oncology treatments.

Our data also reveal some unanticipated results, such as the high level of ecDNA found in HER2[+] breast cancer (39.3–53.7%), including amplification of *ERBB2* on ecDNA in 26% of HER2[+] breast cancers (Supplementary Table 2). Given the known role of ecDNA in driving intercellular heterogeneity, the demonstration that increased HER2[+] copy-number heterogeneity is associated with shorter disease-free survival is notable[28]. Understanding to what extent HER2[+] heterogeneity is ecDNA-driven will be of critical importance, as will the elucidation of the full complement of ecDNA cargo and its function in HER2[+] cancers.

Bioinformatic detection of ecDNA from WGS data has inherent limitations. Some of the limitations of ecDNA detection are tumour-specific, such as the effects of tumour purity and ecDNA copy number. Other limitations are more technical, including the detection of structural variants in repetitive regions of the genome, the effects of sequencing coverage, and algorithmic challenges in distinguishing types of focal amplification. Although the ecDNA detection methods used here have been shown to be robust, improvements in sequencing technologies and methods for ecDNA detection should provide even more refined estimates of ecDNA frequency across cancers.

Finally, we note that ecDNA is associated with poor survival even when accounting for underlying genome instability, suggesting that there are ecDNA-specific effects that contribute to poor outcomes in patients. As the contribution of ecDNA to intratumour heterogeneity, drug resistance and poor survival through the rapid generation and fine-tuning of gene dosage is becoming more apparent[29], the important and ongoing challenge to determine therapeutic vulnerabilities and identify target compounds to limit ecDNA evolution and maintenance will be applicable across the pan-cancer spectrum.

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

**Genomics England Consortium**

J. C. Ambrose[18], P. Arumugam[18], R. Bevers[18], M. Bleda[18], F. Boardman-Pretty[18], C. R. Boustred[18], H. Brittain[18], M. A. Brown[18], M. J. Caulfield[18], G. C. Chan[18], A. Giess[18], J. N. Griffin[18], A. Hamblin[18], S. Henderson[18], T. J. P. Hubbard[18], R. Jackson[18], L. J. Jones[18], D. Kasperaviciute[18], M. Kayikci[18], A. Kousathanas[18], L. Lahnstein[18], A. Lakey[18], S. E. A. Leigh[18], I. U. S. Leong[18], F. J. Lopez[18], F. Maleady-Crowe[18], M. McEntagart[18], F. Minneci[18], J. Mitchell[18], L. Moutsianas[18,19], M. Mueller[18,19], N. Murugaesu[18], A. C. Need[18,19], P. O'Donovan[18], C. A. Odhams[18], C. Patch[18,19], D. Perez-Gil[18], M. B. Pereira[18], J. Pullinger[18], T. Rahim[18], A. Rendon[18], T. Rogers[18], K. Savage[18], K. Sawant[18], R. H. Scott[18], A. Siddiq[18], A. Sieghart[18], S. C. Smith[18], A. Sosinsky[18,19], A. Stuckey[18], M. Tanguy[18], A. L. Taylor Tavares[18], E. R. A. Thomas[18,19], S. R. Thompson[18], A. Tucci[18,19], M. J. Welland[18], E. Williams[18], K. Witkowska[18,19], S. M. Wood[18,19] & M. Zarowiecki[18]

[18]Genomics England, London, UK. [19]William Harvey Research Institute, Queen Mary University of London, London, UK.

## Methods

### Dataset

GEL is a company funded by the Department of Health and Social Care in the UK. Part of the flagship project, 100kGP, was set up to sequence 100,000 whole genomes from National Health Service (NHS) patients with rare diseases and cancer. In this study, we utilized version 12 of the cohort, comprising 14,778 participants. Sequencing libraries generated from tumour and matched germline DNA samples were sequenced using 150-base-pair paired-end reads on Illumina HiSeq platforms. In total, 16,355 tumour and 16,555 germline samples underwent whole-genome sequencing at a target depth of 100× for tumour and 30× for germline.

We included the cancer types from the following tissues (n = 17): breast, lung, stomach, neuroendocrine, skin, oropharyngeal, colorectal, kidney, prostate, hepato-pancreatobiliary, bladder, bone and soft tissue, ovary, endometrium, central nervous system, lymphoid and myeloid. The following tumour subtypes were then included (n = 39): bladder; chordoma; primary conventional osteosarcoma; liposarcoma (both dedifferentiated and myxoid); leiomyosarcoma; myxofibrosarcoma; sarcoma, not otherwise specified; HER2+ breast cancer; luminal (oestrogen receptor positive) breast cancer; triple-negative breast cancer; breast cancer, not otherwise specified; oligodendroglioma; astrocytoma; glioblastoma; adult glioma, not otherwise specified; colorectal cancer; hepato-pancreatobiliary cancer, not otherwise specified; liver hepatocellular carcinoma; cholangiocarcinoma; pancreatic adenocarcinoma; malignant pleural mesothelioma; small cell lung cancer; lung squamous cell carcinoma; lung adenocarcinoma; lung cancer, not otherwise specified; lymphoid; myeloid; neuroendocrine tumour; oropharyngeal cancer; ovarian cancer; clear cell renal carcinoma; malignant melanoma; renal cancer, not otherwise specified; upper GI squamous cell carcinoma; stomach adenocarcinoma; upper GI adenocarcinoma; upper GI cancer, not otherwise specified; and endometrial carcinoma.

Most tumour samples in the GEL cohort come from patients whose cancers were early disease stage and had not yet received treatment (Fig. 1c). Samples with tumour purity of <10% were excluded, as were cancers of unknown primary, paediatric cancers and testicular germ cell tumours (510 samples). In the cohort, 3.8% (598) of samples were fixed-formalin paraffin embedded (FFPE)[30]. Staging information was available for 10,780 (72.9%) patients, with 836 (5.7%) patients recorded as having stage 4 disease (Fig. 1c). Median depth of coverage for tumour samples was 97.6× and for germline samples was 32.6×. A total of 1,800 (12.1%) patients were recorded as receiving systemic anticancer treatment before biopsy. In this group, the treatment type was classified into hormonal (n = 27), immunotherapy (n = 57), targeted (n = 415) or cytotoxic chemotherapy (n = 1,653).

### Inclusion and ethics

The research presented in this manuscript is compliant with ethical regulations and was approved by the East of England−Cambridge South Research Ethics Committee (Research Ethics Committee reference 14/EE/1112, Integrated Research Application System ID 166046). Recruitment of participants was carried out across 13 NHS Genomic Medicine Centres and all participants provided their written and informed consent.

### ecDNA calls

Focal DNA copy-number alterations were identified using CNVKit v0.98. AmpliconArchitect v1.2 was used to construct cyclic paths from identified focal amplifications, and AmpliconClassifier v0.4.12 was used to determine whether these paths were likely to be ecDNA. These steps are packaged into a single workflow available at https://github.com/AmpliconSuite/AmpliconSuite-pipeline.

AmpliconArchitect identifies the structure of focal amplifications by using seed intervals that define regions that are focally amplified and extend beyond them to look for copy-number changes or discordant edges. For this analysis, seed intervals were defined as regions of greater than 50 kb, with a threshold copy number of greater than 4.5, double the ploidy of the tumour and at least 2.5 additional copies above the median arm-level copy number. The regions are then merged to form a breakpoint graph, which can be broken down into simple and complex cycles to identify any circular paths that could be indicative of ecDNA. Within the seed interval, it is possible that an ecDNA reconstruction could be less than 50 kb. AmpliconArchitect also masks highly repetitive regions such as α-satellites in centromeric and peri-centromeric regions.

We conducted additional FISH on 11 tissue samples that we were able to obtain from GEL, demonstrating 90.9% (10/11) accuracy of our computational calls, comparable to previous validation of these methods[4].

Patients were categorized as having ecDNA if ecDNA was detected in their tumour. Patients who had both chromosomal amplifications and ecDNA were included in the ecDNA category.

We then annotated each ecDNA according to whether or not it contained an oncogene as categorized by the Cancer Gene Census (https://cancer.sanger.ac.uk/census). Those that contained an oncogene were denoted as oncogenic ecDNA. We then further divided ecDNA according to whether or not any genes were annotated, classifying it as 'ecDNA without known oncogenes' and 'ecDNA without coding genes'. We then carried out an over-representation analysis for genes encoded on ecDNA without known oncogenes to demonstrate an enrichment for immunomodulatory genes.

### Amplicon complexity

The amplicon complexity score, as defined in ref. 10, is calculated by AmpliconClassifier (Extended Data Fig. 1b). For each seed-interval-defined amplicon, AmpliconArchitect produces a copy-number-aware (CNA)-breakpoint graph. AmpliconArchitect also outputs decompositions that represent cyclic and non-cyclic paths through this CNA-breakpoint graph. These decompositions are passed as input to AmpliconClassifier to produce the complexity score, which aims to capture the diversity of the cyclic and/or non-cyclic paths present. Each path has a copy count and a length in kilobase pairs, which are combined to create a length-weighted copy number (normalized by the total length-weighted copy number present in the CNA-breakpoint graph), and the complexity score is calculated through the sum of three log-transformed measures: the total count of copy-number segments present in the amplicon; the normalized length-weighted copy number of each cyclic path; and the residual normalized length-weighted copy number that is not explained by cyclic paths.

### Over-representation analysis

Over-representation analysis was carried out using the cluster profiler package (v4.6.2)[31]. To determine whether the genes that were annotated on 'ecDNA without known oncogenes' were in predefined sets that were present at frequencies higher than expected by chance, the annotated genes were assigned to a specific gene set (denoted by the GO term). Following this, the observed proportion of genes assigned to that GO term was compared with the expected proportion given the background of all genes using Fisher's exact test.

A false discovery rate-adjusted P value was obtained using the Benjamini−Hochberg method, with the threshold for significance set at q > 0.001. The minimum gene set size considered was 100. ecDNAs were considered immunomodulatory if a gene from the significant gene sets mapped to an ecDNA that did not contain an oncogene, and that significant gene set had an immunomodulatory function (GO terms: 0006968, 0002228, 0042267, 0001906, 0001909, 0002698, 0001910, 0031341, 0002367, 0002695, 0050866, 0051250, 0050777).

### Permutation test for oncogene enrichment

For permutation testing, first, the proportion of focal amplifications that contained oncogenes was calculated. A total of 256 genes were

identified as oncogenes. From the pool of genes identified on ecDNA at least once ($n = 20,713$), a random set of 256 genes were sampled and the proportion was calculated. This calculation was repeated 10,000 times to obtain a background distribution of the proportion of genes that belong to a gene set of equal size.

## Estimates of selection using d*N*/d*S*

The d*N*/d*S* estimates were calculated using the dNdScv package[16], which was run on all mutations available in the cohort. This method uses a maximum-likelihood approach in its analysis of the ratio of non-synonymous (missense, nonsense and essential splice mutations) to synonymous substitutions to infer a measure of the strength of selection acting on protein-coding genes. It also estimates a background mutation rate of each gene through joint analysis of both local and global information that takes into account sequence composition and the contribution of mutational signatures. In our analysis, estimation of d*N*/d*S* ratios was carried out across the genome, as well as stratified by context including ecDNA amplification, non-ecDNA amplification and non-amplified areas of the genome (Extended Data Fig. 10a). Six genes were amplified at >5% of the cohort with a strong signal of positive selection (d*N*/d*S* estimate > 5); *YEATS4*, *CCT2*, *FPS6* (sarcoma), *KRAS*, *ERBB2* (upper gastrointestinal) and *EGFR* (central nervous system; Extended Data Fig. 7f).

## Somatic mutation calling/ploidy and purity estimation

Strelka2 (version 2.4.7)[32] and Manta (version 0.28.0)[33] were used to call mutations and SVs, respectively. Manta combines paired and split-read evidence for SV discovery and scoring. The following filters were applied to the raw variant calls: SVs with a normal sample depth near one or both variant break-ends three times higher than the chromosomal mean; SVs with a somatic quality score of <30; somatic deletions and duplications with a length of >10 kb; somatic small variants (<1 kb) with the fraction of reads with MAPQ0 around either break-end of >0.4. For purity estimates, CCube was used[34]. For ploidy estimates, the CakeTin pipeline from ref. 35 was utilized, available for 9,141 samples.

## Calculation of wGII

The wGII score was calculated as the proportion of the genome with aberrant copy number relative to the median ploidy, weighted on a per chromosome basis[22]. Both median ploidy and copy-number segments were rounded to the nearest integer copy state from CNVKit.

## SBS signature analysis

Ref. 24 quantified the fraction of SBSs found in each of the 96 trinucleotide contexts from the multiple WGS cohorts (including GEL) and analysed these data with non-negative matrix factorization to infer a set of SBS signatures[24]. We then used this reference set of SBS signatures to infer the most likely SBS signature for mutations in our cohort. Using the sample-level SBS exposures and the SBS reference signatures, each trinucleotide channel context is assigned a likelihood value by multiplying the sample exposure weight by the reference signature weight. This allows estimation of the most likely mutational process for each mutation.

## Timing of SBS signatures

To perform this analysis we used the SBS signatures as defined in ref. 24. By analysing the variant allele frequency distribution of single nucleotide variants (SNVs) at focal amplification sites, it becomes possible to temporally assess the formation of ecDNA and the mutational processes occurring in that genomic region. This assessment involves calculating the mutational multiplicity, which is determined by the copy-number state of an SNV within a predicted ecDNA locus. SNVs are classified as occurring either pre- or post-ecDNA formation on the basis of whether the SNV copies are equivalent to the total copy number at the locus. The mutational multiplicity can be determined by the following formula:

$$\text{CPNmut} = \text{VAF} \times (1/p) \times (p \times \text{CPNfocal}) + \text{CPNnorm} \times (1 - p)$$

in which VAF represents the variant allele frequency, $p$ represents tumour purity, CPNfocal represents the focal amplification copy number, and CPNnorm represents the local copy number in the normal genome. Mutations are considered pre-ecDNA formation if CPNmut is greater than 0.8 times CPNfocal. Mutations are classified as post-ecDNA formation if CPNmut is less than 0.8 times CPNfocal and greater than CPNnorm/2.

By aggregating mutations across multiple samples within the same tumour, a maximum-likelihood function is used to determine whether ecDNA tends to occur before or after a mutation process. This involves creating a mutational catalogue that categorizes all mutations on the basis of 96 trinucleotide context channels and their pre- or post-ecDNA formation status. Using the sample-level SBS exposures and the SBS reference signatures, each trinucleotide channel context is assigned a likelihood value by multiplying the sample exposure weight by the reference signature weight. This allows estimation of the most likely mutational process for each mutation and the identification of processes acting early or late in the context of ecDNA formation. We then carried out a Wilcoxon test, comparing the probabilities that a mutation within the ecDNA locus is early with the probability that it is late on each sample, and presented the median difference between the two categories.

## Statistical analysis

All statistical tests were carried out in R (4.1.2). Correlation tests were carried out using cor.test with either Spearman's method or Pearson's method, as specified. Tests comparing distributions were carried out using wilcox.test or t.test.

Proportions were compared using prop.test. For prevalence estimates, the 95% CI of a proportion was reported using propCI. Logistic regression models were fitted using glm(outcome ~ exposure_variables, family = 'logit'), with ORs and 95% CIs reported. For the regression analysis in which we controlled for tumour type, we excluded those tumour types with fewer than five patients sampled.

## GEL sample FISH

FISH was carried out on 4 μM FFPE tissue sections according to a combination of the Agilent Technologies Protocol (Histology FISH Accessory Kit K5799) and the Abbott Molecular Diagnostic FISH probe protocol. Briefly, FFPE sections were dewaxed in xylene for 5 min followed by rehydration in 100%, 80% and 70% ethanol and then washed twice with Agilent Technologies wash buffer. The FFPE tissue was then incubated at 98 °C for 10 min in Agilent Technologies pretreatment solution. The Coplin jar containing the slides was removed from the 98 °C water bath and allowed to slowly cool for an additional 15 min. The FFPE slides were washed twice with Agilent wash buffer. Stock pepsin (Agilent stock pepsin) was applied to the slide for 10 min at 37 °C. FFPE slides were washed twice with Agilent wash buffer and then dehydrated using 70%, 80% and 100% ethanol before probe hybridization. Gene-specific probes containing chromosome-specific centromere enumeration probes (CEP) to *MDM2* (+control CEP12 spectrum green) (Vysis/Abbott), *MDM2* and *CDK4* (Empire Genomics), *PDGFRA* (+control CEP4 spectrum green) (Empire Genomics) and *MYC* (Vysis/Abbott) were applied directly to the tissue sections with the coverslip being sealed with rubber solution glue.

Denaturation of the probes on the tissue was carried out at 75 °C for 7 min. The slide was then incubated overnight in a moist box at 37 °C for 16 h. Slides were washed for 10 min at 73 °C with 0.4× SSC containing 0.3% Igepal (Sigma) followed by a 10-min wash at room temperature with 2× saline-sodium citrate (SSC) containing 0.1% Igepal. Slides were allowed to air dry and then counterstained with a Vectashield mounting medium containing 4′,6-diamidino-2-phenylindole (DAPI; ThermoFisher). Images were captured using the Applied Precision DeltaVision Microscope.

## HER2 FISH

FFPE tissue sections were deparaffinized by two 5-min incubations in Histo-Clear (Electron Microscopy Sciences 64110), followed by 5 min in 100% ethanol and 5 min in 70% ethanol. Samples were then incubated in 0.2 N HCl for 20 min. Slides were then placed in 10 mM antigen retrieval buffer (10 mM citric acid pH 6.0) and incubated in a vegetable steamer (90–95 °C) for 20 min. Slides were briefly washed in 2× SSC and then treated with proteinase K digestion buffer (1:100 dilution of proteinase K NEB P8107 in TE buffer, 100–200 µl per sample) for 1 min at room temperature. Slides were then dehydrated by incubation for 2 min each in 70%, 85% and 100% ethanol. The HER2 FISH and Chr. 17 control centromere enumeration FISH probes (Empire Genomics ERBB2-CHR17-20-ORGR) were diluted 1:5 in hybridization buffer (Empire Genomics), added to each slide, and covered with a coverslip. Slides were denatured at 75 °C for 5 min followed by overnight hybridization at 37 °C in a humidified chamber. Slides were washed twice in 0.4× SSC + 0.3% Igepal630 (5 min, 40–60 °C) and then in 2× SCC + 0.1% Igepal630 (5 min, room temperature). Slides were treated with a TrueVIEW Autofluorescence Quenching kit (Vector laboratories SP-8400) according to the manufacturer's directions for 2 min and then washed in 2× SSC (5 min, room temperature). Slides were mounted with ProLong Gold antifade with DAPI (ThermoFisher P36931). Slides were imaged on a Zeiss LSM880 confocal microscope using a 0.45-µm *Z*-step size. Maximum-intensity projections were generated using ZEN2.3 SP1 FP3 software. This component of the study was approved by the Stanford University Institutional Review Board (number 69198).

## Reporting summary

Further information on research design is available in the Nature Portfolio Reporting Summary linked to this article.

## Data availability

Aggregated information used for analysis is available in Supplementary Tables 1–10. Requests for raw sequencing data, variant calls, survival data, quality metrics and a summary of findings submitted to genomics laboratory hubs can be made through the GEL Research Environment, a secure cloud workspace. To access the genomic and clinical data in this Research Environment, researchers must first apply to become a member of either the GEL Research Network (previously known as the GEL Clinical Interpretation Partnership; https://www.genomicsengland.co.uk/research/academic) or a Discovery Forum industry partner (https://www.genomicsengland.co.uk/research/research-environment). First, a signed participation agreement must be submitted by the institution to gecip-help@genomicsengland.co.uk. Then, following selection of an appropriate research domain, an online application should be submitted. Applications will be reviewed within ten working days, following which institutions must validate the researcher's affiliation. On approval, access to the GEL Research Environment will be granted following successful completion of an online information governance training module. Further details of the types of data available (hospital episodes, survival and treatment data) can be found at https://re-docs.genomicsengland.co.uk/data_overview/. The cohort of patients with cancer and longitudinal clinical information on treatment and mortality can be explored with Participant Explorer (https://re-docs.genomicsengland.co.uk/pxa/).

## Code availability

The code used to run the SBS likelihood assignment, permutations, mutational timing and figures is available in the GEL Research Environment (https://re-docs.genomicsengland.co.uk/access/) under /re_gecip/shared_allGeCIPs/pancancer_ecdna/code/. The link to becoming a member of the GEL Research Network and obtaining access can be found at https://www.genomicsengland.co.uk/research/academic/join-gecip.

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

**Acknowledgements** This research was made possible through access to data in the National Genomic Research Library, which is managed by Genomics England Limited (a wholly owned company of the Department of Health and Social Care). The National Genomic Research Library holds data provided by patients and collected by the NHS as part of their care and data collected as part of their participation in research. The National Genomic Research Library is funded by the National Institute for Health Research and NHS England. The Wellcome Trust, Cancer Research UK and the Medical Research Council have also funded research infrastructure; A. Frankell, E. Gronroos and I. Noorani for their thoughtful review of the manuscript; and staff of the Stanford University Cell Sciences Imaging Facility (RRID:SCR_017787) for assistance with interphase HER2 FISH. Graphics in Figs. 1a,c, 2a,d and 3c and Extended Data Figs. 5 and 6 were created with BioRender.com. C.S. is a Royal Society Napier Research Professor (RSRP\R\210001). The work of C.S. is supported by the Francis Crick Institute, which receives its core funding from Cancer Research UK (CRUK; CC2041), the UK Medical Research Council (CC2041) and the Wellcome Trust (CC2041). For the purpose of open access, the authors have applied a CC BY public copyright licence to any author accepted manuscript version arising from this submission. C.S. is supported by CRUK (TRACERx (C11496/A17786), PEACE (C416/A21999) and CRUK Cancer Immunotherapy Catalyst Network); CRUK Lung Cancer Centre of Excellence (C11496/A30025); the Rosetrees Trust; the Butterfield Trust; the Stoneygate Trust; the NovoNordisk Foundation (ID16584); the Royal Society Professorship Enhancement Award (RP/EA/180007); the National Institute for Health Research University College London Hospitals Biomedical Research Centre; the CRUK–University College London Centre; the Experimental Cancer Medicine Centre; the Breast Cancer Research Foundation (US) (BCRF-22-157); the CRUK Early Detection and Diagnosis Primer Award (grant EDDPMA-Nov21/100034); and The Mark Foundation for Cancer Research Aspire Award (grant 21-029-ASP). This work was supported by a Stand Up To Cancer–LUNGevity–American Lung Association Lung Cancer Interception Dream Team Translational Research Grant (grant number SU2C-AACR-DT23-17 to S. M. Dubinett and A. E. Spira). Stand Up To Cancer is a division of the Entertainment Industry Foundation. Research grants are administered by the American Association for Cancer Research, the Scientific Partner of SU2C. C.S. is in receipt of a European Research Council Advanced Grant (PROTEUS) from the European Research Council under the European Union's Horizon 2020 research and innovation programme (grant agreement number 835297). This work was also delivered as part of the eDyNAmiC team supported by the Cancer Grand Challenges partnership financed by CRUK (P.S.M. and H.Y.C., CGCATF-2021/100012; V.B. and J.L., CGCATF-2021/100025) and the National Cancer Institute (P.S.M. and H.Y.C., OT2CA278688; V.B. and J.L., OT2CA278635). P.S.M. is the eDyNAmiC team lead and S.N.-Z., M.J.-H., H.Y.C. and V.B. are members of the eDyNAmiC team. V.B. was additionally supported in part by the grants U24CA264379 and R01GM114362 from the National Institutes of Health. A.M.F. is supported by the Bone Cancer Research Trust. This study was also supported by a grant from the National Brain Tumour Society (P.S.M.) and the National Institutes of Health (R01-CA238379; P.S.M.).

**Author contributions** C.B. coordinated and carried out computational analyses, processed the data, carried out data quality control and bioinformatics pipeline development, designed and conducted bioinformatics analyses and wrote the manuscript. O.P. designed and conducted bioinformatics analyses, designed and conducted bioinformatics pipeline development and wrote the manuscript. K.T. designed and conducted bioinformatics analysis and assisted with pathology and clinical annotation. T.B.K.W. designed bioinformatics analyses and wrote the manuscript. J.L. designed and carried out bioinformatics pipeline development and conducted bioinformatic analyses. A.R. led and carried out the FISH work. G.S., B.D., R.B., J.K., N.S., M.L. and A.J.C. assisted with data processing, data production and bioinformatic analyses. N.E.W., S.Y.C.Y. and B.E.H. carried out sample selection and FISH. W.-T.L. and K.L.H. assisted with data analysis. R.S. and D.A.M. carried out clinical annotation and pathology review. R.S.H. supported additional data processing, generation and bioinformatic analysis. S.N.-Z. supported additional data processing, generation of bioinformatic analysis and gave feedback on analyses and the manuscript. N.M. and V.B. supported study conception, helped to direct bioinformatics analyses and gave feedback on analyses and the manuscript. H.Y.C. supported study conception, additional data processing and direct analysis and gave feedback on the manuscript. N.K. supported study conception and gave feedback on analyses and the manuscript. A.M.F. supported additional data processing and the FISH work and gave feedback on analyses and the manuscript. P.S.M., M.J.-H. and C.S. jointly designed and supervised the study and wrote the manuscript.

**Funding** Open Access funding provided by The Francis Crick Institute.

**Competing interests** C.S. acknowledges grants from AstraZeneca, Boehringer-Ingelheim, Bristol Myers Squibb, Pfizer, Roche-Ventana, Invitae (previously Archer Dx—collaboration in minimal residual disease sequencing technologies), Ono Pharmaceutical and Personalis. C.S. is chief investigator for the AZ MeRmaiD 1 and 2 clinical trials and is the steering committee chair. C.S. is also co-chief investigator of the NHS Galleri trial financed by GRAIL and a paid member of GRAIL's scientific advisory board (SAB). C.S. receives consultant fees from Achilles Therapeutics (also a SAB member), Bicycle Therapeutics (also a SAB member), Genentech, Medicxi, the China Innovation Centre of Roche (formerly the Roche Innovation Centre—Shanghai), Metabomed (until July 2022) and the Sarah Cannon Research Institute. C.S. has received honoraria from Amgen, AstraZeneca, Bristol Myers Squibb, GlaxoSmithKline, Illumina, MSD, Novartis, Pfizer and Roche-Ventana. C.S. has previously held stock options in Apogen Biotechnologies and GRAIL, and currently has stock options in Epic Bioscience and Bicycle Therapeutics, and has stock options and is co-founder of Achilles Therapeutics. C.S. declares patent applications on methods to detect lung cancer (PCT/US2017/028013), targeting neoantigens (PCT/EP2016/059401), identifying patient response to immune checkpoint blockade (PCT/EP2016/071471), determining HLA loss of heterozygosity (PCT/GB2018/052004), predicting survival rates of patients with cancer (PCT/GB2020/050221), identifying patients who respond to cancer treatment (PCT/GB2018/051912) and methods for lung cancer detection (US20190106751A1). C.S. is an inventor on a European patent application (PCT/GB2017/053289) relating to assay technology to detect tumour recurrence. This patent has been licensed to a commercial entity, and under their terms of employment, C.S. is due a revenue share of any revenue generated from such licence(s). N.M. has stock options in and has consulted for Achilles Therapeutics and holds European patents relating to targeting neoantigens (PCT/EP2016/059401), identifying patient response to immune checkpoint blockade (PCT/ EP2016/071471), determining HLA loss of heterozygosity (PCT/GB2018/052004) and predicting survival rates of patients with cancer (PCT/GB2020/050221). P.S.M. is a co-founder, chairs the SAB of and has equity interest in Boundless Bio. P.S.M. is also an advisor with equity for Asteroid Therapeutics and is an advisor to Sage Therapeutics. V.B. is a co-founder, consultant, SAB member and has equity interest in Boundless Bio and Abterra, and the terms of this arrangement have been reviewed and approved by the University of California, San Diego in accordance with its conflict of interest policies. H.Y.C. is a co-founder of Accent Therapeutics, Boundless Bio, Cartography Bio and Orbital Therapeutics, and is an advisor to 10X Genomics, Arsenal Biosciences, Chroma Medicine and Spring Discovery.

## Additional information
**Correspondence and requests for materials** should be addressed to Paul S. Mischel, Mariam Jamal-Hanjani or Charles Swanton.

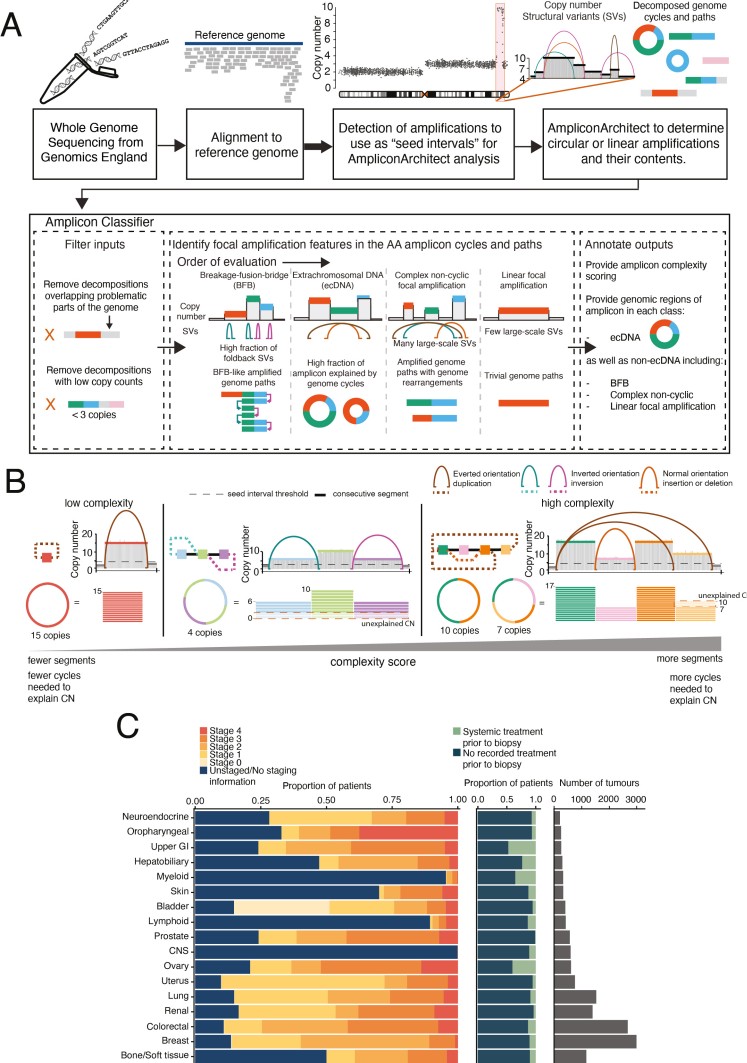

**Extended Data Fig. 1 | Overview of cohort and ecDNA characterisation methodology. a,** Schematic further demonstrating the analysis pipeline applied to the GEL cohort. Each tumour sample is subjected to whole genome sequencing, then aligned to the reference genome (hg38). This sequencing data is then analysed with CNVkit to detect areas of the genome that are amplified with high copy number. The sequencing data corresponding to these areas of amplification are then passed to as seed intervals to AmpliconArchitect which then defines a copy number and structural variant aware amplicon graph that can be decomposed into linear paths and cycles. The output from AmpliconArchitect is then passed to AmpliconClassifier which, after filtering, classifies areas of the amplicon as either breakage-fusion-bridge, ecDNA,

complex non-cyclic focal amplification, or a linear focal amplification as well as performing amplicon complexity scoring. **b,** Schematic illustrating amplicon complexity scoring. (left) Low complexity amplicon - showing a simple graph that consists of a single segment with one cycle that explains all the amplicon copy number. (middle) Higher complexity amplicon showing a more complex graph that consists of three segments whose single cycle is only able to explain the majority of the amplicon copy number. (right) High complexity amplicon showing a graph that consists of 4 segments that can only explain the majority of the amplicon copy number with two cycles. **c,** Stacked bar plots for each tissue type showing the number of samples, whether treatment was received prior to biopsy, and disease stage.

Patient 1 - angiosarcoma
ecDNA predicted by AmpliconArchitect

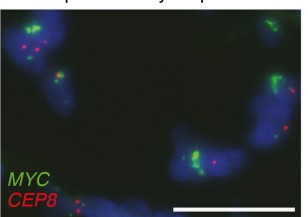

Patient 2 - parosteal osteosarcoma
ecDNA predicted by AmpliconArchitect

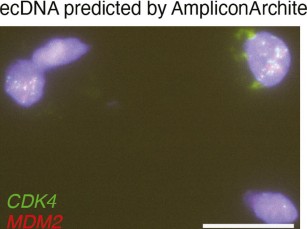

Patient 3 - dedifferentiated liposarcoma
ecDNA predicted by AmpliconArchitect.

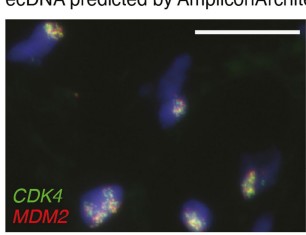

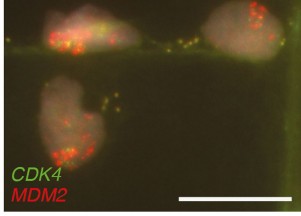

Patient 4 - dedifferentiated liposarcoma
ecDNA predicted by AmpliconArchitect.

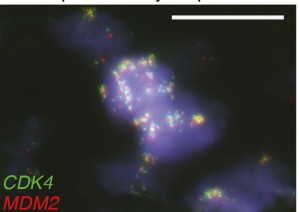

Patient 5 - parosteal osteosarcoma
ecDNA predicted by AmpliconArchitect.

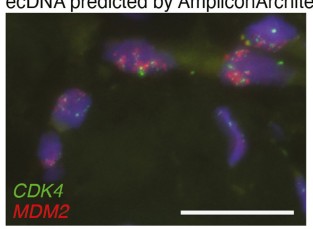

Patient 6 - dedifferentiated liposarcoma
ecDNA predicted by AmpliconArchitect.

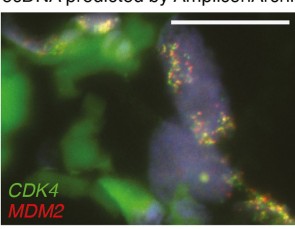
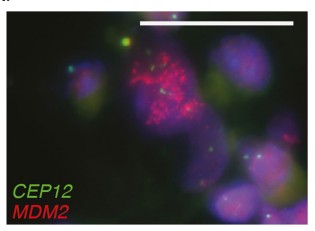

Patient 7 - extraskeletal osteosarcoma
chromosomal amplification predicted by AmpliconArchitect.

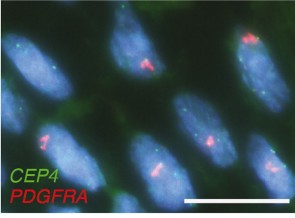

Patient 8 - dedifferentiated liposarcoma
ecDNA predicted by AmpliconArchitect.

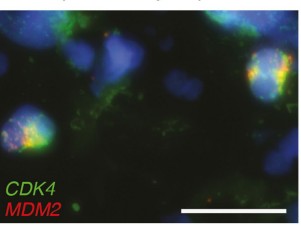

Patient 9 - dedifferentiated liposarcoma
ecDNA predicted by AmpliconArchitect

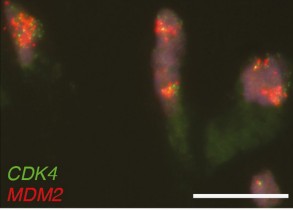

Patient 10 - dedifferentiated liposarcoma
ecDNA predicted by AmpliconArchitect.

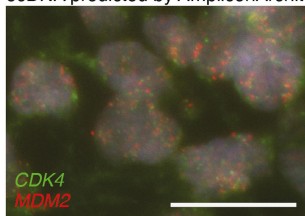

Patient 11 - parosteal osteosarcoma
ecDNA predicted by AmpliconArchitect.

**Extended Data Fig. 2 | FISH of Genomics England sarcoma samples.** Images of representative fields of view from interphase FISH images for each of the 11 sarcoma tumours from the GEL cohort for which material was available. Each image is annotated with the FISH probes applied in addition to DAPI (blue) and the AmpliconArchitect prediction of the presence of either a chromosomal amplification or ecDNA. Scale bar is 20 μm.

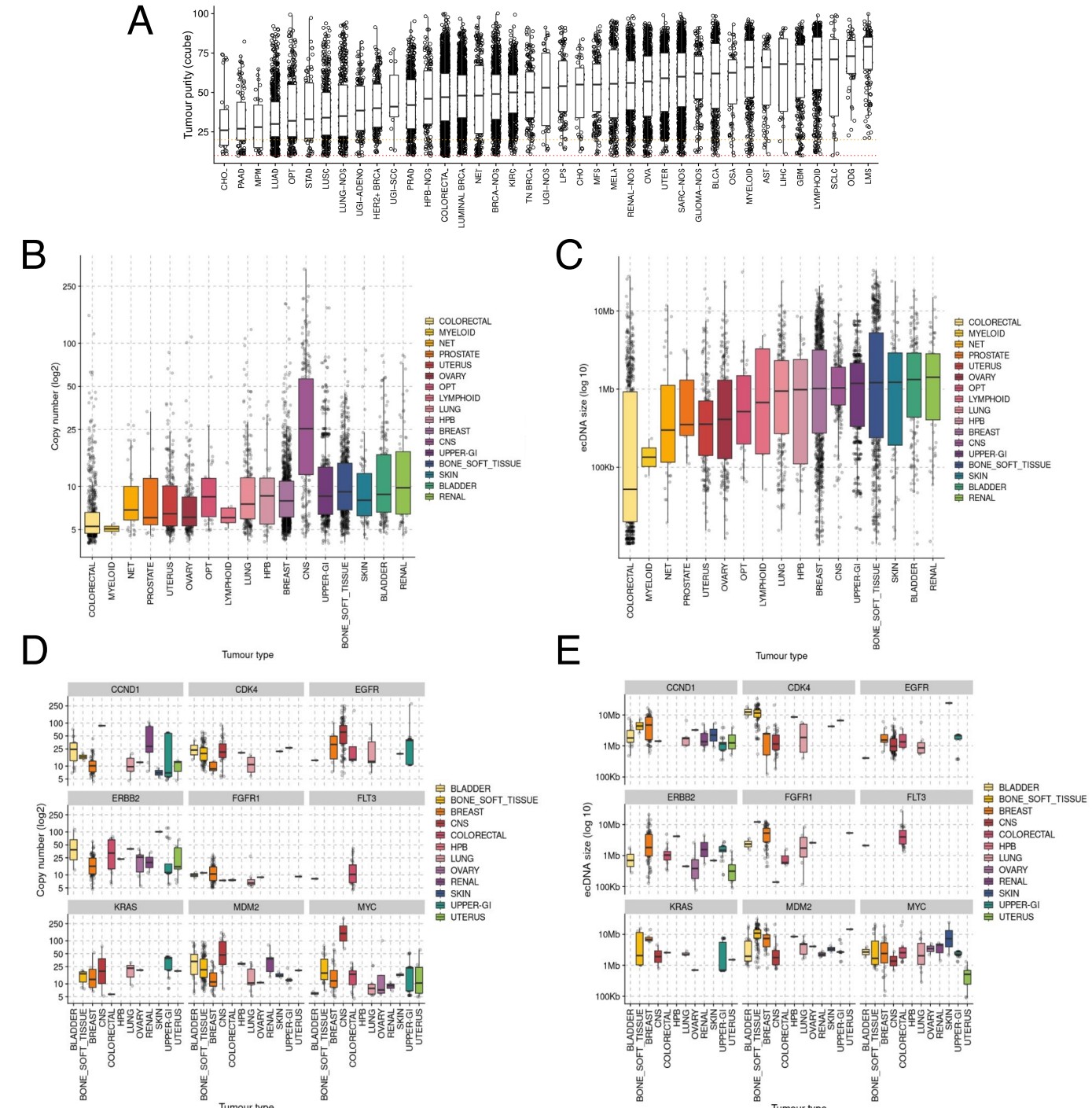

**Extended Data Fig. 3 | Tumour purity, ecDNA copy number, and ecDNA size.** **a**, Box plots showing purity estimates for each of tumour samples from the 39 cancer types in the GEL cohort. **b**, Box plots showing the log-transformed copy number of ecDNA grouped by tissue type. **c**, Box plots showing the size in megabases of ecDNA grouped by tissue type. **d**, Box plots of log-transformed copy number of ecDNA grouped by the oncogenes present on the ecDNA and tissue type. **e**, Box plots of ecDNA size in megabases grouped by the oncogenes present on the ecDNA and tissue type.

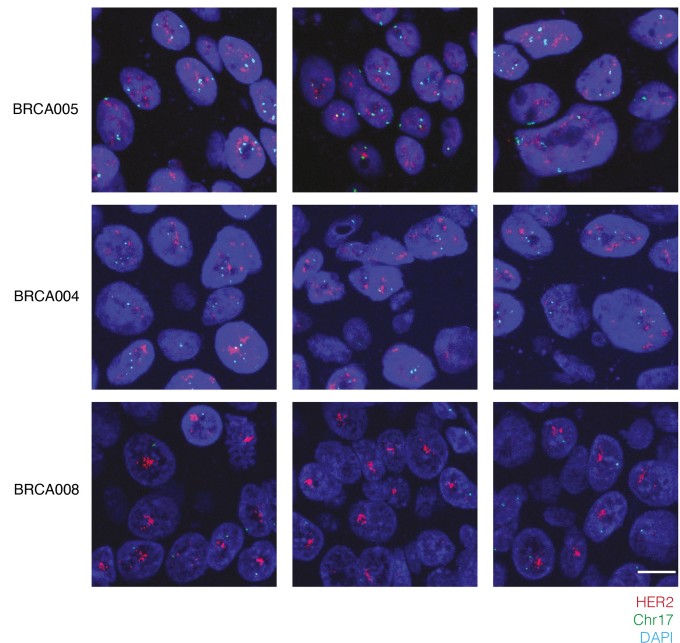

BRCA005

BRCA004

BRCA008

HER2
Chr17
DAPI

Representative fields of view from three HER2-amplified breast cancer samples. Scale bar is 10um

**Extended Data Fig. 4 | FISH of HER2+ breast cancer samples.** Images of representative fields of view from interphase FISH images for each of the 3 HER2+ breast cancers from an independent cohort for which material was available for FISH but not for WGS.

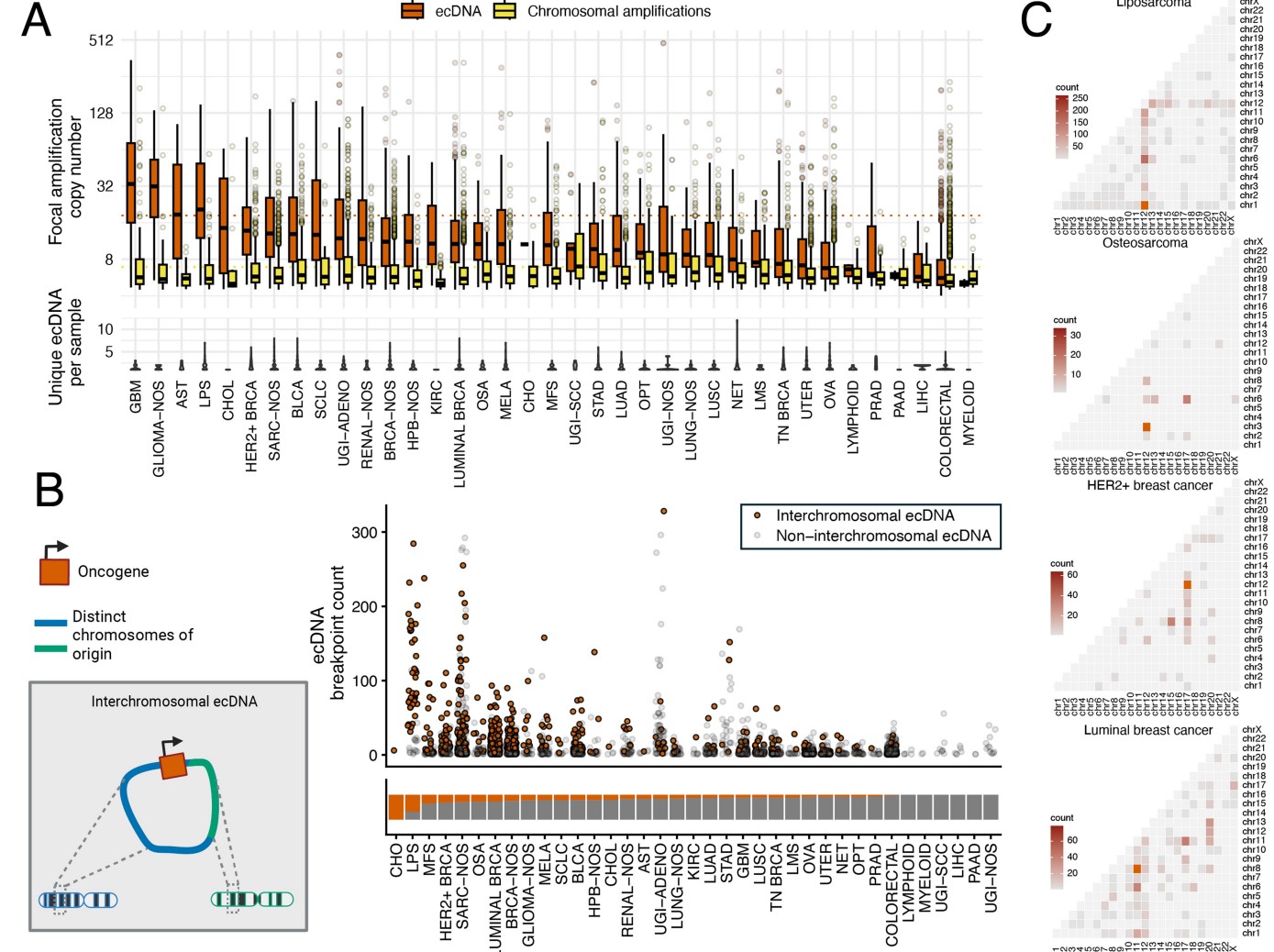

**Extended Data Fig. 5 | Focal amplifications and interchromosomal ecDNA.**
**a**, (top) Boxplots of focal amplification copy number for ecDNA (red) and
chromosomal amplifications (yellow) grouped by cancer type. (bottom) Violin
plot of the number of ecDNA observed per tumour grouped by cancer type.
**b**, (left) schematic demonstrating the concept of interchromosomal ecDNA.
(right-top) Strip plot showing the count of interchromosomal ecDNA
breakpoints for interchromosomal ecDNA (red) and non-interchromosomal

ecDNA (light grey) grouped by cancer type. (right-bottom) Stacked bar plot
showing the proportion of ecDNA observed for each cancer type that is
interchromosomal (red) or non-interchromosomal (grey). The graphics
in the schematic were created with BioRender.com. **c**, Heatmaps depicting
chromosome pairs involved in interchromosomal ecDNA for liposarcoma,
osteosarcoma, HER2+ breast cancer and luminal breast cancer.

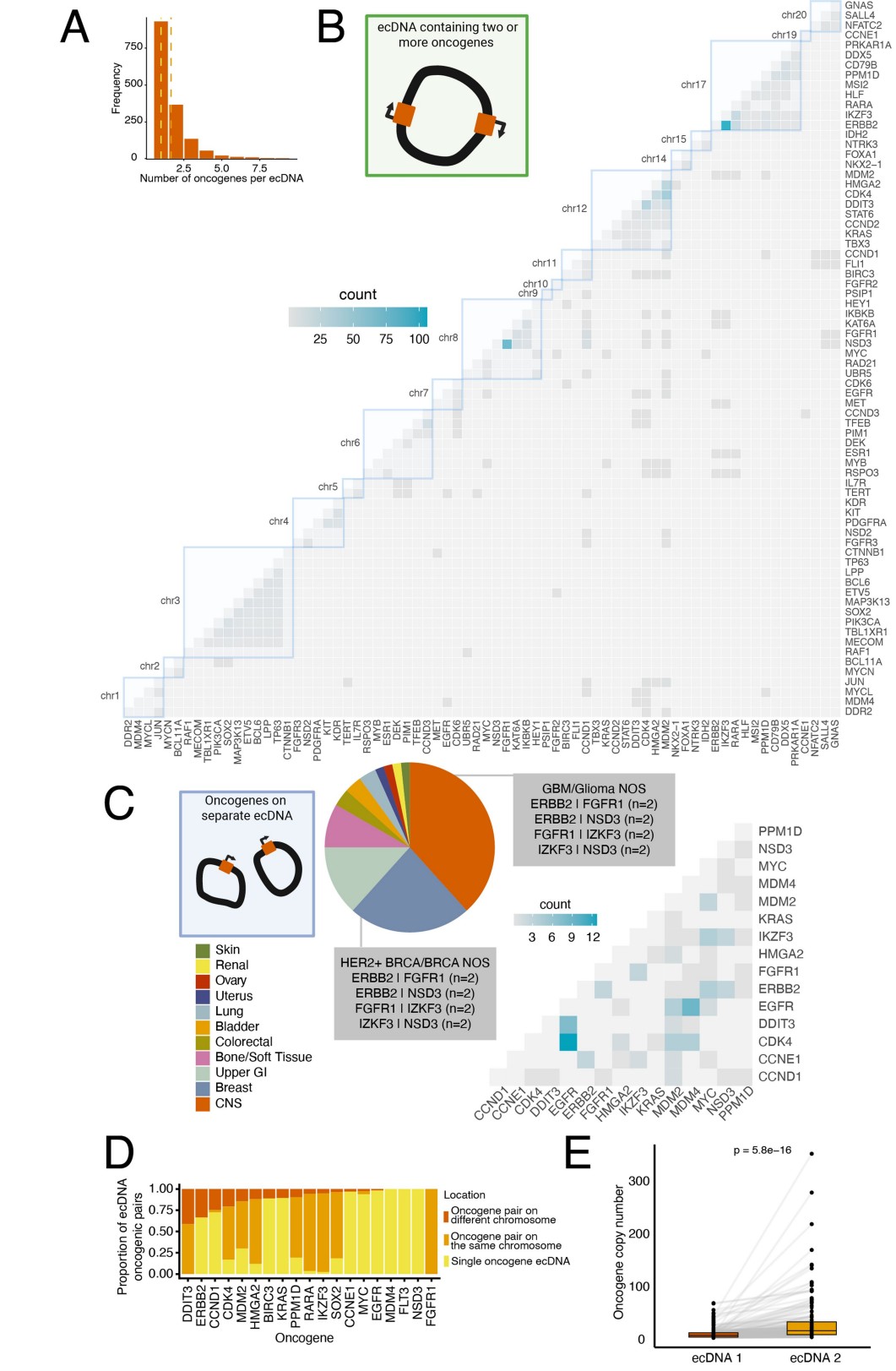

**Extended Data Fig. 6** | See next page for caption.

**Extended Data Fig. 6 | Multi-oncogene ecDNA and multiple ecDNA per sample. a**, Bar plot showing the frequency at which one or more oncogenes are found on the same ecDNA; the yellow dotted line represents the median number of ecDNA per sample and the orange dotted line represents the mean number of ecDNA per sample. **b**, (left) Schematic demonstrating an ecDNA containing two or more oncogenes. (right) Heatmap showing how often pairs of oncogenes are found on the same ecDNA. Blue outlined boxes indicate potential oncogene pairs consisting of oncogenes from the same chromosome. The corresponding chromosome is labeled to the left of the blue outlined box. **c**, (left) Schematic demonstrating the presence of two or more oncogenes present on distinct ecDNA within the same tumour. (middle) Pie chart showing the proportion of tumours demonstrating different oncogenes on separate ecDNA that belong to each tissue type included in the study. Grey boxes highlight specific gene pairs present in the two tissue types with the highest proportion of oncogenes present on separate ecDNA in the same tumour. (right) Heatmap showing pairs of oncogenes found on separate ecDNA in the same tumour. **d**, Stacked bar plot showing the proportion of oncogenes observed on ecDNA in ≥20 tumours that were found in the context of an ecDNA containing that single oncogene (yellow), in a pair of oncogenes from the same chromosome (orange), or in a pair of oncogenes from different chromosomes (brown). **e**, Box plots examining the copy number of pairs of oncogenes identified on distinct ecDNAs found in the same tumour (wilcoxon p value shown at the top of the plot). The graphics of the ecDNAs in **b,c** were created with BioRender.com.

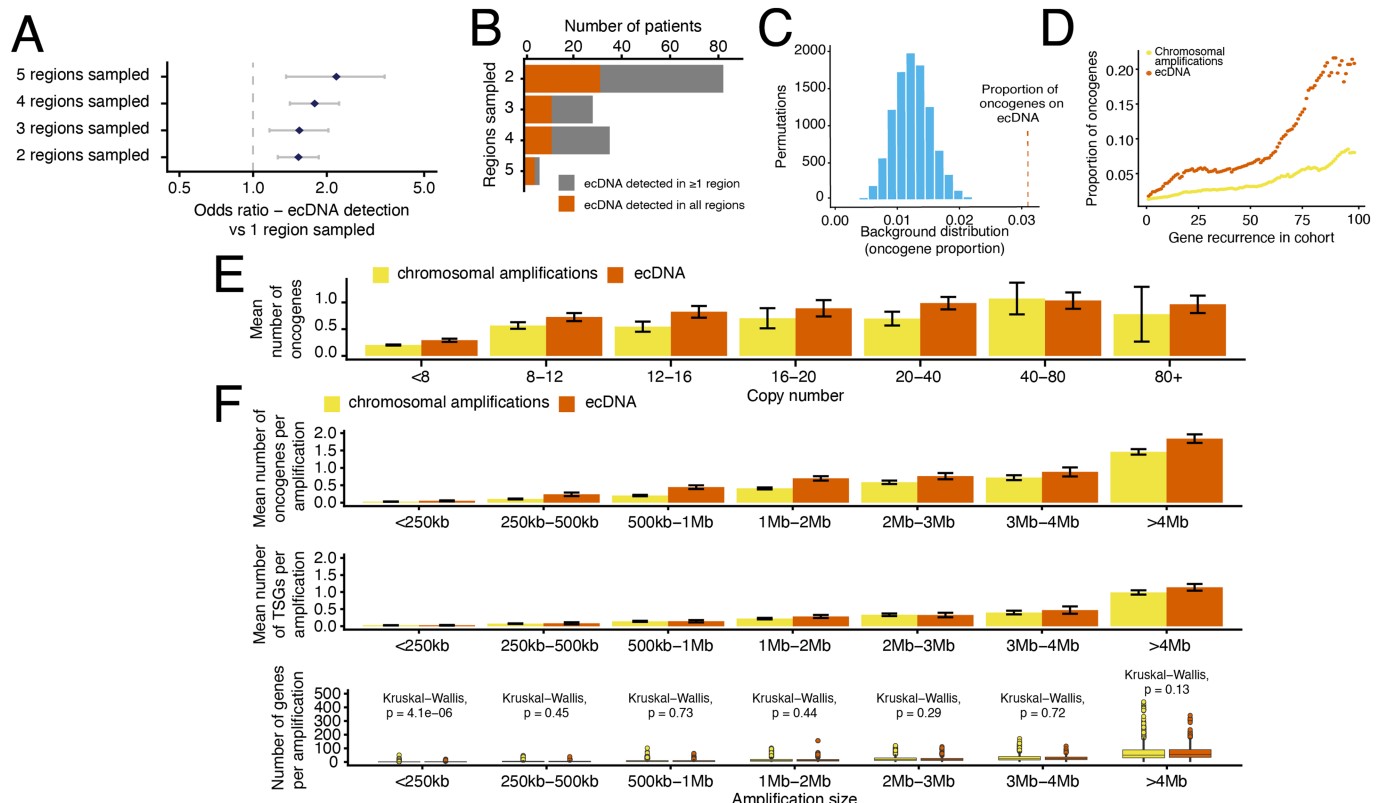

**Extended Data Fig. 7 | Multi-sample ecDNA detection and ecDNA gene complement characteristics. a**, Forest plot of results of a regression model examining the detection of ecDNA in relation to the number regions sampled from the same tumour. **b**, Stacked bar plot showing the number of patients with ecDNA detected and multiple regions sampled from their tumour. The colours indicate whether ecDNA was detected in ≥1 region (but not all regions, grey) or all regions (red). **c**, Bar plot showing permutation testing results demonstrating an enrichment of oncogenes in focal amplifications. **d**, Scatterplot showing comparison of ecDNA (dark red) and chromosomal amplifications (yellow) for oncogene enrichment. The x-axis represents a sliding window that quantifies the proportion of all genes that are oncogenes at each number of recurrences in the cohort. **e**, Bar plots showing the mean number of oncogenes stratified by copy number. **f**, Bar plots showing (top) the mean number of oncogenes, (middle) the mean number of tumour suppressor genes, (bottom) the total number of genes for chromosomal amplifications and ecDNA grouped by size. Bars that are red signify ecDNA and yellow signifies chromosomal amplifications.

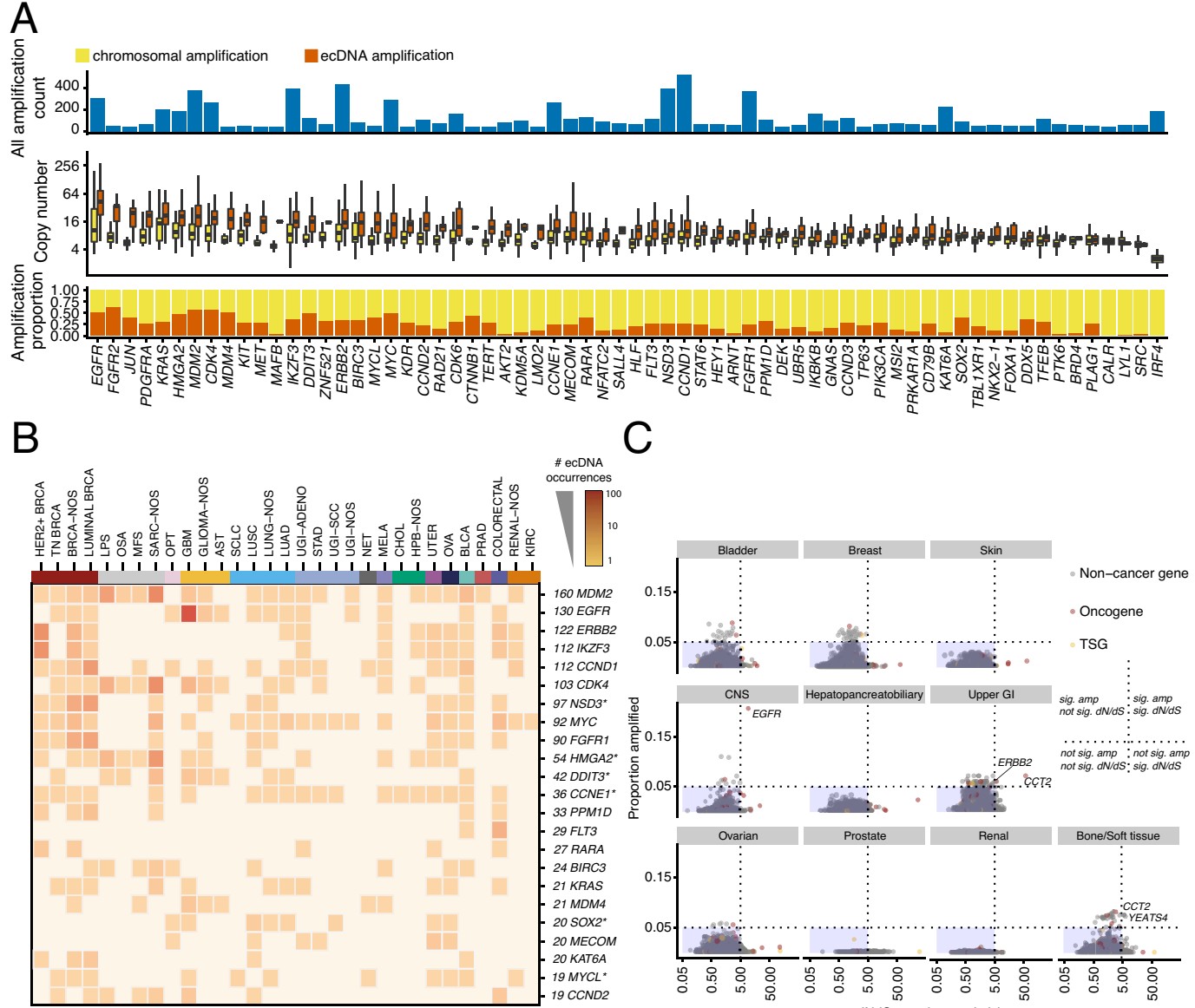

**Extended Data Fig. 8 | Oncogene frequency on ecDNA. a**, (top) Bar plot showing the frequency of amplifications of any kind (either ecDNA or chromosomal) affecting oncogenes in the GEL cohort. (middle) Box plots of copy number for oncogenes when affected by chromosomal amplifications (yellow) or ecDNA (red). (bottom) Stacked bar plot showing the proportion of amplifications for each oncogene that are chromosomal (yellow) or ecDNA (red). **b**, Heatmap showing oncogenes recurrently amplified (n > 18) on ecDNA by cancer type. Tissue type is indicated by the coloured bar at the top of the heatmap. The colour of each cell indicates the relative frequency of its occurrence compared to all other oncogenes on ecDNA. The total number of ecDNAs carrying a particular oncogene is displayed to the right of the heatmap

before the name of the oncogene. Genes marked with an asterisk are those also present in the IntOGen database[36] and the Cancer Gene Census. **c**, Scatter plots for each tissue type showing the dN/dS estimates of genes in the GEL cohort (x-axis) versus fraction of the genes subject to amplification (y-axis). Oncogenes are represented by red dots, tumour suppressor genes by yellow dots, and non-cancer genes by grey dots. "sig. amp" is an abbreviation for "significantly amplified" and "sig. dNdS" is an abbreviation for "significant by dNdSvc analysis". Only genes significantly amplified and found significant by dN/dS analysis are annotated and are found in the top right quadrant of each subplot for a tissue type. Only tissue types with sufficient mutations and amplifications for the analysis were included.

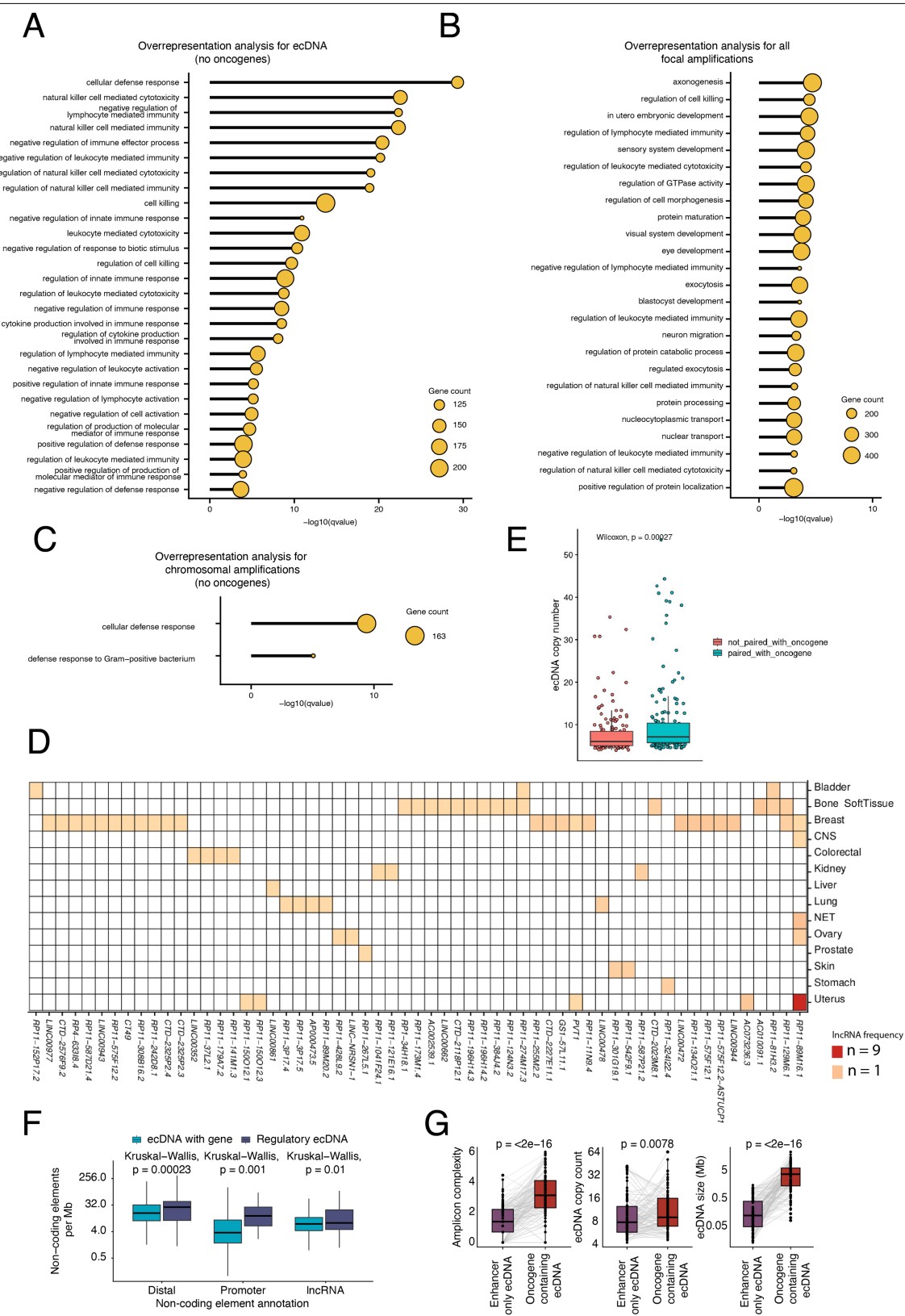

**Extended Data Fig. 9 | Immunomodulatory genes and non-coding elements on ecDNA. a,** Plot showing results of an overrepresentation analysis of genes present on ecDNA that do not contain oncogenes. **b,** Plot showing results of an overrepresentation analysis of genes found in any focal amplification. **c,** Plot showing results of an overrepresentation analysis of genes found in chromosomal amplifications that do not contain oncogenes. **d,** Heatmap of lncRNAs found on ecDNA that do not contain protein coding genes. **e,** Box plot showing the copy number of enhancer-only ecDNAs present in a sample without oncogene containing ecDNA and the copy number of enhancer-only ecDNAs present in a sample that also contains a separate oncogene carrying ecDNA. **f,** Boxplots showing the number of non-coding elements per megabase in ecDNA with oncogenes and ecDNA with only regulatory ecDNA. **g,** Box plots showing comparisons of (left) the amplicon complexity, (middle) ecDNA copy number count, (right) ecDNA size in megabases in enhancer only ecDNA versus ecDNA containing oncogenes.

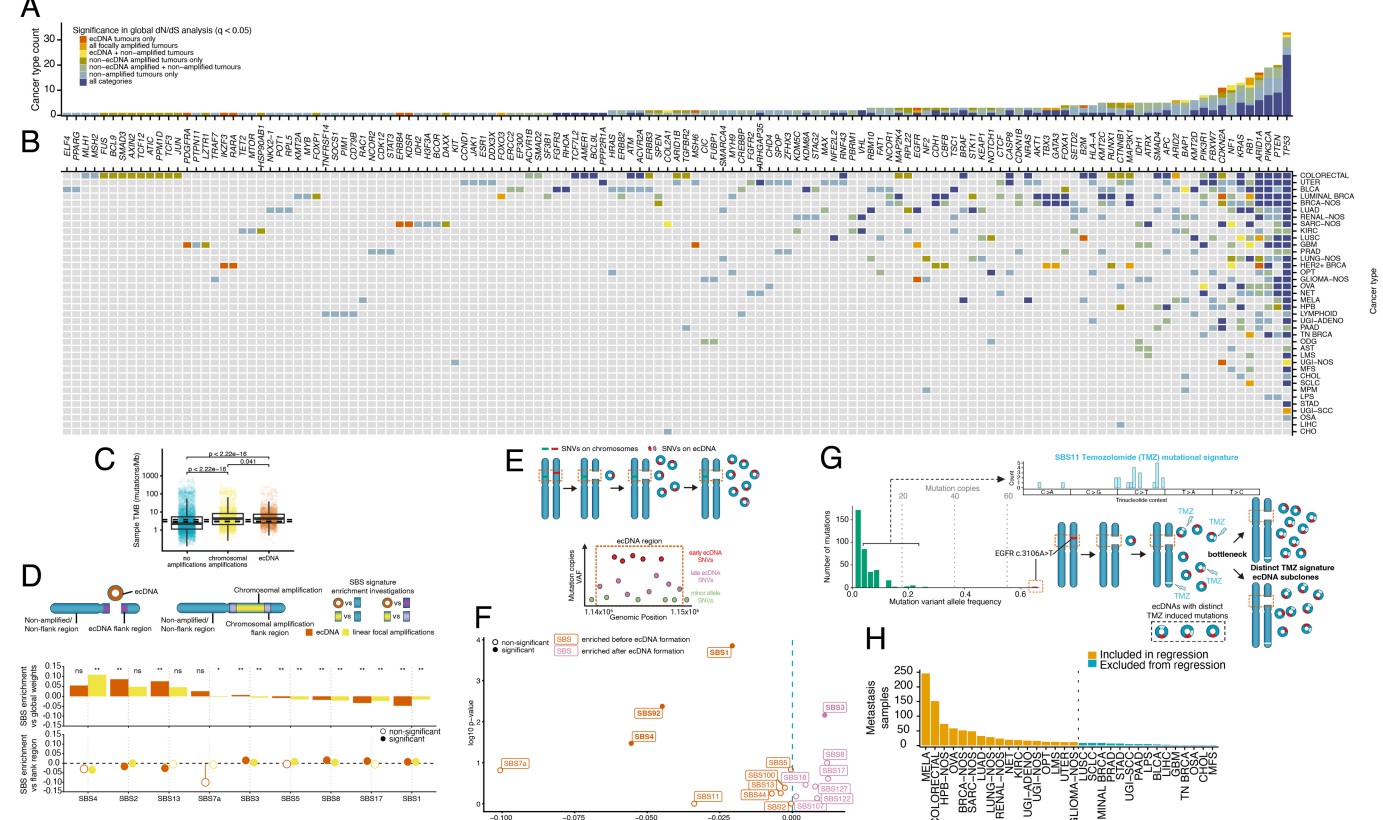

**Extended Data Fig. 10 | Mutational processes associated with focal amplifications.** dN/dS analysis of all genes across the cohort. Tumour samples were categorised by whether they were found to contain ecDNA, chromosomal amplifications, or no amplifications. dN/dS analysis was then performed in these subcategories for each cancer type. **a**, Stacked bar plot showing the total number of categories across all cancer types for which a gene was found to be under positive selection. Colours represent the combination of subcategories in which the gene was found to be under positive selection for a cancer type. **b**, Heatmap showing for each gene which cancer types it was found to be under positive selection and in which subcategories of tumour samples defined by the presence or absence of ecDNA and chromosomal amplifications. **c**, Box plots showing comparison of TMB between samples with ecDNA (dark red), only chromosomal amplifications (yellow), or no focal amplifications

of any kind (blue). Dashed line represents the median TMB per category. **d**, (top) Schematic illustrating the method of comparing SBS signatures in focal amplifications, either chromosomal or ecDNA, with those in flanking regions as well as with global signatures. (bottom) Plots demonstrating SBS enrichments. **e**, Schematic illustrating method of timing mutations in areas of the genome amplified by ecDNA. **f**, Scatter plot showing relative changes in SBS signature probability pre and post ecDNA formation. **g**, Schematic illustrating a case with a post-temozolomide treatment SBS11 mutational signature of mutations present on ecDNA in a tumour that also has an early *EGFR* mutation also present on the same ecDNA. **h**, Bar plot showing the number of metastatic samples per cancer type and which cancer types were included in the regression model.

# Reporting Summary

## Statistics

For all statistical analyses, confirm that the following items are present in the figure legend, table legend, main text, or Methods section.

| n/a | Confirmed | |
|---|---|---|
| ☐ | ☒ | The exact sample size (*n*) for each experimental group/condition, given as a discrete number and unit of measurement |
| ☐ | ☒ | A statement on whether measurements were taken from distinct samples or whether the same sample was measured repeatedly |
| ☐ | ☒ | The statistical test(s) used AND whether they are one- or two-sided<br>*Only common tests should be described solely by name; describe more complex techniques in the Methods section.* |
| ☐ | ☒ | A description of all covariates tested |
| ☐ | ☒ | A description of any assumptions or corrections, such as tests of normality and adjustment for multiple comparisons |
| ☐ | ☒ | A full description of the statistical parameters including central tendency (e.g. means) or other basic estimates (e.g. regression coefficient) AND variation (e.g. standard deviation) or associated estimates of uncertainty (e.g. confidence intervals) |
| ☐ | ☒ | For null hypothesis testing, the test statistic (e.g. *F*, *t*, *r*) with confidence intervals, effect sizes, degrees of freedom and *P* value noted<br>*Give P values as exact values whenever suitable.* |
| ☒ | ☐ | For Bayesian analysis, information on the choice of priors and Markov chain Monte Carlo settings |
| ☐ | ☒ | For hierarchical and complex designs, identification of the appropriate level for tests and full reporting of outcomes |
| ☐ | ☒ | Estimates of effect sizes (e.g. Cohen's *d*, Pearson's *r*), indicating how they were calculated |

*Our web collection on statistics for biologists contains articles on many of the points above.*

## Software and code

Policy information about availability of computer code

| Data collection | No software was used |
|---|---|
| Data analysis | Focal DNA copy number alterations were identified using CNVKit v0.98. Amplicon Architect v1.2 was used to construct cyclic paths from identified focal amplifications, and Amplicon Classifier version 0.4.12 was used to determine whether these paths were likely to be ecDNA. Amplicon Architect is a tool that identifies the structure of focal amplifications by using seed intervals that define regions that are focally amplified and extend beyond them to look for copy number changes or discordant edges. For this analysis, focal amplifications were defined as regions of over 50Kb, with a copy number greater than four and twice the ploidy estimation of the chromosome arm.<br><br>R packages used in version 4.0.2:<br>survival (v3.2.11)<br>survminer (v0.4.9)<br>fst (version 0.9.4)<br>tidyverse (version 1.3.0)<br>ggplot2 (version 3.3.2)<br>dplyr (version 1.0.2)<br>tidyr (version 1.1.2)<br>gridExtra (version 2.3)<br>cowplot (version 1.1.0)<br>ggpubr (version 0.4.0)<br>reshape2 (version 1.4.4)<br>tibble (version 3.0.4) |

RColorBrewer (version 1.1-2)
plyr (version 1.8.6)
dndscv (version 0.0.1.0)
deconstructSigs (version 1.9.0)
ggrepel (version 0.8.2)
GenomicRanges (version 1.38.0)
stringr (version 1.4.0)
data.table (version 1.13.2)
magrittr (version 2.0.1)
ComplexHeatmap (version 2.4.5)

For manuscripts utilizing custom algorithms or software that are central to the research but not yet described in published literature, software must be made available to editors and reviewers. We strongly encourage code deposition in a community repository (e.g. GitHub). See the Nature Portfolio guidelines for submitting code & software for further information.

## Data

Policy information about availability of data

All manuscripts must include a data availability statement. This statement should provide the following information, where applicable:
- Accession codes, unique identifiers, or web links for publicly available datasets
- A description of any restrictions on data availability
- For clinical datasets or third party data, please ensure that the statement adheres to our policy

All data is available following application to access to the Genomics England Research Environment.

## Research involving human participants, their data, or biological material

Policy information about studies with human participants or human data. See also policy information about sex, gender (identity/presentation), and sexual orientation and race, ethnicity and racism.

| | |
|---|---|
| Reporting on sex and gender | Sex (biological attribute) was used as an explanatory variable in the Cox proportional hazards model for the association with ecDNA presence and survival |
| Reporting on race, ethnicity, or other socially relevant groupings | We did not report on race, ethnicity or other socially relevant groupings |
| Population characteristics | For the Cox proportional hazards model, patients were grouped according to sex (biological attribute; male = 3262, female = 5615) and age groups (0-44 years, n = 481; 45-59 years, n = 2215; 60 - 69 years, n = 2648; 70-79 years, n = 2581; 80+ years, n = 951) |
| Recruitment | Patients were recruited to eleven Genomic Medicine Centres across the United Kingdom; |

Recruitment (continued):

- East of England NHS Genomic Medicine Centre (Led by Cambridge University Hospitals NHS Foundation Trust)
- Greater Manchester NHS Genomic Medicine Centre (Led by Central Manchester University Hospitals NHS Foundation Trust)
- West Midlands NHS Genomic Medicine Centre (Led by University Hospitals Birmingham NHS Foundation Trust)
- North East & North Cumbria NHS Genomic Medicine Centre (Led by Newcastle upon Tyne Hospitals NHS Foundation Trust)
- North Thames NHS Genomic Medicine Centre (Led by Great Ormond Street Hospital NHS Foundation Trust)
- North West Coast NHS Genomic Medicine Centre (Led by Liverpool Women's NHS Foundation Trust)
- Oxford NHS Genomic Medicine Centre (Led by Oxford University Hospitals NHS Trust (OUH))
- South London NHS Genomic Medicine Centre (Led by Guy's and St Thomas' NHS Foundation Trust)
- South West NHS Genomic Medicine Centre (Led by Royal Devon & Exeter NHS Foundation Trust)
- Wessex NHS Genomic Medicine Centre (Led by University Hospital Southampton NHS Foundation Trust)
- West London NHS Genomic Medicine Centre (Led by Imperial College Healthcare NHS Trust)

Data release version 11 was used for the analysis and was launched in 17/12/2020, from which 15,609 participants are included.

| | |
|---|---|
| Ethics oversight | Genomics England has approval from the HRA Committee East of England – Cambridge South (REC Ref 14/EE/1112). |

Note that full information on the approval of the study protocol must also be provided in the manuscript.

# Field-specific reporting

Please select the one below that is the best fit for your research. If you are not sure, read the appropriate sections before making your selection.

☒ Life sciences    ☐ Behavioural & social sciences    ☐ Ecological, evolutionary & environmental sciences

For a reference copy of the document with all sections, see nature.com/documents/nr-reporting-summary-flat.pdf

# Life sciences study design

All studies must disclose on these points even when the disclosure is negative.

| | |
|---|---|
| Sample size | We utilised the Genomics England version 7 cohort, comprising 16,355 whole genome sequenced samples. We included the following tumour types: breast, lung, stomach, neuroendocrine, skin, oropharyngeal, colorectal, kidney, prostate, hepato-pancreatobiliary, bladder, bone and soft tissue, ovary, endometrium, central nervous system (CNS), lymphoid and myeloid. |
| Data exclusions | For quality control, we excluded samples with an estimated purity of < 10%. Paediatric and germ cell tumours were also excluded. For mutational signature analysis the tumour purity cut off was 20%. |
| Replication | All available data was analysed. |
| Randomization | Randomisation is not relevant as this is an observational study. |
| Blinding | Blinding is not relevant as this is an observational study |

# Reporting for specific materials, systems and methods

We require information from authors about some types of materials, experimental systems and methods used in many studies. Here, indicate whether each material, system or method listed is relevant to your study. If you are not sure if a list item applies to your research, read the appropriate section before selecting a response.

## Materials & experimental systems

| n/a | Involved in the study |
|---|---|
| ☒ | Antibodies |
| ☒ | Eukaryotic cell lines |
| ☒ | Palaeontology and archaeology |
| ☒ | Animals and other organisms |
| ☒ | Clinical data |
| ☒ | Dual use research of concern |
| ☒ | Plants |

## Methods

| n/a | Involved in the study |
|---|---|
| ☒ | ChIP-seq |
| ☒ | Flow cytometry |
| ☒ | MRI-based neuroimaging |

## Plants

| | |
|---|---|
| Seed stocks | NA |
| Novel plant genotypes | NA |
| Authentication | NA |

