## [Peer Review File · Nature]

Manuscript Title: Origins and Impact of Extrachromosomal DNA

Reviewer Comments & Author Rebuttals

Reviewer Reports on the Initial Version:

Referees' comments:

Referee #1 (Remarks to the Author):

Bailey et al performed DNA sequencing analysis of 14,778 cancer patients from 39 tumour types to investigate the origins and impact of extrachromosomal DNA (ecDNA) in cancer. Using this comprehensive dataset, Bailey et al reveal multitude properties of ecDNA across many cancer types, convincingly showing their importance and contribution to cancer development. Specifically, this work reveals the following points: (1) ecDNA amplification are widespread across multiple cancer types, (2) ecDNA contain not only oncogenes but also immunomodulatory genes and lncRNAs, (3) ecDNA correlates with high tumor mutational burden, and with specific mutation signatures, (4) ecDNA correlate with metastatic disease, and (5) ecDNA correlate with poorer survival.

As ecDNA amplifications are increasingly recognized as major contributors to cancer, this work is of high importance, and provides critical information on their role. However, there are several concerns that need to be addressed as outlined below.

Major comments:

1. There is very little information provided on the Genomics England dataset used. Does this include treated and untreated patients? If so, did the authors analyze such samples separately? Do treated patients have increased immunomodulatory ecDNAs, for example? What sequencing approach was used (coverage etc.)? In the current form, this dataset is a black box. The computational methodology used to infer ecDNA should also be discussed, and the degree of confidence of the approach used needs to be illustrated. Can the authors reconstruct ecDNA structures (even complex ones) with high degree of confidence – and can they provide examples?
2. In continuation to the first point, the methods section could be more informative and detailed. Figure legends and Figures could also include more detailed information, better explaining each feature within the presented data (meaning of colors, for example in Figure 3, but also in other figures). Better explanation of different bioinformatic parameters would make the paper more approachable to non-computational audiences – for example, explanation of dn/ds, SBS, amplicon entropy/complexity.
3. The paper title emphasizes this work studies the origin of ecDNA. There are several aspects that could be improved with this regard:

a. Analysis of the chromosomes of origin of ecDNA would be very informative on the mechanisms of ecDNA formation. Are there any genomic scars detected? Is there enrichment for structural variations (SVs)? Such analysis should consider the fact that in some cases the chromosome of origin is lost (for example through chromothripsis).

b. The authors use the term Amplicon Entropy (which is not well defined in the methods section) to examine ecDNA structure complexity. However, the authors do not provide specific details that would be informative. ecDNA formed through chromothripsis would be expected to include discordant DNA fragments at higher frequency (fragments from distant chromosome loci), and also greater SV complexity (but this could emerge later during the lifetime of the ecDNA).

c. Analysis of the breakpoint junctions within ecDNA could be informative on the repair mechanisms operative in the formation and maintenance of amplifications.

d. Line 105: "...two or more oncogenes on distinct ecDNAs". What about the frequency of two or more oncogenes on the same ecDNA (such as in cases of MDM2 and CDK4)?

e. Mutations in ecDNA (Figure 1D-E): Do the mutations correlate with CN? Are the mutations found in the ecDNA or in the endogenous chromosomal region?

f. Is there selection for non-coding regulatory regions on ecDNA (as seen in PMID: 31761532)?

4. Lines 120-12 and Figure 1C: "... genes recurrently encoded by ecDNA were more likely to be oncogenes compared with non-ecDNA driven focal amplifications". Could oncogene likelihood be correlated with amplicon size rather than location (ecDNA/chromosomal amplification)? Actually, could amplicon size be used to predict the evolutionary stage of amplification? For example, larger amplicons might be newer and through selection and structural variation of amplifications become smaller in size (to contain only the necessary genes). Also, what is the relationship between ecDNA and chromosomal amplifications – can the authors drill down the mechanisms to identify potential break-fusion-bridge (BFB) events and ectopic integration of ecDNAs?

5. Figure 2A shows the distribution of ecDNA CN and size. The distributions appear to be very large. Although ecDNA heterogeneity is expected, to what extent does the confidence of ecDNA calling algorithm affect this result? Would more stringent computational analysis reduce this heterogeneity? Also, the scale of this analysis (pan-cancer) is not very informative on its own. Could this analysis be extended to show CN and sizes per cancer type, and also per amplified genes (would this distribution be similar between MYC and other genes, for example)?

6. Several findings in this work were demonstrated, to some extent, before, in some cases by co-authors of the present study. The authors are encouraged to discuss them in proper context:

a. p53 mutations and MDM2 amplifications are mutually exclusive (PMID: 27194168). Related to Figure 3C.

b. The extent of ecDNA amplification across tumor types and different genes (PMID: 32807987).

c. Correlation of ecDNA with poor survival (PMID: 32807987). Related to Figure 5D.

d. Relationship of ecDNA and clustered mutations (PMID: 33361815, PMID: 35140399).

e. Presence of immunomodulatory genes in ecDNA (PMID: 37046089).

7. The study lacks orthogonal approaches that are important to validate some of the findings:
 - a. The presence of ecDNA could be validated using breakpoint PCR experiments.
 - b. The presence and copy number of ecDNA of different types could be measured using straightforward DNA FISH using tissue sections. Are immunomodulatory genes really amplified, or only exist in ecDNA? To what extent the amplifications called using their computational tools and applied to this particular dataset (Genomics England) is accurate? Validation of oncogene amplifications using DNA FISH is important.
 - c. Figure 2D – T cell exhaustion. For example, study of sections from samples with immunomodulatory ecDNA could show the enrichment (or lack thereof) of T cells in tumor sites that contain immunomodulatory ecDNA (using IF-FISH). Alternatively, single cell RNAseq could be informative.
 - d. The relevance and importance of lncRNA in this study is not clear. Does this lead to any functional outcome, e.g. increased RNA expression? By the same token, do samples with lncRNAs show decreased expression of “sponged” genes? RNA sequencing would be useful here.

8. Analysis of multiregion biopsies provided a glimpse of how measurements could underestimate ecDNA presence (Line 146-148). Could the authors estimate the true ecDNA % based on biopsy sizes? Do we detect more ecDNA when examining a larger biopsy, or do we detect more ecDNA when we sample multiple biopsies (perhaps indicating that in some cases technically we miss them)?

9. Figure 3 does not clearly show the correlates of genome instability and ecDNA. Could the authors improve this figure and distill the main idea?

10. What does Figure 4 teach us regarding ecDNA evolutionary trajectories? Can mutation rates be used to estimate the age of ecDNAs? Theoretically, more DNA copies should have higher chances of acquiring mutations than wild-type conditions, and this difference could be used to infer times.

11. Figure 5B shows a correlation between ecDNA and tumor staging. Does this differentiate between the possibilities that in increased stages the chances to detect ecDNA increase (because it is more widespread now through clonal domination), or that new ecDNA form at advanced stages?

12. Lines 422-435: “We highlight that the relationship between ecDNA and genomic instability differs according to tumour type. This provides further evidence suggesting that ecDNA may arise from different mechanisms, including episomal formation, chromothripsis, breakage-fusion-bridge cycling and translocation-bridge-amplifications.” This statement in the discussion relates to Figure 3 but the data does not clearly illustrate the potential involvement of any of the mentioned mechanisms. Related to this, BFB leading to ecDNA and the presence of kataegis in such cases was also not examined (or discussed).

13. Figure 5A related to metastasis and correlation with ecDNA is not clear. It appears all amplification types correlate with metastasis. Also, do the amplifications pre exist in the primary tumors (can they compare metastatic vs primary from the same patients, if they have such samples?). If there is any way to determine if the amplifications formed de novo at the metastatic site – that would be extremely

interesting.

Minor comments:

1. Line 202 refers to Figure 2E. Shouldn't it refer to Figure 2D instead?
2. Figure 2D: what is the Y axis label?
3. Line 86 claims: "Focal amplifications were defined as regions of the genome between 50kb and 20Mb in size". However, in their analysis in Figure 2A the authors claim: "...the median ecDNA size was larger for ecDNA that 182 encoded oncogenes compared with the other two categories (figure 2A, median ecDNA size 2.26 183 Mb vs 0.535 Mb vs 0.053 Mb, $p < 10^{-16}$). If the limit of amplicon size was set to 50kb, how could non-coding ecDNA have a median of 53kb?"
4. Figures 5E and 5D are mixed in the text?

In summary, the work has great potential, and the dataset used is a treasure trove that could illuminate the biology of ecDNA in cancer. By addressing the points above, this work could have great impact within the field and also to broader audiences interested in cancer genome evolution. The most exciting findings here are (1) the presence of non-oncogenes amplified on ecDNA (immune related, lncRNA), (2) the possibility of using mutations to delineate ecDNA evolution trajectories, and (3) the role of ecDNA in metastatic sites. Strengthening these points, also using orthogonal approaches, will provide substantial novelty and impact.

Referee #2 (Remarks to the Author):

Bailey and colleagues set out to detect and analyse focal amplifications, defined as 50kb-20mb regions with at least four copies and twice the ploidy, across the Genomics England (GEL) cohort, a relatively untapped resource that includes tumour whole-genome sequences from more than 15,000 patients with cancer. In particular, the analysis focused on extrachromosomal DNA amplifications (ecDNAs). The large and diverse number of tumours included in the GEL cohort enabled the authors to expand the ecDNA landscape beyond their previous publication (Kim et al), such as by disease subtype (i.e. HER2+ breast cancers). The depth and breadth of the cohort further resulted in several new discoveries, such as an enrichment for the presence of immunomodulatory genes on ecDNA amplicons, beyond the previously established frequent presence of oncogenes. Cross-referencing ecDNA-containing tumours with mutational signatures detected an association with for example APOBEC signatures and a negative correlation with hypermutation. The results help further define the role that ecDNA plays across cancer and present a landmark for further studies.

Major comments

1. Samples with tumour purity < 0.1 were excluded. How was tumour purity determined? Supporting evidence is needed to justify this tumour purity threshold, in terms of amplicon and mutational signature detection sensitivity.
2. Genome sequencing coverage may be a second possible confounder for sensitivity and specificity of amplicon and mutational signature detection, and supporting evidence is needed to demonstrate that

the comparisons shown are not impacted by sequence coverage.

3. Some details on the patient cohort would be valuable for interpretation, for example to what extent were tumour newly diagnosed or pre-treated, if there were pre-treated tumours, how are those distributed across tumour types. What is the impact of any unevenness in pre-treatment status between tumour types on the downstream analyses?

4. In a subset of samples, two or more distinct ecDNAs were detected. Previous reports have demonstrated the presence of multiple oncogenes on single ecDNAs, i.e. EGFR-CDK4 combination in glioblastomas (PMID 29686388) or MYC-CDK6 in pediatric gliomas (PMID 30267146)? Evidence is needed to support the claim that in this analysis, the oncogenes are on distinct structures, such as the absence of sequence reads connecting the two loci. If the observations are restricted to distinct ecDNA pairs: does it matter if two oncogenes are on two distinct ecDNAs, or part of the same ecDNA molecule structure?

5. How were translocation ecDNAs detected? This is not described in the methods. It is intriguing that translocation ecDNAs were predominantly found in sarcomas, whereas the enrichment of two or more ecDNAs in the same sample was observed in CNS tumours.

6. Genes on ecDNAs were more likely to have oncogenic function compared to non-ecDNA amplifications, is this also true when copy number level is considered? I.e. does this observation remain true when binning by various levels of copy number?

7. Figure 1D appears to reflect the association between amplification and mutation frequency of the ecDNA cargo oncogene. The associated text does not reflect the analysis and appears to repeat the term 'amplification'.

8. In multi-sector analysis (Figure 1F), was the likelihood that an ecDNAs was subclonal (i.e. private to a single region rather than present in all sequenced regions) different compared to non-ecDNA amplifications?

9. The observation with respect to an enrichment for immunomodulatory genes on non-oncogene containing ecDNAs is interesting in light of the structural features (circularity) and possible association between ecDNA and micronuclei (i.e. per PMID 9508765). If this enrichment is indeed related to the unique behaviour of ecDNAs relative to other amplifications, it may be expected that non-ecDNA amplicons do not reflect this pattern, thus corroborating the observation. What gene classes do non-oncogene containing non-ecDNAs carry?

10. Figure 2E is implied to show T-cell fraction/depletion in immunomodulatory ecDNA samples, but the actual figure appears to reflect lncRNAs across ecDNAs which are described in the next paragraph. Fig 1F is referenced in that paragraph but this should have been 2E. The actual Figure 2E shows a particular enrichment for the RP11-89m16.1 lncRNA on ecDNAs across cancer but in particular in uterine cancers. Does this lncRNA have any known function and is there evidence for a role of this lncRNA in cancer, that could help explain the uterine cancer enrichment?

11. Extended Data Figure 4 shows genes that are co-altered in the presence of ecDNA and non-ecDNA amplifications. The text gives the impression that TP53 is significantly more frequently associated with ecDNA than non-ecDNA amplifications/no amplifications, but this is not obvious from the figure nor is there clear evidence in the form of a statistical test.

12. In the same figure, CDKN2A seems to proportionally stand out. This suggests that included in this figure are homozygous deletions, the most common type of CDKN2A deactivation, but this is not

mentioned. The next section (Fig 3A) aims to identify tumour suppressors associated with ecDNA in a tumour type independent manner. I was unable to find how this analysis was performed and more details are needed. CDKN2A homozygous loss is frequent in tumour types enriched for ecDNA such as glioblastoma and lung adenocarcinoma. It would be of interest to perform an analysis to detect associations between tumour suppressors and ecDNA that includes homozygous losses, controlling for tumour type, to determine whether CDKN2A is indeed significantly enriched.

13. Associations between ecDNA presence and tumour suppressor pathway losses as presented in Fig 3C should be tested for statistical significance. The X-axis in Fig 3C is not annotated. Tumour type acronyms should be written in full to accommodate a broad audience.

14. Is the analysis presented in Fig 4A corrected for tumour type imbalance, as was done for Fig 4B? Would this association between the presence of an ecDNA and increased mutational burden hold true when age is considered as a covariate?

15. SBS3 is also associated with response to platinum treatment. Is the observation that SBS3 is enriched on ecDNAs post-formation informed by tumours obtained from patients pre-treated with platinum therapy? Or is this enrichment already detected in untreated tumours, and if so, what kind of mechanism might that suggest?

16. It is not really clear what the relevance of the single example in Fig 5C is – with a cohort size of this magnitude, surely there are many interesting ecDNA amplifications in key driver genes, such as whether ERBB2 ecDNA status in BRCA is associated with HER2i treatment response. A justification on the significance of the specific example in 5C is needed.

17. The discussion is mostly a re-hashing of the results and , it would be of interest to get the authors' perspective on whether their results provide any insights into directions for ecDNA-specific therapy development.

Minor comments

1. Page 9, reference to Figure 1D should probably to Figure 2D? Also page 9, reference to Extended Figure 2A should be Extended Figure 3A?

2. Typo temozolamide -> temozolomide

Roel Verhaak

Referee #3 (Remarks to the Author):

The manuscript by Bailey et al describes computational analysis of ~15,000 cancer genomes from the Genomics England project. The analyses are what a team of excellent bioinformaticians would do with this large dataset. The authors identify likely ecDNAs from WGS and carrying out a variety of correlative analysis, including (i) estimation of frequency of samples with ecDNA across tumor types and subtypes, (ii) characterization of genes and functional elements encapsulated in ecDNAs, (iii) association with many factors including TMB, mutational signatures, MMRD, HRD, and survival.

Despite the extensive analyses, however, I'm unable to pinpoint what new insights have been uncovered about ecDNAs, given many excellent papers in this area including those by some of the

authors on this paper. Examples:

- Previous analysis of TCGA data (Kim et al, Nature Genetics, 2020) had estimated the frequencies of samples across 30+ tumor types using >3200 samples. The overall frequency of 17% in the present paper is similar to the 14% in the TCGA paper, as the authors note. The 5X increase in sample size allowed more accurate estimation of frequencies per tumor type as well as for subtypes, but I am not sure if this is a substantial advance.

- There are some difference in the list of genes found, but the top ones seem to be not much different from Kim et al. as far as I can tell (enrichment of oncogenes shown in Figure 1B is trivial at this point).

- Enrichment of immunomodulatory genes in ecDNA was already described by some of same the authors (Luebeck et al, Nature, 2023). A small number of ecDNAs contained lncRNA, enhancer, or promoter, but their role is not explored further.

Correlative analysis with other variables:

- I will take TMB as an example because this is the first item in the section, but I think some other analyses suffer from the same problem. The authors show that the median TMB is 4.34 for the samples with ecDNA and 2.24 for those with non-ecDNA focal amplification. First, as Figure 4 shows, there is a huge range for TMB and, with so many samples, even a small difference will give statistical significance (even for rank-based tests). Second, I'm not sure if it's possible to infer much from this difference because there are many other covariates that may have contributed to the difference. For instance, the range of TMB is very different across tumor types, and the sample size per tumor type is uneven across tumor types. I would think that this makes the conclusion dependent on the composition of the cohort, i.e., for a random subset of tumor types, sometimes it will be significant and other times it will not. Thus, one must control for the tumor type, i.e., by perform this comparison within each tumor type and see a consistent difference, I think. Also, do the focal amplifications with and without ecDNA differ in other characteristics such as size and amplitude? If they are not different, is it possible that other differences you see between the groups due to those (e.g., more oncogenes covered in non-ecDNA amplifications or ecDNAs have more copies of oncogenes) rather than the mechanistic aspect of ecDNA generation?

- Correlation with survival was also done in Kim et al and I think the findings are similar.

Overall, it is totally worthwhile to replicate their previous analysis on this much larger dataset, and the authors have done an extensive analysis. There are some new observations such as more extensive gene lists, timing of ecDNA formation with respect to some signatures, and some others. But the amount of new insights seems limited to this reader.

Minor points:

- Focal amplifications were defined as regions between 50kb and 20Mb in size. 20Mb seems to very

large, >40% of the smallest chromosome.

- The authors include foremost experts on ecDNAs including their computational detection. But how accurate are the ecDNA calls? If I understand correct, AmpliconArchitect is a graph-based method that relies on accurate segmentation of copy number, followed by construction of the possibly non-unique graph paths based on discordant reads before making an educated guess. Are there some estimates of false positives and false negatives (perhaps based on long-read data or other validation methods)? I understand that it is an inexact method, but I ask because the frequency estimates per tumor type are given down to 0.1% throughout the paper, and I wasn't sure whether the authors actually believe that they can estimate with that accuracy.

- The figures had tiny fonts and poor resolution. I could barely read the text within the figures even when viewing the pdf full screen on my desktop monitor.

Author Rebuttals to Initial Comments:

POINT-BY-POINT RESPONSES

Referee #1 (Remarks to the Author):

Bailey et al performed DNA sequencing analysis of 14,778 cancer patients from 39 tumour types to investigate the origins and impact of extrachromosomal DNA (ecDNA) in cancer. Using this comprehensive dataset, Bailey et al reveal multitude properties of ecDNA across many cancer types, convincingly showing their importance and contribution to cancer development. Specifically, this work reveals the following points: (1) ecDNA amplification are widespread across multiple cancer types, (2) ecDNA contain not only oncogenes but also immunomodulatory genes and lncRNAs, (3) ecDNA correlates with high tumor mutational burden, and with specific mutation signatures, (4) ecDNA correlate with metastatic disease, and (5) ecDNA correlate with poorer survival.

As ecDNA amplifications are increasingly recognized as major contributors to cancer, this work is of high importance, and provides critical information on their role. However, there are several concerns that need to be addressed as outlined below.

We thank the reviewer for the overall assessment of the work and for the immensely constructive comments and suggestions. We have addressed each one, as detailed in the point-by-point response below. As we hope will become clear, we addressed each comment through additional new analyses, and additional clarifications, as requested. We have embedded many of these new data within this revision letter, and indicate where they are included in the new MS.

1. There is very little information provided on the Genomics England dataset used.

We have revised both the main text and the figures to provide the much-needed data and explanations on: (1) the Genomics England dataset¹, (2) treatment history, (3) sequencing approaches used, and (4) computational methodology in both the main text and the methods, as follows:

Genomics England Dataset: For the dataset; in the methods we have added a section at the beginning of the introduction, titled “Genomics England v12 cohort.” As part of this section we elaborate on both the (1) dataset and (3) sequencing approaches:

“To map the frequency and contents of ecDNAs in human cancer, we analysed a cohort of 16,341 samples of whole genome–sequenced cancers from 15,223 patients recruited across 13 UK National Health Service (NHS) Genomic Medicine Centres as part of the Genomics England (GEL) 100,000 Genomes Project (100kGP) (GEL v12 data release). Samples with tumour purity of <10% were excluded, as were cancers of unknown primary, pediatric cancers and testicular germ cell tumours (510 samples). Within the cohort, 3.8% (598) of samples were fixed-formalin paraffin embedded (FFPE). Staging information was available for 10,780 (72.9%) of patients, with 836 (5.7%) patients recorded as having Stage 4 disease (Figure 1c). Median depth of coverage for tumour samples was 97.6x and for germline

samples was 32.6x. 1,800 (12.1%) patients were recorded as receiving systemic anti-cancer treatment (SACT) prior to biopsy.”

In the methods, we have added the following text:

“Genomics England is a company funded by the Department of Health & Social Care in the United Kingdom. The flagship project, known as the 100,000 Genomes Project, was established to sequence 100,000 whole genomes from NHS patients with rare diseases and cancer. In this study, we utilised version 12 of the cohort, comprising 14,778 participants. Sequencing libraries generated from tumour and matched germline DNA samples were sequenced using 150 bp paired-end reads on Illumina HiSeq platforms. In total 16,355 tumour and 16,555 germline samples underwent whole genome sequencing at a target depth of 100x for tumour and 30x for germline.”

Treatment effect on ecDNA: Does this include treated and untreated patients? If so, did the authors analyze such samples separately? Do treated patients have increased immunomodulatory ecDNAs, for example?

We appreciate this extremely insightful suggestion, and conducted the analysis in the revised manuscript.

The v12 Genomics England dataset has curated treatment information on the dataset. Of the 14,778 cancer patients in the study, 1,800 patients were recorded as receiving hormonal (n = 27), immunotherapy (n = 57), targeted (n = 415) or cytotoxic chemotherapy (n = 1653) prior to biopsy. The frequency of biopsies sampled from patients with prior systemic anti-cancer treatment, provided here for the reviewer, are shown below:

Reviewer Figure 1.1: Stacked bar plot showing the distribution of treatment type by cancer type

We then conducted the requested analysis by looking at the association between ecDNA and whether or not patients were treated. Due to low numbers, those patients who had received hormonal treatment were excluded.

In a logistic regression model, adjusted for age, stage, purity and tumour type, the detection of ecDNA was associated with prior chemotherapy (OR 2.38 95% CI 1.73-3.27) and targeted therapy (OR 2.87, 95% CI 1.12-6.43) (Figure 5c of the revised manuscript).

Figure 5c: Forest plot demonstrating the association between treatment and ecDNA detection. This model adjusted for cancer type, age, sex and purity.

These data provide strong evidence that systemic cancer treatment can increase ecDNA levels, which, in line with our previous work, suggests that it is involved in treatment resistance. This could potentially arise as subclones existing at low levels that contain ecDNA are selected during treatment, or it is also possible that treatment itself may be involved in the de novo formation of ecDNA. Future studies will be needed to confirm the mechanistic basis. We believe that the reviewer's suggested analysis has revealed critical new information that is highlighted now in Figure 5c of the revised manuscript.

To drill down deeper on whether systemic treatment could potentially impact immunomodulatory ecDNA levels, we analyzed the association between the ecDNA detection and receiving systemic anti-

cancer treatment. In a logistic regression model adjusted for tumour type, age, sex and purity, the odds of detecting immunomodulatory ecDNA was 2.2x higher in patients who had received prior treatment (OR 2.2, 95% CI 1.72 - 2.82) compared with oncogenic ecDNA (OR 1.41, 95% CI 1.24-1.61), non-oncogenic (non-immunomodulatory) ecDNA (OR 1.34, 95% CI 1.20-1.52) and regulatory (no gene) ecDNA (OR 1.25, 95% CI 1.10-1.42).

Reviewer Figure 1.2: Forest plot showing results of a logistic regression model that determines the odds of receiving treatment given the sample has ecDNA

It will be important to determine whether immunomodulatory ecDNAs will increase in copy number in patients treated with immunotherapy, but, the reviewer will appreciate from the figure shown above regarding treatment breakdown by tumour type, there are not enough patients treated with immunotherapy in the sample set for a fruitful analysis.

What sequencing approach was used (coverage etc.)?

Thank you for suggesting that we clarify. In the methods section of the revised manuscript, we state:

“Sequencing libraries generated from tumour and matched germline DNA samples were sequenced using 150 bp paired-end reads on Illumina HiSeq platforms. In total 16,355 tumour and 16,555 germline samples underwent whole genome sequencing at a target depth of 100× for tumour and 30× for germline”.

In the current form, this dataset is a black box. The computational methodology used to infer ecDNA should also be discussed, and the degree of confidence of the approach used needs to be illustrated.

Can the authors reconstruct ecDNA structures (even complex ones) with high degree of confidence – and can they provide examples

The reviewer’s thoughtful comment motivated us to provide a clearer presentation of the methodology of AmpliconArchitect, including a high-level graphic of how it works. Please see Panel A of Figure 1 of the revised manuscript, also shown below.

Figure 1a: Schematic demonstrating the analysis pipeline applied to the GEL cohort

Each tumour sample is subjected to whole genome sequencing, then aligned to the reference genome (hg38). This sequencing data is then analysed with CNVkit to detect areas of the genome that are amplified with high copy number. The sequencing data corresponding to these areas of amplification are then passed to as seed intervals to AmpliconArchitect which then defines a copy number and structural variant aware amplicon graph that can be decomposed into linear paths and cycles. The output from AmpliconArchitect is then passed to AmpliconClassifier which, after filtering, classifies areas of the amplicon as either breakage-fusion-bridge, ecDNA, complex non-cyclic focal amplification, or a linear focal amplification as well as performing amplicon complexity scoring.

We note that this technology is now widely used and widely adopted by many labs across the world. As shown below, and highlighted in our paper, Kim et al.², and now improved in a manuscript currently under review at Nature Methods (NMEMH-BC55179A), AmpliconArchitect and AmpliconClassifier are powerful and accurate tools for ecDNA detection from WGS data. In this figure presented below for the reviewer, each point represents the number of ecDNA detected using metaphase FISH for an amplicon detected with sequencing. In total, results for 67 genes from 42 cytogenetically validated cell lines are shown. Each cell line was subjected to whole genome sequencing then analysed with AmpliconArchitect and AmpliconClassifier (AC). AC assigns the amplicons from each cell line to one of four categories: linear, complex non-cyclic, breakage fusion bridge, or ecDNA.

Reviewer Figure 1.3: AmpliconArchitect and AmpliconClassifier performance on cytogenetically validated cell lines using metaphase FISH.

Each point represents the number of ecDNA detected using metaphase FISH for an amplicon detected with sequencing. In total, results for 67 genes from 42 cytogenetically validated cell lines are shown. Each cell line is subjected to next generation sequencing then analysed with AmpliconArchitect and AmpliconClassifier (AC). AC then assigns the amplicons from each cell line to one of four categories: linear, complex non-cyclic, breakage fusion bridge, or ecDNA. Please note that this figure is from another manuscript under review. We provide these data for the reviewer and note that an earlier version of this plot and its source data is in Kim et al. 2020 “Extrachromosomal DNA is associated with oncogene amplification and poor outcome across multiple cancers” Nature Genetics.

We also present another figure below that is now Extended Data Fig. 1a in our revised manuscript, that summarises the analytic process of AmpliconArchitect and AmpliconClassifier in a cartoon form:

Extended Data Fig. 1a: Schematic of AmpliconArchitect and AmpliconClassifier analysis

Schematic further demonstrating the analysis pipeline applied to the GEL cohort. Each tumour sample is subjected to whole genome sequencing, then aligned to the reference genome (hg38). This sequencing data is then analysed with CNVkit to detect areas of the genome that are amplified with high copy number. The sequencing data corresponding to these areas of amplification are then passed to AmpliconArchitect which then defines a copy number and structural variant aware amplicon graph that can be decomposed into linear paths and cycles. The output from AmpliconArchitect is then passed to AmpliconClassifier which, after filtering, classifies areas of the amplicon as either breakage-fusion-bridge, ecDNA, complex non-cyclic focal amplification, or a linear focal amplification as well as performing amplicon complexity scoring.

Further, we provide orthogonal confirmation of AmpliconArchitect and Amplicon/Classifier outputs, in the revised manuscript, including on select examples for which we could get tissue for FISH analysis, as shown in Figure 1b of the revised manuscript, shown below.

Figure 1b: Representative FISH images and AmpliconArchitect structural variant views from two GEL cases

The first image (left) demonstrates an amplicon predicted to be a chromosomal amplification consistent with its FISH image and the second image (right) shows an amplicon predicted to be an ecDNA consistent with its FISH image.

In the revised manuscript, we also provide considerably more detail on additional tools, such as the amplicon complexity score, including with graphic examples.

2. In continuation to the first point, the methods section could be more informative and detailed. Figure legends and Figures could also include more detailed information, better explaining each feature within the presented data (meaning of colors, for example in Figure 3, but also in other figures). Better explanation of different bioinformatic parameters would make the paper more approachable to non-computational audiences – for example, explanation of dn/ds, SBS, amplicon entropy/complexity.

We appreciate the reviewer's suggestion and have addressed these important concerns regarding clarity. Figures 1 and 2, as well as their figure legends have been completely redone and greatly improved including providing clear color keys. To address the additional salient comments about Figure 3, and the need for better explanation of these bioinformatic metrics and clearer presentation, please see the heavily reworked and strengthened Figure 3 of the revised manuscript, shown below. Please note, we have also rewritten all legends in the manuscript to provide more detailed descriptions of their corresponding figures. Beneath the figure presented below, we also discuss each of the bioinformatic analyses and how they were done, as well as how they are presented much more clearly in the revised manuscript.

Figure 3 - Correlates of genome instability and ecDNA: a, Forest plot showing results of a regression model that investigates associations between tumour suppressor losses and the presence of ecDNA or chromosomal amplifications across the entire cohort controlling for cancer type. Associations with ecDNA are represented by circles and chromosomal amplifications represented by diamonds. b, Bar plot showing the proportion of tumours across the cohort with any ecDNA (blue), *MDM2* ecDNA (yellow) or no ecDNA (grey) grouped by *TP53* mutation status. c, Body map with panels for selected cancer types. Each panel contains (top) a forest plot showing associations between ecDNA presence or absence with tumour suppressor

losses. (bottom left) A forest plot showing associations between ecDNA presence of absence with weighted genome instability index, structural variant burden, and whole genome duplication. (bottom right) A violin plot demonstrating the amplicon complexity scores for the tumours of that cancer type.

1) **dN/dS** - we have provided the following brief explanation regarding dN/dS analysis in the main text of the manuscript where the concept is first introduced:

Line number 154-155: "...a mutation-based signal of positive selection as derived from the adjusted ratio of non-synonymous to synonymous mutations (dN/dS see Methods, Extended Data Fig. 7f)."

We have added a methods section titled "Estimates of selection using dN/dS" that we have reproduced at the bottom of the response to this reviewer comment. In this new methods section we cite Martincorena et al. 2017 "Universal Patterns of Selection in Cancer and Somatic Tissues" that explains the dN/dS concept in detail as well as the computational method we employ³.

"The dN/dS point mutation estimate was calculated using the dNdScv package that was run on all mutations available in the cohort. This method uses a maximum likelihood approach in its analysis of the ratio of non-synonymous (missense, nonsense, and essential splice mutations) to synonymous substitutions to infer a measure of the strength of selection acting on protein-coding genes. It also estimates a background mutation rate of each gene through joint analysis of both local and global information that takes into account sequence composition and the contribution of mutational signatures. In our analysis, estimation of dN/dS ratios was performed at multiple levels including at for individual genes, across the genome, as well as stratified by context including ecDNA amplification, non-ecDNA amplification, and non-amplified areas of the genome."

We have also presented the data differently in panel I of revised Figure 1. By showing it right below the bar chart of mutation frequency in oncogenes on ecDNA, this new format helps readers link how coding mutations interact with ecDNA selection.

2) **SBS** - we have added the following line to the main text introducing the concept of single-base substitution signatures:

Line number 260-261: "...single-base substitution (SBS) signatures that utilise the mutated bases and the bases immediately 5' and 3' to infer mutational processes..."

We have also added a methods section describing the use of this technique in the manuscript:

"Degasperi et al. quantified the fraction of single base substitutions found in each of the 96 trinucleotide contexts from the multiple WGS cohorts (including GEL) and analysed these data with non-negative matrix factorization to infer a set of SBS signatures. We then used this reference set of SBS signatures to infer the most likely SBS signature for mutations in our cohort. Using the sample-level single base substitution exposures and the SBS reference signatures, each trinucleotide channel context is assigned

a likelihood value by multiplying the sample exposure weight by the reference signature weight. This allows estimation of the most likely mutational process for each mutation.”

3) **Amplicon complexity** - our tools AmpliconArchitect and AmpliconClassifier enable us to examine ecDNA structure. The key features that go into determining how complex to amplicons from the same patient are:

a) The number of segments that are participating in the ecDNA.

b) The number of cycles that can be used to explain the copy number of the focal amplification on ecDNA.

We developed this computational tool and complexity scoring and published it in Luebeck et al. 2023⁴.

In the revised manuscript we now provide the following sentence in the text to explain what goes into the complexity score and cite Luebeck et al. 2023.

Line number 210-211: *“We next measured the ‘amplicon complexity’ of enhancer only ecDNA, which quantifies the number of segments and the diversity of structure decompositions inferred by AmpliconArchitect (see Methods, Extended Data Fig. 1b)...”*

We provide further explanation in the Methods (reproduced at the bottom of this reviewer comment response) as a graphical explanation of the amplicon complexity score in Extended Data Fig. 1b of the revised manuscript.

Finally, in order to clarify our language we now only refer to “amplicon complexity” rather than “amplicon entropy” and “amplicon complexity”. Our use of both terms in the last version of our manuscript referred to the measure of amplicon complexity described in Luebeck et al. 2023. We apologise for any confusion caused by the use of the term “amplicon entropy” which has been removed from the revised manuscript.

“The amplicon complexity score, as defined in Luebeck et al 2023, is calculated by AmpliconClassifier. For each seed-interval defined amplicon, AmpliconArchitect produces a copy-number-aware breakpoint graph (CNA-breakpoint graph). AmpliconArchitect also outputs decompositions that represent cyclic and non-cyclic paths through this CNA-breakpoint graph. These decompositions are passed as input to AmpliconClassifier to produce the complexity score which aims to capture the diversity of the cyclic and/or non-cyclic paths present. Each path has a copy number and a length in kilobase pairs which are combined to create a length-weighted copy number (normalised by the total length-weighted copy number present in the CNA-breakpoint graph) and the complexity score is calculated through the sum of three log-transformed measures:

1)The total count of copy number segments present in the amplicon

2)The normalised length-weighted copy number of each cyclic path

3)The residual normalised length-weighted copy number that is not not explained by cyclic paths”

Extended Data Fig. 1b: Schematic illustrating amplicon complexity scoring.

(left) Low complexity amplicon - showing a simple graph that consists of a single segment with one cycle that explains all the amplicon copy number. (middle) Higher complexity amplicon showing a more complex graph that consists of three segments whose single cycle is only able to explain the majority of the amplicon copy number. (right) High complexity amplicon showing a graph that consists of 4 segments that can only explain the majority of the amplicon copy number with two cycles.

3. The paper title emphasizes this work studies the origin of ecDNA. There are several aspects that could be improved with this regard:

a. Analysis of the chromosomes of origin of ecDNA would be very informative on the mechanisms of ecDNA formation. Are there any genomic scars detected?

AmpliconArchitect does provide detailed information about the chromosomal origin of the content of the ecDNAs that are found. The amplified regions are mapped to the reference genome and therefore the chromosomal origin of the ecDNA is revealed. We can provide a plot that explores the distribution of immunomodulatory ecDNA across the genome (Extended data figure 3) and can do the same for all ecDNA if the reviewer felt that would be useful.

Regarding the reviewer’s question about mechanisms of ecDNA formation and whether finding genomic scars could provide insight, we appreciate that understanding the processes of how ecDNA forms is important and is currently understudied. It is known that ecDNA can arise from chromothripsis, also potentially from translocation-bridge-amplification, and there is also strong evidence that paired double strand breaks can lead to ecDNA formation. In fact we have a manuscript under review at Cancer Discovery examining this question titled “Disparate pathways for extrachromosomal DNA biogenesis and genomic DNA repair” that we have also uploaded as a preprint⁵.

Extensive further work will be needed to understand these mechanisms of formation and further to determine how they arose in individual patients.

3a continued: is there enrichment for structural variations (SVs)? Such analysis should consider the fact that in some cases the chromosome of origin is lost (for example through chromothripsis).

This is a great question. The ecDNA context actually provides such information. We know that in some cases the context is very simple; appearing as a simple excision from a single chromosome.

Reviewer Figure 1.4: Simple ecDNA on a simple background

In other cases, such as in those that may arise from chromothripsis, we see a highly complex, heavily rearranged ecDNA which can contain contributions from multiple chromosomes, as shown below.

Reviewer Figure 1.5: Heavily rearranged ecDNA on a complex uni-chromosomal background

Reviewer Figure 1.6: Heavily rearranged ecDNA on a complex multi-chromosomal background

Future analyses will be required to better understand the clinical implications of such differences. We provide these figures for the reviewer. We have not added them to the manuscript, however if the reviewer felt strongly, we could consider putting them in the Extended Data Figures.

b. The authors use the term Amplicon Entropy (which is not well defined in the methods section) to examine ecDNA structure complexity. However, the authors do not provide specific details that would be informative. ecDNA formed through chromothripsis would be expected to include discordant DNA fragments at higher frequency (fragments from distant chromosome loci), and also greater SV complexity (but this could emerge later during the lifetime of the ecDNA).

To address this important point, we have provided a more detailed description of amplicon complexity. In the revised manuscript we now provide the following sentence in the main text to briefly describe complexity score and cite Luebeck et al. 2023 in which the measure is defined in detail.⁴

Line number 238-239: "...the amplicon complexity, a measure based on the number of segments and diversity of structural decompositions inferred by AmpliconArchitect (see Methods), of these..."

We then provide a detailed explanation in the Methods (see below) as well as further detail and a graphical explanation in the appendix of the revised manuscript.

"Amplicon complexity

The amplicon complexity score, as defined in Luebeck et al 2023, is calculated by AmpliconClassifier. For each seed-interval defined amplicon, AA produces a copy-number-aware breakpoint graph (CNA-breakpoint graph). AA also outputs decompositions that represent cyclic and non-cyclic paths through this CNA-breakpoint graph. These decompositions are passed as input to AC to produce the complexity score which aims to capture the diversity of the cyclic and/or non-cyclic paths present. Each path has a copy number and a length in kilobase pairs which are combined to create a length-weighted copy

number (normalised by the total length-weighted copy number present in the CNA-breakpoint graph) and the complexity score is calculated through the sum of three log-transformed measures:

- 1)The total count of copy number segments present in the amplicon
- 2)The normalised length-weighted copy number of each cyclic path
- 3)The residual normalised length-weighted copy number that is not explained by cyclic paths”

Extended Data Fig. 1b: Schematic illustrating amplicon complexity scoring.

(left) Low complexity amplicon - showing a simple graph that consists of a single segment with one cycle that explains all the amplicon copy number. (middle) Higher complexity amplicon showing a more complex graph that consists of three segments whose single cycle is only able to explain the majority of the amplicon copy number. (right) High complexity amplicon showing a graph that consists of 4 segments that can only explain the majority of the amplicon copy number with two cycles.

Finally, in order to clarify our language we now only refer to “amplicon complexity” rather than “amplicon entropy” and “amplicon complexity”. Our use of both terms in the last version of our manuscript referred to the measure of amplicon complexity. We apologise for any confusion caused by the use of the term “amplicon entropy” which has been removed from the revised manuscript.

In addition, we have reproduced worked examples of amplicon complexity from Luebeck et al. 2023 for the reviewer below. If the reviewer feels it would be helpful we could consider creating a figure stepping through new examples and include them in the appendix of our manuscript.

Number of CN segments in ecDNA region: 1
AA genome paths (segments and length-scaled weight):
 Two paths overlapping ecDNA region:
 - Path 1: cyclic, weight=0.52
 - Path 2: non-cyclic, weight=0.48

Low complexity case: score 0.69

Complexity from CN segments = $\log(1) = 0$

Complexity of non-ecDNA-like paths
 = $-0.48 \times \log(0.48) = 0.34$

Complexity of ecDNA-like paths
 = $-0.52 \times \log(0.52) = 0.35$

Total complexity score:
 = complexity from number of segments +
 ecDNA complexity + residual complexity +
 = $0 + 0.34 + 0.35$
 = **0.69**

Number of CN segments in ecDNA region: 10
AA genome paths (segments and length-scaled weight):
 Eight paths overlapping ecDNA regions:
 - Two cyclic paths, both in top 80% of weights: 0.21, 0.07
 - Six non-cyclic paths, combined weight = 0.72

Medium complexity case: score 3.05

Complexity from CN segments = $\log(10) = 2.30$

Complexity of non-ecDNA-like paths
 = $-0.72 \times \log(0.72) = 0.24$

Complexity of ecDNA-like paths
 = $-0.21 \times \log(0.21) + -0.07 \times \log(0.07) = 0.51$

Total complexity score:
 = $2.30 + 0.24 + 0.51$
 = **3.05**

Number of CN segments in ecDNA region: 84
AA genome paths (segments and length-scaled weight):
 Twenty-four paths overlapping ecDNA regions:
 - Eleven cyclic paths in top 80% of all weights:
 $D = [0.23, 0.14, 0.077, \dots, 0.023, 0.014, 0.012]$
 - Thirteen non-cyclic paths & cyclic paths in bottom 20%
 of weights, combined weight = 0.30

High complexity case: score 6.44

Complexity from CN segments = $\log(84) = 4.43$

Complexity of non-ecDNA-like paths
 = $-0.30 \times \log(0.30) = 0.36$

Complexity of ecDNA-like paths
 = $-\sum D_i \times \log(D_i) = 1.65$

Total complexity score:
 = $4.43 + 0.36 + 1.65$
 = **6.44**

Reviewer figure 1.7: Worked examples of the amplicon complexity score. Reproduced from Luebeck et al 2023⁴.

To address the reviewer's second point as to whether chromothripsis would be expected to include discordant DNA fragments at a higher frequency and a higher SV complexity, our data suggest that the reviewer is exactly right. As shown in the response to the reviewer above, certain amplicons appear to be very simple and arise from a single chromosome. In other contexts there is a lot of structural variation within the ecDNA and in neighboring parts of the genome. In Figure 3, we see that in the case of sarcoma samples, highly prevalent for chromothripsis⁶, that the distribution of amplicon complexity favours highly complex complex amplicons with multi-chromosomal rearrangements.

Figure 3c - Sarcoma panel

Panel contains (top) a forest plot showing associations between ecDNA presence or absence with tumour suppressor losses. (bottom left) A forest plot showing associations between ecDNA presence or absence with weighted genome instability index, structural variant burden, and whole genome duplication. (bottom right) A violin plot demonstrating the amplicon complexity scores for the tumours of that cancer type.

c. Analysis of the breakpoint junctions within ecDNA could be informative on the repair mechanisms operative in the formation and maintenance of amplifications.

We thank the reviewer and agree that analysing breakpoint junctions could be informative for helping to decipher the DDR mechanisms that may be involved in the formation, maintenance, and evolution of ecDNAs. Due to bulk whole genome sequencing we noted the number of small insertions and deletions *within* ecDNA that we could detect were limited. As a consequence we are unable to perform the analysis that we could with single base substitutions (though we do believe that looking at indel

signatures within ecDNA would have been a valuable analysis). We were however able to stratify the structural variants (SVs) detected within the ecDNAs. As reflected in the analysis of amplicon complexity (Figure 3) we found that the structural variant burden within ecDNA was highest in Upper GI and sarcoma samples (Reviewer Figure 8).

Figure 3c - Sarcoma and upper gastrointestinal adenocarcinoma panels

Panels contain (top) a forest plot showing associations between ecDNA presence or absence with tumour suppressor losses. (bottom left) A forest plot showing associations between ecDNA presence or absence with weighted genome instability index, structural variant burden, and whole genome duplication. (bottom right) A violin plot demonstrating the amplicon complexity scores for the tumours of that cancer type.

Reviewer Figure 1.8 Structural variant burden of ecDNA according to cancer type

We then plotted the distribution of foldback breakpoints and inter-chromosomal breakpoints per cancer type. The translocation-bridge amplifications in breast cancer and chromothripsis highly prevalent in sarcomas may explain the increase in inter-chromosomal breakpoints. Foldback breakpoints are a hallmark of breakage-fusion-bridge cycles (BFB), and the enrichment in Upper GI and oropharyngeal cancer types hint at BFBs driving ecDNA formation in these cancer types.

Reviewer Figure 1.9: Distribution of ecDNA foldback and interchromosomal breakpoints according to cancer type

We begin to infer the repair mechanisms operative in ecDNA maintenance in Figure 4f,g. By quantifying the mutational signatures that tend to occur later in ecDNA evolution we see that SBS3 (homologous recombination deficiency - HRD) occurs after the formation of ecDNA, suggesting that HRD is not necessary for ecDNA maintenance.

To understand the repair mechanisms operative in ecDNA formation, we have developed a model system, CRISPR-C, for developing ecDNA de novo. This allows us to analyse the breakpoint junctions in the model system that begins to address some of the reviewer's important points. We thank the reviewer and point them to this manuscript currently under review and in preprint: Rose et al.⁵

“Disparate pathways for extrachromosomal DNA biogenesis and genomic DNA repair” -
<https://www.biorxiv.org/content/10.1101/2023.10.22.563489v1>

d. Line 105: “..two or more oncogenes on distinct ecDNAs”. What about the frequency of two or more oncogenes on the same ecDNA (such as in cases of *MDM2* and *CDK4*)?

To answer this point we first calculated the proportion of samples with ecDNA that had more than one oncogene on the same ecDNA. This worked out at 46% (1,167 of 2,532 samples). We then constructed a heatmap of genes to demonstrate that the co-amplification of oncogenes is heavily dependent on proximity.

For the example of *MDM2*; the co-amplification with *CDK4* occurs in 51 out of 207 ecDNA (24.6%). We then determined the ‘promiscuity’ of oncogenes by establishing the proportions of co-amplifications that are inter-chromosomal. *DDIT3*, *ERBB2*, and *CCND1* had the highest proportion of ecDNA with a non-native chromosomal oncogene partner, compared with *MYC* and *EGFR*, which were rarely found with other oncogenes (Extended data Figure 2 of the revised manuscript) .

Extended Data Fig. 6b: heatmap demonstrating the co-amplification of oncogenes on ecDNA

(left) Schematic demonstrating an ecDNA containing two or more oncogenes. (right) Heatmap showing how often pairs of oncogenes are found on the same ecDNA. Blue outlined boxes indicate potential oncogene pairs consisting of oncogenes from the same chromosome. The corresponding chromosome is labeled to the left of the blue outlined box.

Extended Data Fig. 6d: The tendency of oncogenes to pair with other oncogenes

Stacked bar plot showing the proportion of oncogenes observed on ecDNA in ≥ 20 tumours that were found in the context of an ecDNA containing that single oncogene (yellow), in a pair of oncogenes from the same chromosome (orange), or in a pair of oncogenes from different chromosomes (brown).

e. Mutations in ecDNA (Figure 1D-E): Do the mutations correlate with CN? Are the mutations found in the ecDNA or in the endogenous chromosomal region?

To understand whether mutations correlated with CN and whether or not they are found on the ecDNA or endogenous locus, we calculated a mutational multiplicity, defined as the copy number state of an SNV within a predicted ecDNA locus. Given the variant allele frequency of the mutation, tumour purity, and focal amplification copy number), the mutational multiplicity can be determined:

$$[CPN]^{mut} = VAF * (1/p) * (p * [CPN]^{focal} + [CPN]^{norm} * (1 - p))$$

With the copy number state of the SNV we can then determine if the SNV is present on ecDNA (as it is the ecDNA that is highly amplified). If the SNV copy number is higher than the ploidy state of the tumour (2 in diploid tumours), then we can determine that the SNV is present on ecDNA. We then determined whether the SNV is present on all ecDNA copies (early) or few ecDNA copies (late).

Figure 4f: Schematic illustrating method of timing mutations in areas of the genome amplified by ecDNA.

This schematic is present in Figure 4 and we have also added this as it pertains to mutations found on ecDNA in genes that are commonly amplified on ecDNA. In addition, we have modified Figure 1 to include a barplot showing the timing of mutations found on ecDNA and used for dN/dS analysis. We have reproduced Figure 1h below.

Figure 1h: ecDNA mutations and their timing

Stacked bar plots displaying (top) the proportion of types of non-synonymous mutations observed in the oncogenes present on ecDNA and (bottom) the proportion of these non-synonymous mutations in different timing categories (see Methods). Only the mutations affecting the 20 oncogenes most commonly present on ecDNA are shown.

f. Is there selection for non-coding regulatory regions on ecDNA (as seen in PMID: 31761532)?

We did not see an increase of non-coding elements on ecDNA compared with non-ecDNA focal amplifications, however we did see an enrichment of non-coding regulatory regions on regulatory ecDNA compared with oncogenic and non-oncogenic ecDNA. We calculated this by annotating distal enhancers and promoters on ecDNA and calculating the number of non-coding elements seen per Mb.

Figure 2I: Enrichment for non-coding elements on regulatory ecDNA

Boxplots showing the number of different categories of non-coding elements per megabase in ecDNA containing only regulatory elements or those ecDNA containing genes.

4. Lines 120-12 and Figure 1C: "... genes recurrently encoded by ecDNA were more likely to be oncogenes compared with non-ecDNA driven focal amplifications". Could oncogene likelihood be correlated with amplicon size rather than location (ecDNA/chromosomal amplification)? Actually, could amplicon size be used to predict the evolutionary stage of amplification? For example, larger amplicons might be newer and through selection and structural variation of amplifications become smaller in size (to contain only the necessary genes). Also, what is the relationship between ecDNA and chromosomal amplifications – can the authors drill down the mechanisms to identify potential break-fusion-bridge (BFB) events and ectopic integration of ecDNAs?

We thank the reviewer for this comment and have answered their comment in several parts:

1. Lines 120-12 and Figure 1C: "... genes recurrently encoded by ecDNA were more likely to be oncogenes compared with non-ecDNA driven focal amplifications. Could oncogene likelihood be correlated with amplicon size rather than location (ecDNA/chromosomal amplification)?"

To address this point we binned focal amplifications and compared a) the number of oncogenes per amplification, b) the number of tumour suppressor genes per amplification and c) the number of genes per amplification. Comparing ecDNA with chromosomal amplifications we found that there was a significant increase in the mean number of oncogenes per amplification across all amplification sizes,

but the comparison of tumour suppressor genes and all genes were non-significant across all amplification sizes (Kruskal-Wallis test). The results of this are highlighted in Extended Data Fig. 7c.

Extended Data Fig. 7c: Comparison of oncogene density according to size and stratified by amplification type
 Bar plots showing (top) the mean number of oncogenes, (middle) the mean number of tumour suppressor genes, (bottom) the total number of genes for chromosomal amplifications and ecDNA grouped by size. Bars that are red signify ecDNA and blue signifies chromosomal amplifications.

2. “Actually, could amplicon size be used to predict the evolutionary stage of amplification?”

This was a very interesting point raised by the reviewer. To investigate this we examined the area of each amplicon designated as ecDNA by AmpliconClassifier and then tested whether different tumour stages demonstrated significantly different distributions of ecDNA lengths or oncogene densities. We found that when we compared oncogenic ecDNA we found that the ecDNA in stage 4 tumours tended to be smaller than limited stage tumours (Reviewer Figure 1.10), however when subsetting by oncogene containing ecDNA only this difference in size was non-significant (Reviewer Figure 1.11).

Reviewer Figure 1.10: The distribution of ecDNA size according to stage (all ecDNA)

Reviewer Figure 1.11: The distribution of ecDNA size according to stage (oncogenic ecDNA)

Reviewer Figure 1.12: Comparing number of oncogenes per Mb according to stage

Interestingly, when we took oncogenic ecDNA alone we found a marginal increase in the number of oncogenes per Mb at later cancer stages (p values represent t tests vs the other groups in the analysis). Despite the associations between smaller size, oncogene density and late-stage disease, in terms of alterations to ecDNA structure and whether such a characterisation could be used to explore the evolutionary stage of the amplicon we are limited by the single timepoint and short-read nature of our data. Longitudinal datasets are needed to understand what happens to the same ecDNA in a given sample.

3. Also, what is the relationship between ecDNA and chromosomal amplifications – can the authors drill down the mechanisms to identify potential break-fusion-bridge (BFB) events and ectopic integration of ecDNAs?

We thank the reviewer for this comment, as part of our computational pipeline to identify ecDNA we run both AmpliconArchitect and then AmpliconClassifier. As well as identifying ecDNA, AmpliconClassifier also predicts non-ecDNA amplifications and further classified them as breakage-fusion-bridges, complex, and linear amplifications. Identifying integration points with short reads alone is technically extremely challenging and, to our knowledge, there is no existing computational method for this task.

Approaches that have identified the occurrence of integration have involved observations of cultured cells losing ecDNA and gaining homogeneously staining regions, others have attempted to describe integration points is to use long read optical genome mapping to reveal integration points that were found to be in highly repetitive regions not amenable to short read sequencing.

In Koche et al. 2020, circle-seq, nanopore long read, visual inspection of split reads, allele-specific PCR and Sanger sequencing enabled them to describe integration sites. These integration points are often close to repetitive regions.^{7,8}

Furthermore, in Turner et al 2017 it is shown that re-integration is very heterogeneous event, with multiple different integration points for an HSR confirmed using FISH.⁹

Additional modalities will be needed to reliably detect integration events and obtaining extra tissue within Genomics England has proven extremely challenging.

5. Figure 2A shows the distribution of ecDNA CN and size. The distributions appear to be very large. Although ecDNA heterogeneity is expected, to what extent does the confidence of ecDNA calling algorithm affects this result? Would more stringent computational analysis reduce this heterogeneity? Also, the scale of this analysis (pan-cancer) is not very informative on its own. Could this analysis be extended to show CN and sizes per cancer type, and also per amplified genes (would this distribution be similar between MYC and other genes, for example)?

The reviewer has raised a very important point. As we demonstrated in Lange et al. Nature Genetics 2022 “The evolutionary dynamics of extrachromosomal DNA in human cancers”, the random segregation that occurs in cell division by definition creates the heterogeneity, it is not a function of our computational analysis but is a product of evolution as was rigorously measured in that paper.¹⁰ We have reproduced two figures from Lange et al. 2022 (originally Figure 2c, d, and 2e) in order to help demonstrate this point¹⁰. The analysis combined individual-based stochastic computer simulations of growing cell populations assuming random ecDNA segregation and tumor tissue from two cancer types, glioblastoma (GBM) and neuroblastoma (NB).

Reviewer Figure 1.13: ecDNA simulation and tumour tissue data from Lange et al, Nature Genetics 2022

Schematic showing how data from simulations and interphase FISH data from tumour samples were combined. - Reproduced from Lange et al. Nature Genetics 2022

In both of these patients the amplified oncogene was present on ecDNA and the ecDNA copy number distributions that were quantified showed extreme cell-to-cell variation matching the distributions predicted by simulations. Consequently this level of heterogeneity in copy number is very much inline with what is predicted by random segregation and selection for ecDNA.

Reviewer Figure 1.14: ecDNA copy number distributions in patient samples.

Dots signify ecDNA copy number distribution in six patients with glioblastoma (right) and four patients with neuroblastoma emerge from the same process of random ecDNA segregation (black dashed line). - Reproduced from Lange et al. Nature Genetics 2022

In the same paper we carried out work that suggests that ecDNA copy number is a function of the strength of selective pressure. ecDNAs containing the dihydrofolate reductase gene were generated in a cancer cell line and a strong, dose-dependent rise in ecDNA copy number in response to methotrexate treatment was observed.^{10,11}

Experimental evidence from another previously published study¹² (Hung et al. 2022 “Targeted profiling of human extrachromosomal DNA by CRISPR-CATCH”) shows that heterogeneity in ecDNA size distribution is a function of the underlying biology rather simply a result of computational analysis. This paper describes a method to specifically isolate ecDNAs, when applied to a human melanoma tumour sample a range of ecDNA sizes were identified.

Reviewer Figure 1.15: Tumor processing and ecDNA enrichment from patient tumor samples using CRISPR-CATCH

(Left) A schematic for the tumor processing and electrodepletion protocol for preparing tumor DNA for CRISPR-CATCH and normalized short-read sequencing coverage of the expected NRAS ecDNA in melanoma patient tumor (Pt9) after CRISPR-CATCH. (Right) A PFGE image from melanoma patient sample Pt9 after electrodepletion and CRISPR-CATCH using NRAS-targeting guide 194 (guide sequence in Supplementary Table 1). Brackets on the right correspond to gel-extracted regions shown in Fig. 2c. One independent experiment was performed. Figures reproduced from Main Figure 2b,c and Extended Data Fig. 3d from Hung et al. 2022 "Targeted profiling of human extrachromosomal DNA by CRISPR-CATCH"¹².

As suggested by the reviewer we will add gene and tumour specific distributions of copy number and ecDNA size to the supplementary data (Extended Data Fig. 3b-e of the revised manuscript), shown below.

Extended Data Fig. 3b: Distribution of ecDNA copy number according to tissue type.
 Box plots showing the copy number of ecDNA grouped by tissue type.

Extended Data Fig. 3c: Distribution of ecDNA size according to tissue type.
 Box plots showing the size in megabases of ecDNA grouped by tissue type.

Extended Data Fig. 3d: Distribution of ecDNA copy number according to cancer type and oncogene
 Box plots of log-transformed copy number of ecDNA grouped by the oncogenes present on the ecDNA and tissue type

Extended Data Fig. 3e: Distribution of ecDNA size according to tumour type and oncogene
 Box plots of ecDNA size in megabases grouped by the oncogenes present on the ecDNA and tissue type.

6. Several findings in this work were demonstrated, to some extent, before, in some cases by co-authors of the present study. The authors are encouraged to discuss them in proper context:

- a. p53 mutations and MDM2 amplifications are mutually exclusive (PMID: 27194168). Related to Figure 3C.
- b. The extent of ecDNA amplification across tumor types and different genes (PMID: 32807987).
- c. Correlation of ecDNA with poor survival (PMID: 32807987). Related to Figure 5D.
- d. Relationship of ecDNA and clustered mutations (PMID: 33361815, PMID: 35140399).
- e. Presence of immunomodulatory genes in ecDNA (PMID: 37046089).

We thank the reviewer for this comment and we feel each of these observations include novel aspects. We will, in the manuscript, discuss these findings in the context of their original description in the literature.

For the reviewer's point 6a - we show that the *MDM2* ecDNA has a stronger negative correlation than *MDM2* focal amplifications with *TP53* mutations in tumour samples.

For the reviewer's point 6b - Kim et al explore the distribution of oncogenes across tumour types, and we build on this observation to describe immunomodulatory and regulatory ecDNA previously undescribed in patient cohorts.

For the reviewer's point 6c - We do perform a survival analysis as seen in Kim et al, however we attempt to address some questions not addressed in that paper. These include exploring the impact of disease stage and underlying genome instability on survival. Furthermore, we control for many more confounding factors in the Cox proportional hazards model.

For the reviewer's point 6d - We build upon APOBEC kataegis and omliki events by comparing the SBS signature makeup of the ecDNA themselves against global signatures and use our mutational timing method to determine which mutational process are active before and after ecDNA formation.

6e - We show that immunomodulatory ecDNA are pervasive across multiple tumour types and correlates with T cell depletion.

7. The study lacks orthogonal approaches that are important to validate some of the findings:

- a. The presence of ecDNA could be validated using breakpoint PCR experiments.
- b. The presence and copy number of ecDNA of different types could be measured using straightforward DNA FISH using tissue sections. Are immunomodulatory genes really amplified, or only exist in ecDNA? To what extent the amplifications called using their computational tools and applied to this particular dataset (Genomics England) is accurate? Validation of oncogene amplifications using DNA FISH is important.

We appreciate the reviewer's comment and the need for orthogonal approaches to validate some of our findings.

We validated oncogene amplification in this particular dataset (Genomics England). Genomics England is a company owned by the Department of Health from which its flagship project (100,000 Genomes Project) sequenced 100,000 whole genomes of NHS patients with rare diseases and cancer from 13 NHS genomics centres across the UK. As we own neither the data nor the samples, reliably getting hold of archival tissue has proved extremely difficult, however we were able to obtain samples 11 from patients in the dataset.

Of these 11 patients: 6 had dedifferentiated liposarcoma, 3 had parosteal osteosarcoma, 1 had angiosarcoma, and 1 had extraskeletal osteosarcoma. We validated the amplification of 4 genes using

interphase FISH; *MDM2*, *CDK4*, *PDGFRA* and *MYC*. As a control, we were able to identify a sample with predicted chromosomal amplification of *PDGFRA* (a BFB that had not formed an ecDNA). In all 11 samples copy number amplifications that were predicted using AmpliconArchitect were confirmed as having amplified the corresponding genes. Furthermore, in the 10 samples where ecDNA was predicted, we confirmed the characteristic appearance of ecDNA in 9/10 of the FISH images, with the exception being the angiosarcoma from patient 1. For the extraskeletal osteosarcoma of patient 7 that was described as having focal a *PDGFRA* amplification from WGS analysis, we also confirmed the characteristic appearance of chromosomal amplification in 20/20 interphase nuclei.

Representative interphase FISH images are shown in Figure 1b and Extended Data Fig. 10

Figure 1b: Representative FISH images and AmpliconArchitect structural variant views from two GEL cases

The first image (left) demonstrates an amplicon predicted to be a chromosomal amplification consistent with its FISH image and the second image (right) shows an amplicon predicted to be an ecDNA consistent with its FISH image.

Patient 1 - angiosarcoma
ecDNA predicted by AmpliconArchitect

Patient 6 - dedifferentiated liposarcoma
ecDNA predicted by AmpliconArchitect.

Patient 2 - parosteal osteosarcoma
ecDNA predicted by AmpliconArchitect

Patient 7 - extraskeletal osteosarcoma
chromosomal amplification predicted by AmpliconArchitect.

Patient 3 - dedifferentiated liposarcoma
ecDNA predicted by AmpliconArchitect.

Patient 8 - dedifferentiated liposarcoma
ecDNA predicted by AmpliconArchitect.

Patient 3 - dedifferentiated liposarcoma
ecDNA predicted by AmpliconArchitect.

Patient 9 - dedifferentiated liposarcoma
ecDNA predicted by AmpliconArchitect

Patient 4 - dedifferentiated liposarcoma
ecDNA predicted by AmpliconArchitect.

Patient 10 - dedifferentiated liposarcoma
ecDNA predicted by AmpliconArchitect.

Patient 5 - parosteal osteosarcoma
ecDNA predicted by AmpliconArchitect.

Patient 11 - parosteal osteosarcoma
ecDNA predicted by AmpliconArchitect.

Extended Data Fig. 2 - Interphase FISH on GEL sarcoma samples

Images of representative fields of view from interphase FISH images for each of the 11 sarcoma tumours from the GEL cohort for which material was available. Each image is annotated with the FISH probes applied in addition to DAPI (blue) and the AmpliconArchitect prediction of the presence of either a chromosomal amplification or ecDNA.

Then, in order to provide support for our findings in HER2-positive breast cancer we confirmed the appearances of ecDNA and non-ecDNA amplifications in a separate cohort of 5 HER2-positive breast cancer cases. Three cases are shown below which illustrate the respective patterns of chromosomal and ecDNA amplifications in these samples. BRCA005, BRCA004 - HER2 ecDNA, BRCA008 - HER2 intrachromosomal amplification. These data are present in the revised manuscript as Extended Data Fig. 4.

Representative fields of view from three HER2-amplified breast cancer samples. Scale bar is 10um

Extended Data Fig. 4: Interphase FISH of HER2-amplified breast cancer cases

Images of representative fields of view from interphase FISH images for each of the 3 HER2+ breast cancers from an independent cohort for which material was available for FISH but not for WGS.

c. Figure 2D – T cell exhaustion. For example, study of sections from samples with immunomodulatory ecDNA could show the enrichment (or lack thereof) of T cells in tumor sites that contain immunomodulatory ecDNA (using IF-FISH). Alternatively, single cell RNAseq could be informative.

We thank the reviewer for this comment and agree that obtaining sections with immunomodulatory ecDNA could show the lack of enrichment of T cells, however obtaining sections from the Genomics England cohort has proven extremely difficult. Furthermore, Genomics England has not provided

researchers with an RNA seq cohort. Orthogonal validation of the T Cell ExTRECT scores is shown in Bentham et al. 2021¹³.

We believe that both AmpliconArchitect¹⁴ and TCell ExTRECT¹³ are both extensively validated in orthogonal datasets.

d. The relevance and importance of lncRNA in this study is not clear. Does this lead to any functional outcome, e.g. increased RNA expression? By the same token, do samples with lncRNAs show decreased expression of “sponged” genes? RNA sequencing would be useful here.

We agree that this is a good suggestion. We find that ecDNA that do not contain genes often amplify lncRNAs. However while RNAseq would be extremely useful to better characterise the effect of lncRNA amplifications on the expression of ‘sponged genes’, unfortunately the GEL cohort does not have any RNA seq performed. In addition, as already mentioned in a previous comment to the reviewer, obtaining tissue from these samples has already been thoroughly explored with very limited results.

8. Analysis of multiregion biopsies provided a glimpse of how measurements could underestimate ecDNA presence (Line 146-148). Could the authors estimate the true ecDNA % based on biopsy sizes? Do we detect more ecDNA when examining a larger biopsy, or do we detect more ecDNA when we sample multiple biopsies (perhaps indicating that in some cases technically we miss them)?

This is an important issue and we are able to show that there is a sampling bias associated with ecDNA detection. Unfortunately, the Genomics England dataset does not annotate the biopsy or tumour sizes. However, we performed an analysis that shows that the number of biopsies themselves (in a single tumour) associates with ecDNA detection (Extended Data Fig. 7a,b). We also know that tumour size (as indicated by stage) correlates positively with ecDNA detection (Figure 5e).

To provide an estimate of the possible true ecDNA prevalence based on biopsy number, we extrapolated from the regression analysis. For every biopsy sampled the odds of % ecDNA detection increases by 6%. If we were to extend this to the cohort we could estimate that the true ecDNA prevalence this cohort lies around 22.9% of tumours.

9. Figure 3 does not clearly show the correlates of genome instability and ecDNA. Could the authors improve this figure and distill the main idea?

We thank the reviewer for their feedback and in response we have altered Figure 3, its legend, the main text, and methods. As a result, we hope that the improved figure keys and legend have increased the clarity of the figure (Figure 3, reproduced below).

We investigate genome instability in the context of ecDNA using three measures: detection of whole genome duplication, the calculation weighted genome instability index (wGII), and the quantification of

structural variant burden. Each of these measures was found to be associated with the presence of ecDNA in at least one tumour type, however in different tumour types different measures of genome instability were found to be significant (Figure 3c). Each measure captures a different aspect of genome instability. Whole genome duplication detection described drastic shifts in ploidy while wGII (which takes ploidy into account) aims to capture the proportion of the genome affected by both structural and numerical chromosomal alterations resulting from chromosomal instability. Finally, structural variant burden quantifies the number of individual alterations that result from genomic rearrangements.

In the bottom right panel the distribution of the amplicon complexity score illustrates the context with which ecDNA occurs (whether it is simple or complex). The top panel provides the reader with the mutations that are associated with ecDNA presence. Taken together the reader should come away with the impression of which tumour suppressor mutations are more likely to occur with ecDNA, which “category” of genome instability is more likely to occur and what the context of the ecDNA are themselves. We hope the new figure represents this idea more clearly.

Figure 3 - Correlates of genome instability and ecDNA: a, Forest plot showing results of a regression model that investigates associations between tumour suppressor losses and the presence of ecDNA or chromosomal amplifications across the entire cohort controlling for

cancer type. Associations with ecDNA are represented by circles and chromosomal amplifications represented by diamonds. b, Bar plot showing the proportion of tumours across the cohort with any ecDNA (blue), *MDM2* ecDNA (yellow) or no ecDNA (grey) grouped by *TP53* mutation status. c, Body map with panels for selected cancer types. Each panel contains (top) a forest plot showing associations between ecDNA presence or absence with tumour suppressor losses. (bottom left) A forest plot showing associations between ecDNA presence or absence with weighted genome instability index, structural variant burden, and whole genome duplication. (bottom right) A violin plot demonstrating the amplicon complexity scores for the tumours of that cancer type.

10. What does Figure 4 teach us regarding ecDNA evolutionary trajectories? Can mutation rates be used to estimate the age of ecDNAs? Theoretically, more DNA copies should have higher chances of acquiring mutations than wild-type conditions, and this difference could be used to infer times.

Many thanks for the comment. To understand *when* ecDNA forms we use the distribution of mutational copies to infer whether they are present on all ecDNA copies, or just a subset. Those that are present on all ecDNA copies are concluded to have formed before the ecDNA did. We can then use the SBS signatures to then time mutational processes; Figures 4f,g for example show that signatures of homologous recombination deficiency tends to occur after ecDNA formation and suggests that homologous recombination is not necessary for the maintenance of ecDNA.

11. Figure 5B shows a correlation between ecDNA and tumor staging. Does this differentiate between the possibilities that in increased stages the chances to detect ecDNA increase (because it is more widespread now through clonal domination), or that new ecDNA form at advanced stages?

This is an interesting question and one we would like to explore further. The question of clonal domination vs de-novo formation can equally be leveled at drug resistance clones. In this scenario, drug resistant clones exist in pre-treated samples at minimal frequencies that may be undetectable using a bulk whole genome sequencing approach. They become widespread due to clonal domination, and are then detectable at later time points. This is in contrast to tumours that acquire de-novo alterations following treatment.

With single sample timepoints it is extremely difficult to differentiate between the two scenarios but previous studies tell us both scenarios are a possibility. In the case of clonal domination, ecDNA has been shown to occur in pre-malignant lesions such as Barrett's oesophagus⁴. These ecDNA are present following transformation and in this scenario it is possible that clonal domination has occurred. In terms of de-novo ecDNA formation, it has been shown in HeLa cells that methotrexate can induce ecDNA through repeated rounds of chromothripsis¹¹.

In Figure 4h, we show that temozolomide can cause a bottleneck and subsequent clonal sweep by virtue of the fact that mutations are mapped to a fraction of ecDNA copies, further supporting a clonal dominance hypothesis.

The optimal way of studying this phenomenon in patients is through longitudinal and multi-region sampling, however the Genomics England dataset is not a longitudinal dataset.

12. Lines 422-435: “We highlight that the relationship between ecDNA and genomic instability differs according to tumour type. **This provides further evidence** suggesting that ecDNA may arise from different mechanisms, including episomal formation, chromothripsis, breakage-fusion-bridge cycling and translocation-bridge-amplifications.” This statement in the discussion relates to Figure 3 but the data does not clearly illustrate the potential involvement of any of the mentioned mechanisms. Related to this, BFB leading to ecDNA and the presence of kataegis in such cases was also not examined (or discussed).

We thank the reviewer for the comment and agree that greater clarity regarding how we have inferred the potential involvement of any mechanisms is required. Therefore, we have removed this statement from the discussion and instead refer to each relevant mechanism in the results as evidence for its involvement arises in the data:

Translocation-bridge-amplification relates to the presence of interchromosomal ecDNA shown in Extended Data Fig. 5b,c and we have expanded on this finding in the main text:

Line 116-118: “Most ecDNAs arose from a locus on one chromosome (89.9% n = 3705). However, ecDNAs composed of distinct inter-chromosomal translocations, and were seen most commonly in sarcomas and breast cancers, Extended Data Fig. 5b,c). Of note, inter-chromosomal translocations observed in breast cancer such as t(8;11), t(8;17) and t(11;17) might arise through the recently described translocation-bridge mechanism”

Episomal formation is inferred through the low amplicon complexity and chromothripsis we infer through highly complex amplicons which we now show in Figure 3c, and explain the amplicon complexity score in Extended Figure 1b and its own dedicated methods section.

Line 243 - 245: “Low complexity amplicons may be associated with the episomal formation of ecDNA, whereas high complexity amplicons are likely associated with catastrophic events such as chromothripsis”

13. Figure 5A related to metastasis and correlation with ecDNA is not clear. It appears all amplification types correlate with metastasis. Also, do the amplifications pre exist in the primary tumors (can they compare metastatic vs primary from the same patients, if they have such samples?). If there is any way to determine if the amplifications formed de novo at the metastatic site – that would be extremely

interesting.

We did find that both non-ecDNA focal amplifications and ecDNA are enriched in metastasis, albeit the odds of finding ecDNA in metastatic samples was higher than in non-ecDNA amplifications. The question of whether the ecDNA existed in the primary is extremely interesting, unfortunately the Genomics England dataset does not have the longitudinal sampling cohort required to answer this question fully.

Minor comments:

1. Line 202 refers to Figure 2E. Shouldn't it refer to Figure 2D instead?

We thank the author for drawing our attention to this error that has now been corrected.

2. Figure 2D: what is the Y axis label?

We now provide the Y axis label "Total ecDNA occurrences". Please see the bar chart from Figure 2D reproduced below.

3. Line 86 claims: "Focal amplifications were defined as regions of the genome between 50kb and 20Mb in size". However, in their analysis in Figure 2A the authors claim: "...the median ecDNA size was larger for ecDNA that 182 encoded oncogenes compared with the other two categories (figure 2A, median ecDNA size 2.26 183 Mb vs 0.535 Mb vs 0.053 Mb, $p < 10^{-16}$). If the limit of amplicon size was set to 50kb, how could non-coding ecDNA have a median of 53kb?

The reviewer makes a good point; in this case - the seed interval (See Methods) that was provided to Amplicon Architect has a minimum threshold of 50 kb. Within that seed interval it is possible that an ecDNA reconstruction could be less than 50 kb.

We have altered the text:

Line number 577: " For this analysis, seed intervals were defined as regions of over 50Kb, with a threshold copy number of greater than 4.5, double the ploidy of the tumour and at least 2.5 additional copies above the median arm-level copy number. The regions are then merged to form a breakpoint graph, which can be broken down into simple and complex cycles to identify any circular paths that could be indicative of ecDNA. Within the seed interval, it is possible that an ecDNA reconstruction could be less than 50Kb."

4. Figures 5E and 5D are mixed in the text?

We thank the author for drawing our attention to this error that has now been corrected.

In summary, the work has great potential, and the dataset used is a treasure trove that could illuminate the biology of ecDNA in cancer. By addressing the points above, this work could have great impact within the field and also to broader audiences interested in cancer genome evolution. The most exciting findings here are (1) the presence of non-oncogenes amplified on ecDNA (immune related, lncRNA), (2) the possibility of using mutations to delineate ecDNA evolution trajectories, and (3) the role of ecDNA in metastatic sites. Strengthening these points, also using orthogonal approaches, will provide substantial novelty and impact.

Referee #2 (Remarks to the Author):

Bailey and colleagues set out to detect and analyse focal amplifications, defined as 50kb-20mb regions with at least four copies and twice the ploidy, across the Genomics England (GEL) cohort, a relatively untapped resource that includes tumour whole-genome sequences from more than 15,000 patients with cancer. In particular, the analysis focused on extrachromosomal DNA amplifications (ecDNAs). The large and diverse number of tumours included in the GEL cohort enabled the authors to expand the ecDNA landscape beyond their previous publication (Kim et al), such as by disease subtype (i.e. HER2+ breast cancers). The depth and breadth of the cohort further resulted in several new discoveries, such as an enrichment for the presence of immunomodulatory genes on ecDNA amplicons, beyond the previously established frequent presence of oncogenes. Cross-referencing ecDNA-containing tumours with mutational signatures detected an association with for example APOBEC signatures and a negative correlation with hypermutation. The results help further define the role that ecDNA plays across cancer and present a landmark for further studies.

We thank the reviewer for their thoughtful and generous assessment and for their specific comments and suggestions.

Major comments

1. Samples with tumour purity < 0.1 were excluded. How was tumour purity determined? Supporting evidence is needed to justify this tumour purity threshold, in terms of amplicon and mutational signature detection sensitivity.

We have a detection threshold of CN=4.5 set in AmpliconSuite-pipeline (PrepareAA), as this helps to effectively separate events which are large segmental genomic duplications, on both chromosome copies which have an apparent CN=4 against the reference genome (for example where there is whole genome doubling), from events that are bona fide focal amplifications.

With that in mind, sample purity has an effect on the minimum copy number of the focal amplification to surpass the detection threshold. At 100% purity, the threshold is CN=4.5, but at lower purity it goes up as the following function of purity (p):

$$\text{min amplification CN} = 2.5/p + 2$$

The 2.5 value is the number of additional required copies over the baseline genome copy number of 2. So for p = 0.1, the focal amplification must at least reach CN=27 to be theoretically detectable.

Adding purity analysis to AmpliconArchitect is a target for additional methods development, however the state of the art in purity-correction of copy number methods rely on B-allele frequencies, which do not lend themselves well to sensitive genome segmentation and become very noisy at low B-allele frequencies. Furthermore, the profile of b-allele frequencies on ecDNA may further confound these methods and lead to completely unreliable copy numbers. Instead, we make a choice to accept the caveat that lower purity samples may have undetectable ecDNA - a tradeoff that prioritizes an understood mode of error over an unexplored mode of error (B-allele based purity correction of copy number for focal amplifications).

To demonstrate the effect of purity on ecDNA detection, we have previously selected 10 cancer cell lines – 6 lines from Contino et al. 2017¹⁵, and 4 from Cancer Cell Line Encyclopaedia,¹⁶ for which we had previously predicted ecDNA. Across those cell lines AmpliconClassifier predicted 43 distinct ecDNAs, ranging in copy number from 4.8 to 271.3. 37/43 (86%) of the ecDNA in the simulation study contained copy number under 20⁴.

We mixed hg38-aligned cancer cell line whole-genome sequencing reads with reads from hg38-aligned diploid NA12878 cells to form 29 different purity levels between 4.8% purity and 100% purity. All mixed BAM files had coverage >10x and were then subsequently run using Amplicon Architect and Amplicon Classifier. In brief, focal amplification seeds were defined using CNVKit (default parameters for Amplicon Architect, Amplicon Classifier, and CNVKit). The seed intervals were then passed to Amplicon Architect

alongside the BAM files and the default downsampling to coverage 10x was used by Amplicon Architect so that all simulated purity levels and for all ecDNA had the same coverage during analysis.

We then used AmpliconClassifier to detect ecDNA from the resulting outputs using the default threshold of copy number > 4.5 . A sample was assumed to have correctly identified ecDNA if it produced an ecDNA classification overlapping the original cell line's ecDNA locations in the genome. We plotted the relationship of the original copy number, purity and ecDNA detection status below:

Reviewer Figure 2.1: Effects of purity simulations on ecDNA detection - Reproduced from the point-by-point response to reviewers from Luebeck et al. "Extrachromosomal DNA in the cancerous transformation of Barrett's oesophagus" Nature 2023.

We found 71.8% of theoretically detectable ecDNA (ecDNA with a copy number of > 4.5) were recovered across all purity levels. We plotted the sensitivity of the method against the simulated purity and observed the sensitivity of ecDNA recovery was higher for higher levels of purity.

Reviewer Figure 2.2: Effect of purity simulations on AmpliconArchitect sensitivity - Reproduced from the point-by-point response to reviewers from Luebeck et al. “Extrachromosomal DNA in the cancerous transformation of Barrett’s oesophagus” Nature 2023.

Perfect recovery of ecDNA is impossible with short read sequencing, and more challenging with lower purity. This is somewhat highlighted as the median purity is higher for samples where ecDNA is detected (56 vs 51 vs 48, $p < 1 \times 10^{-16}$). Moreover and unsurprisingly, adjusting for tumour type, the odds of detecting ecDNA is 1.15 (95% CI 1.12-1.18) for every 10% increase in purity.

Reviewer Figure 2.3: Box plots showing tumour purity by amplification category across the GEL cohort.

The decision to use a 10% purity threshold was taken based on three factors. First, that 10% was the cut-off used for analyses in the Genomics England landmark paper (Sosinsky et al. 2024)¹ apart from their section on clinical actionability. We used the same method of purity detection as used in that paper¹⁷.

Secondly, we ran AA across all samples and continued to detect ecDNA at purities <20%, with the lowest purity from which ecDNA was detected at 7%. By reducing the threshold from 20% to 10% we were able to include 99.9% ($n = 2797$) of samples with ecDNA in our analysis from 96.2% ($n = 2692$). We felt that the 10% purity threshold and its increased inclusion of samples was important for some of the descriptive aspects of the paper (e.g. the description of ecDNA contents and the overrepresentation analysis in Figure 2).

However, it is important to note that for the downstream analysis of mutational signatures we use a purity threshold of 20%. This threshold was used by Degasperis et al.¹⁸ in their analysis of mutational signatures in the same Genomics England dataset as our analyses and as such was likely chosen as the most appropriate purity threshold for mutational signature analysis. Furthermore, in all logistic

regression analyses involving clinical correlates (and T cell infiltration estimates) we adjust for purity as we recognise it as an essential confounding factor in all analyses.

Thirdly, in our analysis we aimed to get robust estimates for as many tumour types as possible; for many tumour types getting high purity samples is extremely difficult either due to the location of the tumour or on the basis that some tumour types are heavily infiltrated with stroma, such as cholangiocarcinoma and pancreatic adenocarcinoma. Indeed, these differences in cancer type purities can be seen in Extended Data Fig. 3a which we have added to our revised manuscript to allow readers to take this variability into account.

Extended Data Fig. 3a: Box plots showing purity estimates for each of tumour samples from the 39 cancer types in the GEL cohort.

Finally, we have added the following text to our discussion to highlight the various limiting factors on ecDNA detection including purity:

Line 348-354: *“Bioinformatic detection of ecDNA from WGS data has inherent limitations. Some of these limitations are biological, such as the effects of tumor purity and ecDNA copy number. However, some are more technical, including the detection of SVs in repetitive regions of the genome, the effects of sequencing coverage, and algorithmic challenges in distinguishing types of focal amplifications. While the ecDNA-detection methods used here have been shown to be robust, improvements in sequencing technologies and methods for ecDNA detection should provide even more refined estimates of ecDNA frequency across cancers.”*

2. Genome sequencing coverage may be a second possible confounder for sensitivity and specificity of amplicon and mutational signature detection, and supporting evidence is needed to demonstrate that the comparisons shown are not impacted by sequence coverage.

We thank the reviewer for their excellent and rigorous assessment of the methods used. All samples analysed in this study were downsampled to 10x baseline coverage by AmpliconArchitect. Consequently,

the effect of varying levels of total sequencing coverage across the samples was controlled for, as each sample was analysed at the same sequencing coverage.

Critically, this downsampling procedure also served the practical purpose of reducing the runtime of AmpliconArchitect to a degree at which we could reasonably analyse such a large collection of samples.

However, to understand the effects of downsampling to 10x coverage, we performed a carefully controlled comparison to this effect on a panel of WGS data from 21 cancer cell lines with original sequencing coverage of the standard 30x coverage or higher, with a generally high rate of focal amplifications. We deployed AmpliconSuite (AmpliconArchitect + AmpliconClassifier, with CNVkit used for seeding), in the same manner as used in the analysis of the Genomics England samples, and performed a comparison of ecDNA prediction between the standard 10x downsampling and reduced downsampling (higher coverage) of 30x.

At 10x coverage, 16 of 21 samples were predicted to be ecDNA-positive (16 samples with 49 total ecDNA species), while at 30x coverage only one additional sample was predicted as ecDNA-positive (17 samples with 57 total ecDNA species) – due to a BFB prediction that switched to an ecDNA prediction. The change in ecDNA-positive status was acceptably small, and the increase in total discovered ecDNA species was fairly modest. Notably the changes in number of ecDNA species were largely driven by shifts in classification of focal amplification types detected at both levels, not by previously undetected focal amplifications. When we removed downsampling altogether, one additional sample was predicted as ecDNA-positive compared to 30x (18 samples, 58 ecDNA), however runtimes became significantly longer.

Because it is important to communicate the limitations of the bioinformatic methods used, we have added the following lines to our Discussion:

Line 348-354: “Bioinformatic detection of ecDNA from WGS data has inherent limitations. Some of these limitations are biological, such as the effects of tumor purity and ecDNA copy number. However, some are more technical, including the detection of SVs in repetitive regions of the genome, the effects of sequencing coverage, and algorithmic challenges in distinguishing types of focal amplifications. While the ecDNA-detection methods used here have been shown to be robust, improvements in sequencing technologies and methods for ecDNA detection should provide even more refined estimates of ecDNA frequency across cancers.”

3. Some details on the patient cohort would be valuable for interpretation, for example to what extent were tumour newly diagnosed or pre-treated, if there were pre-treated tumours, how are those distributed across tumour types. What is the impact of any unevenness in pre-treatment status between tumour types on the downstream analyses?

We agree that this is a valid criticism as details on the patient cohort, including treatment status, was lacking. In response we have elaborated on the patient cohort, including more information on the treatment status.

For the dataset; in the main text we have added a section at the beginning of the introduction, titled “Genomics England v12 cohort”:

“To map the frequency and contents of ecDNAs in human cancer, we analysed a cohort of 16,341 samples of whole genome–sequenced cancers from 15,223 patients recruited across 13 UK National Health Service (NHS) Genomic Medicine Centres as part of the Genomics England (GEL) 100,000 Genomes Project (100kGP) (GEL v12 data release). Samples with tumour purity of <10% were excluded, as were cancers of unknown primary, pediatric cancers and testicular germ cell tumours (510 samples). Within the cohort, 3.8% (598) of samples were fixed-formalin paraffin embedded (FFPE). Staging information was available for 10,780 (72.9%) of patients, with 836 (5.7%) patients recorded as having Stage 4 disease (Figure 1a). Median depth of coverage for tumour samples was 97.6x and for germline samples was 32.6x. 1,800 (12.1%) patients were recorded as receiving systemic anti-cancer treatment (SACT) prior to biopsy.”

With regards to treatment, the v12 Genomics England dataset have curated treatment information for the dataset. We took those patients who had received systemic anticancer therapy (SACT) then explored the distribution of different types of treatment across the cohort. Of the 14,778 cancer patients in the study, 1,800 patients were recorded as receiving hormonal (n = 27), immunotherapy (n = 57), targeted (n = 415) or cytotoxic chemotherapy (n = 1653) prior to biopsy:

Reviewer Figure 2.4: Stacked bar plot showing the distribution of treatment type by cancer type

We then conducted an analysis looking at the association between ecDNA and whether or not patients were treated. Due to low numbers, those patients who had received hormonal treatment were

excluded. In a logistic regression model, adjusted for age, stage, purity and tumour type, the detection of ecDNA was associated with prior chemotherapy (OR 2.38 95% CI 1.73-3.27) and targeted therapy (OR 2.87, 95% CI 1.12-6.43, Figure 5c).

Figure 5c: Forest plot demonstrating the association between treatment and ecDNA detection. This model adjusted for cancer type, age, sex and purity.

To explore the impact of unevenness in pre-treatment status, we included it in a regression model to understand the association of pre-treatment with ecDNA status. We find a significant association between pre-treatment with chemotherapy and targeted therapy. We have added this finding to the paper as Figure 5c.

4. In a subset of samples, two or more distinct ecDNAs were detected. Previous reports have demonstrated the presence of multiple oncogenes on single ecDNAs, i.e. EGFR-CDK4 combination in glioblastomas (PMID 29686388) or MYC-CDK6 in pediatric gliomas (PMID 30267146)? Evidence is needed to support the claim that in this analysis, the oncogenes are on distinct structures, such as the absence of sequence reads connecting the two loci. If the observations are restricted to distinct ecDNA

pairs: does it matter if two oncogenes are on two distinct ecDNAs, or part of the same ecDNA molecule structure?

In the Genomics England cohort we find evidence of the CDK4-EGFR on the same ecDNA in some cases ($n = 2$, now shown in the heatmap on Extended Data Fig. 7c), and on separate ecDNA in other cases ($n = 11$). We make the claim that these ecDNA are on distinct structures on the basis that AmpliconArchitect does not detect any structural variant breakpoints that connect the decompositions. An example is provided below:

The coverage and structural variant plot of an amplicon containing *CDK4*:

Reviewer Figure 2.5: The coverage and structural variant plot of an amplicon containing *CDK4* from a GBM patient called as having both an ecDNA with a *CDK4* and a separate ecDNA with *EGFR*.

And an amplicon from the same patient demonstrating an *EGFR* containing ecDNA:

Reviewer Figure 2.6: The coverage and structural variant plot of an amplicon containing *EGFR* from a GBM patient called as having both an ecDNA with a *CDK4* and a separate ecDNA with *EGFR*.

To understand if there could be any discordance between SV calls from AmpliconArchitect and SV calls from an alternative caller, we explored the cases of *CDK4-EGFR* predicted on separate ecDNA and reviewed the output from Manta, an alternative structural variant caller. We found evidence of breakpoints within 4Mb of the *CDK4* locus in 2 of 11 samples, indicating that a t(7;12) interchromosomal breakpoint is likely present in the sample. However, from the AmpliconArchitect decomposition we find no evidence that these genes recombine to the same ecDNA. It should be noted that both samples are above the mean purity of 50.1% (83% and 53%).

The importance of this phenomenon is an interesting subject of debate. We find that the oncogene copy number is significantly different between oncogenes on 2 separate ecDNA (Extended Data Fig. 7e) - it is likely that the interplay between these ecDNA may have implications for tumour behaviour, and will certainly have evolutionary constraints. The numbers were too small for a robust survival analysis that controlled for all appropriate confounding factors but this should be the subject of future research.

5. How were translocation ecDNAs detected? This is not described in the methods. It is intriguing that translocation ecDNAs were predominantly found in sarcomas, whereas the enrichment of two or more ecDNAs in the same sample was observed in CNS tumours.

We detect translocation-ecDNAs (now termed multi chromosomal-ecDNA in the revised manuscript) by the presence of interchromosomal (translocation) structural variants that are used by AmpliconArchitect to construct cyclic paths as part of its ecDNA calling algorithm. Specifically, AmpliconArchitect uses discordant read pairs in the BAM file located within the seed intervals and if the discordant pairs are mapped to separate chromosomes this is annotated as an interchromosomal breakpoint which make up the translocation-ecDNA.

We agree that it is interesting that these ecDNAs are predominantly found in sarcomas (and ER+ breast cancers). For breast cancers it is likely that translocation-bridge amplifications form the initial interchromosomal connection from which the ecDNA forms. For sarcomas it would be interesting to understand whether there is a specific process generating these ecDNA types or whether it is just a general consequence of chromothripsis and these questions could form the basis of future work.

6. Genes on ecDNAs were more likely to have oncogenic function compared to non-ecDNA amplifications, is this also true when copy number level is considered? I.e. does this observation remain true when binning by various levels of copy number?

This is a great question. Focal amplifications driven by ecDNA exhibited higher copy number for oncogenes compared to non-ecDNA driven focal amplifications (see Extended Data Fig. 5a). However this does address whether ecDNA are enriched in oncogenes irrespective of copy number. To answer this - we binned amplifications according to copy number (<8, 8-12, 12-16, 16-20, 20-40, 40-80, 80+). We

found that the mean number of oncogenes was significantly higher in ecDNA focal amplifications vs chromosomal amplifications in 4 of the 7 copy number bins (the precision of estimates at higher copy number counts get larger as there are fewer chromosomal amplifications at extreme copy number).

Reviewer Figure 2.7: Bar plot showing the mean number of oncogenes per ecDNA binned by copy number

We then performed a similar analysis according to amplification size (Extended Data Fig. 7c); finding an enrichment for oncogenes in ecDNA compared to chromosomal amplifications. We could include in the result if the reviewer felt necessary.

Extended Data Fig. 7c: Bar plot showing the mean number of oncogenes per ecDNA binned by amplification size.

7. Figure 1D appears to reflect the association between amplification and mutation frequency of the ecDNA cargo oncogene. The associated text does not reflect the analysis and appears to repeat the term ‘amplification’.

We apologise for the lack of clarity regarding this result in our original submission. Figure 1 has undergone substantial modification in our revised manuscript. The figure the reviewer referred to in their comment has now been split into two separate figures that we have reproduced below.

The frequency of amplification of particular oncogenes on ecDNA across tissue types is now depicted by Figure 1e, reproduced below:

Figure 1e: Oncogene frequency on ecDNA

Bar plot showing number of occurrences of ecDNA containing oncogenes with the colour of the bar indicating the number of cases from each tissue type. Only oncogenes present on ≥ 20 ecDNA in the GEL cohort are shown.

The examination of the non-synonymous mutations found affecting oncogenes present on ecDNA is shown in Figure 1h, also reproduced below:

Figure 1h: ecDNA mutations and their timing

Stacked bar plots displaying (top) the proportion of types of non-synonymous mutations observed in the oncogenes present on ecDNA and (bottom) the proportion of these non-synonymous mutations in different timing categories (see Methods). Only the mutations affecting the 20 oncogenes most commonly present on ecDNA are shown.

We hope that splitting the figure in this manner has improved the readability of our revised manuscript.

8. In multi-sector analysis (Figure 1F), was the likelihood that an ecDNAs was subclonal (i.e. private to a single region rather than present in all sequenced regions) different compared to non-ecDNA amplifications?

We thank the reviewer for the comment, and find that chromosomal amplifications are more likely to be clonal compared to ecDNA (see Reviewer Figure 2.7 below). We think that this is in part related to the inherent differences between spatial localisation of ecDNA within a tumour compared to chromosomal amplifications. If ecDNA is inherited in a non-mendelian fashion¹⁰ then it is possible that sub-clones without ecDNA are more likely to emerge even if the 'most recent common ancestor' had ecDNA.

Reviewer Figure 2.7: Using patients with multiple samples to investigate focal amplification clonality

Stacked bar plots showing the proportion of patients with multiple tumour regions and chromosomal amplifications grouped by the number of regions sampled. (left) Stacked bar plot examines chromosomal amplifications and the fill indicates (grey) a chromosomal amplification detected in ≥ 1 region from the patient's tumour or (red) a chromosomal amplification detected in all regions. (right) Stacked bar plot examines ecDNA and the fill indicates (grey) an ecDNA detected in ≥ 1 region from the patient's tumour or (red) an ecDNA detected in all regions.

9. The observation with respect to an enrichment for immune modulatory genes on non-oncogene containing ecDNAs is interesting in light of the structural features (circularity) and possible association between ecDNA and micronuclei (i.e. per PMID 9508765). If this enrichment is indeed related to the unique behaviour of ecDNAs relative to other amplifications, it may be expected that non-ecDNA amplicons do not reflect this pattern, thus corroborating the observation. What gene classes do non-oncogene containing non-ecDNAs carry?

We agree that the enrichment for immune-modulatory genes may be related to the unique structural features of ecDNA. To corroborate the observation we performed a similar overrepresentation test on the separate focal amplification classes in the dataset. Our first observation was that the number of significant hits on ecDNA was far higher than the other classes enriched on non-oncogenic focal amplifications. Non-ecDNA amplifications such as linear amplifications and breakage-fusion-bridge amplifications (BFBs) had no significant hits given the gene set threshold of 100, so we reduced our gene set threshold to 20 for these focal amplification types, keeping q value threshold at 0.01 and p value threshold at 0.005). Linear and BFB amplifications show no enrichment for any gene set we would consider immune-modulation (see Reviewer Figure 2.8 below).

Reviewer Figure 2.8: Overrepresentation analyses of ecDNA, BFBs, linear, and complex non-cyclic amplifications.

In addition, genes found in chromosomal amplifications that did not encompass oncogenes showed enrichment for only a geneset corresponding to cellular defense response.

Extended Data Fig. 8c: Plot showing results of an overrepresentation analysis of genes found in chromosomal amplifications that do not contain oncogenes.

We also took the ‘background enrichment’, that is an overrepresentation analysis for all focal amplifications (including oncogenes) and non-ecDNA, non-oncogenic focal amplifications (using the same gene set and q value thresholds in the original analysis) and have detailed this in Extended Data Fig. 8b.

B**Overrepresentation analysis for all focal amplifications**
Extended Data Fig. 8b: Plot showing results of an overrepresentation analysis of genes found in any focal amplification.

10. Figure 2E is implied to show T-cell fraction/depletion in immunomodulatory ecDNA samples, but the actual figure appears to reflect lncRNAs across ecDNAs which are described in the next paragraph. Fig 1F is referenced in that paragraph but this should have been 2E. The actual Figure 2E shows a particular enrichment for the RP11-89m16.1 lncRNA on ecDNAs across cancer but in particular in uterine cancers. Does this lncRNA have any known function and is there evidence for a role of this lncRNA in cancer, that could help explain the uterine cancer enrichment?

We apologise for the figure mislabelling and have corrected accordingly. We originally touched on this in the discussion “increased expression of lncRNA RP11-89 has been shown to facilitate tumourigenesis in GBM through the sponging of miR-623, which naturally functions to downregulate cyclin-D1”. This has been shown in GBM¹⁹ and bladder cancer²⁰. We were unable to show evidence of RNA sponging due to the lack of bulk RNA sequencing data in the Genomics England dataset.

11. Extended Data Figure 4 shows genes that are co-altered in the presence of ecDNA and non-ecDNA amplifications. The text gives the impression that TP53 is significantly more frequently associated with ecDNA than non-ecDNA amplifications/no amplifications, but this is not obvious from the figure nor is there clear evidence in the form of a statistical test.

Extended Data Figure 4 in our original manuscript (now Extended Data Fig. 9) is a plot that looks at which genes are under selection in the presence and absence of ecDNA (in all combinations). This is in contrast to Figure 3a, which is a regression model that compares the likelihood that a sample with ecDNA also has an inactivating mutation in a tumour suppressor gene. The purpose of both plots is to try and quantify which genes are associated with ecDNA formation; we have amended the text as below:

Line 228-236: “Controlling for tumour type, *TP53* mutations were most strongly associated with ecDNA (OR 2.26 95%, CI 1.96-2.62) (Fig. 3a). *TP53* mutant tumours (nonsense and missense mutations) and *MDM2* encoded within ecDNAs were mutually exclusive (χ^2 $p = 0.00006$, Fig. 3b). We also detected tissue-type specific tumour suppressor pathway mutations and their association with ecDNA. ecDNA was strongly associated with *TP53* mutations in endometrial, melanoma, renal and luminal ER+ breast cancer (Fig. 3c). ecDNA presence was also associated with *NF1* mutations in sarcoma, *ARID1A* mutations in renal cancers and *RB1* mutations in bladder cancer (Fig. 3c). *TP53* was most commonly under selection (using dN/dS) across cancer types associated with ecDNA (Extended Data Fig. 9).”

12. In the same figure, *CDKN2A* seems to proportionally stand out. This suggests that included in this figure are homozygous deletions, the most common type of *CDKN2A* deactivation, but this is not mentioned. The next section (Fig 3A) aims to identify tumour suppressors associated with ecDNA in a tumour type independent manner. I was unable to find how this analysis was performed and more details are needed. *CDKN2A* homozygous loss is frequent in tumour types enriched for ecDNA such as glioblastoma and lung adenocarcinoma. It would be of interest to perform an analysis to detect associations between tumour suppressors and ecDNA that includes homozygous losses, controlling for tumour type, to determine whether *CDKN2A* is indeed significantly enriched.

Extended Data Figure 4 (now Extended Data Fig. 9) takes into account mutations and small indels in a dN/dS analysis. Figure 3 is a model that determines the odds that a tumour will have a nonsense or missense mutation given the presence or absence of ecDNA; this has been added to the legend for clearer interpretation. An excellent follow up analysis would be to incorporate homozygous deletions as it would be an alternative mechanism of tumour suppressor loss and as such in tumours where *CDKN2A*

homozygous deletion is common there is likely to be an enrichment. Unfortunately, reliable calls for homozygous losses are currently available in the Genomics England dataset across the 15,000 cohort but this is the subject of an ongoing endeavour within the environment.

13. Associations between ecDNA presence and tumour suppressor pathway losses as presented in Fig 3C should be tested for statistical significance. The X-axis in Fig 3C is not annotated. Tumour type acronyms should be written in full to accommodate a broad audience.

Please find the updated version of Figure 3 from our revised manuscript below. We have taken care to ensure that all axes are labeled and that full tumour type names have been used as the reviewer suggests.

Figure 3 - Correlates of genome instability and ecDNA: a, Forest plot showing results of a regression model that investigates associations between tumour suppressor losses and the presence of ecDNA or chromosomal amplifications across the entire cohort controlling for

cancer type. Associations with ecDNA are represented by circles and chromosomal amplifications represented by diamonds. b, Bar plot showing the proportion of tumours across the cohort with any ecDNA (blue), *MDM2* ecDNA (yellow) or no ecDNA (grey) grouped by *TP53* mutation status. c, Body map with panels for selected cancer types. Each panel contains (top) a forest plot showing associations between ecDNA presence or absence with tumour suppressor losses. (bottom left) A forest plot showing associations between ecDNA presence or absence with weighted genome instability index, structural variant burden, and whole genome duplication. (bottom right) A violin plot demonstrating the amplicon complexity scores for the tumours of that cancer type.

14. Is the analysis presented in Fig 4A corrected for tumour type imbalance, as was done for Fig 4B? Would this association between the presence of an ecDNA and increased mutational burden hold true when age is considered as a covariate?

This is an excellent point and in response we have altered Figure 4 to reflect this.

For the reviewer: we first show that TMB and age are correlated:

Reviewer Figure 2.9: Boxplots showing TMB by age and grouped by amplification status.

Following this we performed a regression analysis that looked at the effect of increasing TMB on the detection of ecDNA, controlled for tumour type, purity and age. TMB was categorised into deciles. We found that the odds of detecting ecDNA was increased for every increasing TMB decile (tmb_decile in the plot) compared with chromosomal amplifications (OR 1.04 95% CI 1.02-1.06) and no amplifications (OR 1.14 95%CI 1.13-1.16):

Reviewer Figure 2.10: Regression analyses investigating TMB by amplification category

Forest plots showing (left) associations with increased TMB decile in tumours with chromosomal amplifications compared with those with no focal amplifications. (middle) Associations with increased TMB decile in tumours with ecDNA compared with those with no focal amplifications. (right) Associations with increased TMB decile in tumours with ecDNA compared with those with chromosomal amplifications.

To simplify this we have summarised only the odds ratio for increasing TMB in each category as part of Figure 4b. These represent the individual odds ratios of increasing TMB deciles controlled for tumour type, purity, and age.

Figure 4b: ecDNA mutations and their timing

Forest plot depicting the odds ratio of an increased TMB decile according to the presence of ecDNA or chromosomal amplifications adjusted for purity, age, and tumour type.

15. SBS3 is also associated with response to platinum treatment. Is the observation that SBS3 is enriched on ecDNAs post-formation informed by tumours obtained from patients pre-treated with platinum therapy? Or is this enrichment already detected in untreated tumours, and if so, what kind of mechanism might that suggest?

This is an excellent question; SBS3 is a “noisy” signature which is implicated in HRD and in response to platinum treatment. We wanted to explore this further so we took the samples which had been annotated as having previously received platinum therapy. As now noted in the manuscript; 1,800/14,778 (12.1%) of tumours were pre-treated, with 54.9% (988/1,800) of those having received platinum. Amongst those samples included in the SBS3 analysis (i.e. patients with ecDNA with at least 1 mutation within the ecDNA locus with a probability that the mutation is SBS3 is greater than 0), 152 of 1375 were pre-treated. We compared the distribution of probabilities between those who had received platinum vs those with no platinum pre-treatment recorded. The timing of SBS3 in those pre-treated cases trended towards being early (with no statistical significance - likely due to low numbers). Moreover, when we ran the same analysis using only those pre-treated with platinum, there was a non-significant trend towards early timing (-0.007 , $p = 0.78$). We surmise that as these mutations are more likely to be detected late in untreated tumours, that this may represent a scar left by ongoing defective homologous recombination.

Reviewer Figure 2.11: Examining the clonality of mutations potentially in an SBS3 context

Box plots showing mutations present in a potential SBS3 context in tumours pre-treated with platinum chemotherapy (green) or in tumours with no platinum treatment recorded (red). (left) Shows box plots examining clonal mutations. (right) Shows box plots examining subclonal mutations.

16. It is not really clear what the relevance of the single example in Fig 5C is – with a cohort size of this magnitude, surely there are many interesting ecDNA amplifications in key driver genes, such as whether ERBB2 ecDNA status in BRCA is associated with HER2i treatment response. A justification on the significance of the specific example in 5C is needed.

We agree with the sentiment here and have decided to remove this single case.

17. The discussion is mostly a re-hashing of the results and , it would be of interest to get the authors' perspective on whether their results provide any insights into directions for ecDNA-specific therapy development.

We agree - to rectify this we have completely rewritten the discussion to highlight the novel aspects of the study with respect to ecDNA biology and the challenges of designing therapeutics that target both the formation and maintenance of ecDNA.

Minor comments

1. Page 9, reference to Figure 1D should probably to Figure 2D? Also page 9, reference to Extended Figure 2A should be Extended Figure 3A?

We thank the reviewer for bringing this to our attention and have corrected these errors.

2. Typo temozolamide -> temozolomide

We have corrected this spelling error.

Roel Verhaak

Referee #3 (Remarks to the Author):

The manuscript by Bailey et al describes computational analysis of ~15,000 cancer genomes from the Genomics England project. The analyses are what a team of excellent bioinformaticians would do with this large dataset. The authors identify likely ecDNAs from WGS and carrying out a variety of correlative analysis, including (i) estimation of frequency of samples with ecDNA across tumor types and subtypes, (ii) characterization of genes and functional elements encapsulated in ecDNAs, (iii) association with many factors including TMB, mutational signatures, MMRD, HRD, and survival.

We appreciate the reviewer's appreciation of the bioinformatic strengths, and appreciate the thoughtful suggestions, which we have addressed in the revised manuscript.

Despite the extensive analyses, however, I'm unable to pinpoint what new insights have been uncovered about ecDNAs, given many excellent papers in this area including those by some of the authors on this paper. Examples:

- Previous analysis of TCGA data (Kim et al, Nature Genetics, 2020) had estimated the frequencies of samples across 30+ tumor types using >3200 samples. The overall frequency of 17% in the present paper is similar to the 14% in the TCGA paper, as the authors note. The 5X increase in sample size allowed more accurate estimation of frequencies per tumor type as well as for subtypes, but I am not sure if this is a substantial advance.

We appreciate the reviewer spurring us on to more clearly present the aspects of this paper that are novel and important, and have done so in the revision. There are multiple, highly novel, and important findings presented. First, the frequency of ecDNA in breast cancer comes as a major surprise. Second, the frequency of immunomodulatory element amplification was hinted at by our previous work, but never demonstrated at scale, nor its relationship to oncogene amplification revealed. Further, so much of our work on altered ecDNA transcription highlights combinatorial interactions in trans between regulatory elements such as enhancers on one ecDNA and interactions with oncogene promoters on another. To see it at scale is an important finding. Further, the elucidation of key mutational signatures, including many not seen before, is also quite important.

As stated, the overall frequency of 17% we identify here is similar to the 14% estimate in the TCGA paper. There would be great concern about the methodology if the answer were otherwise. We believe we have built on the observations made in that paper as we present novel findings. Specifically:

- We describe immunomodulatory and regulatory ecDNA previously undescribed in patient cohorts.
- We describe additional cancer types that have not previously been described, such as triple negative breast cancer, estrogen receptor positive breast cancer, myxofibrosarcomas, neuroendocrine tumours, chordomas, astrocytomas, cholangiocarcinoma and mesothelioma. We also emphasise the importance of the observation that 46% of the HER2+ breast cancer cohort were predicted to have ecDNA and discuss implications for treatment resistance following this observation.
- We analyse the impact of stage on the estimations that we make, finding that ecDNA are present at increased frequencies in metastatic and late-stage disease.
- Following this observation we control for stage in our Cox proportional hazards model (something not done in Kim et al.), and adjust for underlying genome instability through wGII. The conclusion here being that ecDNA is not just a proxy for underlying genome instability.

- Another new, key observation made is the impact of treatment on ecDNA. Adjusting for underlying tumour type and stage we find that ecDNAs are enriched in patients who have received prior targeted or chemotherapy leading to the assertion that treatment either creates a selection pressure that drives a clonal sweep in ecDNA that were present pre-treatment, or that treatment has induced *de novo* ecDNA. This has been observed in cell lines with the formation of *DHFR* ecDNA following treatment with methotrexate (Shoshani et al. 2022) but not observed in patients.

These are where we believe the similarities with Kim et al. are found and where we made new observations on the back of that study.

We believe the paper takes a substantial departure from Kim et al. after the first half of Figure 1, as we describe how mutations on ecDNA also undergo positive selection, and how the oncogene enrichment is stronger in ecDNA than it is in intrachromosomal amplifications. We quantify the impact of sampling multiple regions on the detection of ecDNA. Beyond this, we assert that ecDNA differs in its complexity and arises from multiple mechanisms including as a result of chromothripsis and through episomes. We show that *TP53* is most strongly associated with ecDNA formation across multiple cancer types (Figure 3a) and negatively associated mismatch repair deficiency and DNA polymerase delta and epsilon proofreading deficiency (Figure aA). We believe these observations are novel findings.

- There are some differences in the list of genes found, but the top ones seem to be not much different from Kim et al. as far as I can tell (enrichment of oncogenes shown in Figure 1B is trivial at this point).

- We were heartened by the fact that broadly, the list of oncogenes described in Kim et al is similar to our own. We take this observation further by describing the phenomenon of multiple oncogenes on the same ecDNA and oncogenes that arise on independent ecDNAs seen in Extended Data Fig. 5c (we have elaborated on this further in the paper).
- We list most frequently found genes in immunomodulatory ecDNA and in lncRNAs, however we feel that the description of genes on ecDNA is a small part of the overall paper.

- Enrichment of immunomodulatory genes in ecDNA was already described by some of same the authors (Luebeck et al, Nature, 2023). A small number of ecDNAs contained lncRNA, enhancer, or promoter, but their role is not explored further.

Both immunomodulatory and regulatory ecDNA have been observed previously. Luebeck et al. described immunomodulatory ecDNA in Barrett's oesophagus⁴ and Hung et al. describe the presence of (*FGFR2*) enhancer only ecDNA in SNU16 cells.²¹ We expand on these observations by describing regulatory ecDNA in patients (not previously done) and describing immunomodulatory ecDNA across multiple tumour types, providing an estimate of how common immunomodulatory ecDNA are in human cancer. Furthermore, we integrate data on T cell infiltration to try and understand the functional

consequences of these ecDNA subspecies. We also show that compared with other ecDNA types, regulatory ecDNAs are enriched for regulatory regions.

Correlative analysis with other variables:

- I will take TMB as an example because this is the first item in the section, but I think some other analyses suffer from the same problem. The authors show that the median TMB is 4.34 for the samples with ecDNA and 2.24 for those with non-ecDNA focal amplification. First, as Figure 4 shows, there is a huge range for TMB and, with so many samples, even a small difference will give statistical significance (even for rank-based tests). Second, I'm not sure if it's possible to infer much from this difference because there are many other covariates that may have contributed to the difference. For instance, the range of TMB is very different across tumor types, and the sample size per tumor type is uneven across tumor types. I would think that this makes the conclusion dependent on the composition of the cohort, i.e., for a random subset of tumor types, sometimes it will be significant and other times it will not. Thus, one must control for the tumor type, i.e., by perform this comparison within each tumor type and see a consistent difference, I think. Also, do the focal amplifications with and without ecDNA differ in other characteristics such as size and amplitude? If they are not different, is it possible that other differences you see between the groups due to those (e.g., more oncogenes covered in non-ecDNA amplifications or ecDNAs have more copies of oncogenes) rather than the mechanistic aspect of ecDNA generation?

We thank the reviewer for this observation and agree the correlative analysis requires some care in the interpretation of the results and strong consideration as to the possible confounding factors that may mask any true difference between the groups. We have taken great care to adjust for relevant confounders in all models in our analysis. To the reviewers' point regarding Figure 4a; we agree that what is presented is an adjusted figure (however subsequent figures adjust for relevant confounders), we have rectified this in the following way. We first show that TMB and age are correlated:

Reviewer Figure 3.1: Boxplots showing TMB by age and grouped by amplification status.

Following this we performed a regression analysis that looked at the effect of increasing TMB on the detection of ecDNA, controlled for tumour type, purity and age. TMB was categorised into deciles. We found that the odds of detecting ecDNA was increased for every increasing TMB decile (tmb_decile in the plot) compared with chromosomal amplifications (OR 1.04 95% CI 1.02-1.06) and no amplifications (OR 1.14 95%CI 1.13-1.16):

Reviewer Figure 3.2: Regression analyses investigating TMB by amplification category

Forest plots showing (left) associations with increased TMB decile in tumours with chromosomal amplifications compared with those with no focal amplifications. (middle) Associations with increased TMB decile in tumours with ecDNA compared with those with no focal amplifications. (right) Associations with increased TMB decile in tumours with ecDNA compared with those with chromosomal amplifications.

To simplify this we have summarised only the ORs for increasing TMB in each category as part of figure 4b. These represent the individual odds ratios of increasing TMB deciles controlled for tumour type, purity and age:

Figure 4b: ecDNA mutations and their timing

Forest plot depicting the odds ratio of an increased TMB decile according to the presence of ecDNA or chromosomal amplifications adjusted for purity, age, and tumour type.

- Correlation with survival was also done in Kim et al and I think the findings are similar.

We do perform a survival analysis as seen in Kim et al., however we attempt to address some questions not addressed in that paper, such as the impact of stage and underlying genome instability on survival. We control for many more confounding factors in the cox proportional hazards model.

Overall, it is totally worthwhile to replicate their previous analysis on this much larger dataset, and the authors have done an extensive analysis. There are some new observations such as more extensive gene lists, timing of ecDNA formation with respect to some signatures, and some others. But the amount of new insights seems limited to this reader.

Minor points:

- Focal amplifications were defined as regions between 50kb and 20Mb in size. 20Mb seems to very large, >40% of the smallest chromosome.

A wide variation in the definition of focal amplification size is present in the field of aneuploidy and somatic copy number alteration research. As such we appreciate the reviewer's comments and the need to place our thresholds within the context of the literature. Therefore, we have provided information regarding the size of the amplifications we observe allowing readers to examine our results in the context of this debate (see Extended Data Fig. 3c and 3e, reproduced below).

C

E

Extended Data Fig. 3c,e: Distribution of ecDNA size according to tissue type and oncogene present

(top) Box plots showing the copy number of ecDNA grouped by tissue type. (bottom) Box plots of ecDNA size in megabases grouped by the oncogenes present on the ecDNA and tissue type.

We took 20Mb as a more stringent cut off to the definition of focal amplification in the GISTIC algorithm used in TCGA and developed at the Broad Institute (default here is $0.98 * \text{chromosomal arm}$)

<https://broadinstitute.github.io/gistic2/>, and in the original GISTIC paper by Zack et al. (2013), where the cut off for focal amplification was 50% of the chromosome arm size. Moreover, in Beroukhim et al. 2010, focal SCNAs occurred at a frequency inversely related to their lengths, with a median length of 1.8 Mb, and range from 0.5 kb to 85 Mb.

- The authors include foremost experts on ecDNAs including their computational detection. But how accurate are the ecDNA calls? If I understand correct, AmpliconArchitect is a graph-based method that relies on accurate segmentation of copy number, followed by construction of the possibly non-unique graph paths based on discordant reads before making an educated guess. Are there some estimates of false positives and false negatives (perhaps based on long-read data or other validation methods)? I understand that it is an inexact method, but I ask because the frequency estimates per tumor type are given down to 0.1% throughout the paper, and I wasn't sure whether the authors actually believe that they can estimate with that accuracy.

We thank the reviewer for these comments. Yes, we can identify ecDNA structures with a high level of confidence, in fact, with a sensitivity of 90% and specificity of 77.8%, as we have previously demonstrated² as we have benchmarked it against the gold standard of FISH on metaphase spreads of cancer.

Reviewer Figure 3.3: AmpliconArchitect and AmpliconClassifier performance on cytogenetically validated cell lines using metaphase FISH.

Each point represents the number of ecDNA detected using metaphase FISH for an amplicon detected with sequencing. In total, results for 67 genes from 42 cytogenetically validated cell lines are shown. Each cell line is subjected to next generation sequencing then analysed with AmpliconArchitect and AmpliconClassifier (AC). AC then assigns the amplicons from each cell line to one of four categories: linear, complex non-cyclic, breakage

fusion bridge, or ecDNA. Please note that this figure is from another manuscript under review. We provide these data for the reviewer and note that an earlier version of this plot and its source data is in Kim et al. 2020 “Extrachromosomal DNA is associated with oncogene amplification and poor outcome across multiple cancers” Nature Genetics.

- The figures had tiny fonts and poor resolution. I could barely read the text within the figures even when viewing the pdf full screen on my desktop monitor.

We apologise for the small fonts and poor resolution of the figures in the previous version of our manuscript. All figures are now of sufficient resolution and have a minimum font size that conforms with Nature’s specifications.

References

1. Sosinsky, A. *et al.* Insights for precision oncology from the integration of genomic and clinical data of 13,880 tumors from the 100,000 Genomes Cancer Programme. *Nat. Med.* **30**, 279–289 (2024).
2. Kim, H. *et al.* Extrachromosomal DNA is associated with oncogene amplification and poor outcome across multiple cancers. *Nat. Genet.* **52**, 891–897 (2020).
3. Martincorena, I. *et al.* Universal Patterns of Selection in Cancer and Somatic Tissues. *Cell* **171**, 1029–1041.e21 (2017).
4. Luebeck, J. *et al.* Extrachromosomal DNA in the cancerous transformation of Barrett’s oesophagus. *Nature* (2023) doi:10.1038/s41586-023-05937-5.
5. Rose, J. C. *et al.* Disparate pathways for extrachromosomal DNA biogenesis and genomic DNA repair. Preprint at <https://doi.org/10.1101/2023.10.22.563489> (2023).
6. Cortés-Ciriano, I. *et al.* Comprehensive analysis of chromothripsis in 2,658 human cancers using whole-genome sequencing. *Nat. Genet.* **52**, 331–341 (2020).
7. Song, K. *et al.* Plasticity of Extrachromosomal and Intrachromosomal *BRAF* Amplifications in Overcoming Targeted Therapy Dosage Challenges. *Cancer Discov.* **12**, 1046–1069 (2022).
8. Luebeck, J. *et al.* AmpliconReconstructor integrates NGS and optical mapping to resolve the complex structures of focal amplifications. *Nat. Commun.* **11**, 4374 (2020).
9. Turner, K. M. *et al.* Extrachromosomal oncogene amplification drives tumour evolution and genetic heterogeneity. *Nature* **543**, 122–125 (2017).
10. Lange, J. T. *et al.* The evolutionary dynamics of extrachromosomal DNA in human cancers. *Nat. Genet.* **54**, 1527–1533 (2022).

11. Shoshani, O. *et al.* Chromothripsis drives the evolution of gene amplification in cancer. *Nature* **591**, 137–141 (2021).
12. Hung, K. L. *et al.* Targeted profiling of human extrachromosomal DNA by CRISPR-CATCH. *Nat. Genet.* **54**, 1746–1754 (2022).
13. Bentham, R. *et al.* Using DNA sequencing data to quantify T cell fraction and therapy response. *Nature* **597**, 555–560 (2021).
14. Deshpande, V. *et al.* Exploring the landscape of focal amplifications in cancer using AmpliconArchitect. *Nat. Commun.* **10**, 392 (2019).
15. Contino, G., Vaughan, T. L., Whiteman, D. & Fitzgerald, R. C. The Evolving Genomic Landscape of Barrett’s Esophagus and Esophageal Adenocarcinoma. *Gastroenterology* **153**, 657-673.e1 (2017).
16. Barretina, J. *et al.* The Cancer Cell Line Encyclopedia enables predictive modelling of anticancer drug sensitivity. *Nature* **483**, 603–607 (2012).
17. Yuan, K., Macintyre, G., Liu, W., PCAWG-11 working group & Markowitz, F. *Ccube: A Fast and Robust Method for Estimating Cancer Cell Fractions*. <http://biorxiv.org/lookup/doi/10.1101/484402> (2018) doi:10.1101/484402.
18. Degasperi, A. *et al.* Substitution mutational signatures in whole-genome–sequenced cancers in the UK population. *Science* **376**, abl9283 (2022).
19. Liao, J., Xu, J., Feng, K., Lai, W. & Wen, X. MiR-623 links lncRNA RP11-89 and cyclin D1 to regulate the proliferation of glioblastoma cells. *Int. J. Neurosci.* 1–7 (2022) doi:10.1080/00207454.2022.2098734.
20. Luo, W. *et al.* lncRNA RP11-89 facilitates tumorigenesis and ferroptosis resistance through PROM2-activated iron export by sponging miR-129-5p in bladder cancer. *Cell Death Dis.* **12**, 1043 (2021).
21. Hung, K. L. *et al.* ecDNA hubs drive cooperative intermolecular oncogene expression. *Nature* (2021) doi:10.1038/s41586-021-04116-8.

Reviewer Reports on the First Revision:

Referees' comments:

Referee #1 (Remarks to the Author):

I have thoroughly went through the revised manuscript and the point-by-point response letter. The authors provided excellent clarifications and new information, and addressed most of my concerns and questions. I understand that some of my comments are hard to address due to the study limitations and sample accessibility. Nevertheless, the comprehensive analyses performed further highlights the widespread role of ecDNA amplifications across cancer types. The study identifies several important novelties that encourage future efforts, including the relationship between amplifications and metastasis, immune response, and therapy resistance.

Overall, I enjoyed reading the revised manuscript and have no further comments.

Referee #2 (Remarks to the Author):

Through the revision, the authors have strengthened their manuscript and made it more cohesive. The Extended Data Figures are rich in detail and provide substantial additional information.

Ext. Data Fig. 5A shows that ecDNA amplifications are of higher copy number compared to chromosomal amplifications, with Ext. Data Fig. 7D providing additional support for individual oncogenes. It would be valuable to this reviewer to include Reviewer Figure 2.7, to demonstrate that the significantly higher number of oncogenes per ecDNA compared to chromosomal amplifications is copy number independent.

Referee #3 (Remarks to the Author):

The authors have done an excellent job of responding to not only my comments but to those of other reviewers. It took a considerable amount of time to read through all the responses, and so I can only imagine the effort it took to put them together. I still feel that the advances in this paper are more incremental than what one would like to see in a Nature paper, given the many excellent papers by some of the same authors that originally reported most of the key points. But with the substantially larger sample size, this paper could be a standard reference on the topic for many papers to come.